# Critical assessment of LC3/GABARAP ligands used for degrader development and ligandability of LC3/GABARAP binding pockets

Martin P. Schwalm [1,2,3], Johannes Dopfer [1,2], Adarsh Kumar [1,2], Francesco A. Greco [1,2], Nicolas Bauer [1,2], Frank Löhr [4], Jan Heering [5], Sara Cano-Franco [6,7], Severin Lechner [8], Thomas Hanke [1,2], Ivana Jaser [9], Viktoria Morasch [1,2], Christopher Lenz [1,2], Daren Fearon [10], Peter G. Marples [10], Charles W. E. Tomlinson [10], Lorene Brunello [6,7], Krishna Saxena [1,2], Nathan B. P. Adams [9], Frank von Delft [10], Susanne Müller [1,2], Alexandra Stolz [6,7], Ewgenij Proschak [1,5], Bernhard Kuster [8], Stefan Knapp [1,2,3] ✉ & Vladimir V. Rogov [1,2] ✉

Recent successes in developing small molecule degraders that act through the ubiquitin system have spurred efforts to extend this technology to other mechanisms, including the autophagosomal-lysosomal pathway. Therefore, reports of autophagosome tethering compounds (ATTECs) have received considerable attention from the drug development community. ATTECs are based on the recruitment of targets to LC3/GABARAP, a family of ubiquitin-like proteins that presumably bind to the autophagosome membrane and tether cargo-loaded autophagy receptors into the autophagosome. In this work, we rigorously tested the target engagement of the reported ATTECs to validate the existing LC3/GABARAP ligands. Surprisingly, we were unable to detect interaction with their designated target LC3 using a diversity of biophysical methods. Intrigued by the idea of developing ATTECs, we evaluated the ligandability of LC3/GABARAP by in silico docking and large-scale crystal-lographic fragment screening. Data based on approximately 1000 crystal structures revealed that most fragments bound to the HP2 but not to the HP1 pocket within the LIR docking site, suggesting a favorable ligandability of HP2. Through this study, we identified diverse validated LC3/GABARAP ligands and fragments as starting points for chemical probe and ATTEC development.

Targeted protein degradation (TPD) has received a great deal of attention due to the potential of chemical degraders to become a new modality in drug development[1,2]. Two major strategies are currently in use: molecular glues (glues) and PROTACs (PROteolysis TArgeting Chimeras)[3]. Glues bind to an E3 ligase and recruit a protein of interest (POI) with their solvent exposed moieties. This chemically induced proximity of the POI to the E3 ligase leads to POI ubiquitination and subsequent proteasomal degradation[4]. PROTACs induce selective

degradation by a similar mechanism, but they are chimeric molecules using two distinct ligands, one binding to an E3 ligase and one to the POI, connected by an appropriate linker moiety[5]. PROTACs and glues have vastly expanded the druggable target space by being able to bind anywhere on a POI, and not just a specific binding site relevant to disease pathogenesis. Additionally, their properties of acting catalytically and often highly selectively to degrade the POI holds the promise that these new drug modalities could be effective at very low compound concentrations reducing drug toxicity[6]. Inspired by the potential of selective degraders in drug development, new pathways have been explored to expand the toolbox that can be used for the design of these molecules. Among them are LYTACs[7] (LYsosome-TArgeting Chimeras) for the degradation of membrane proteins as well as ATTECs[8] (AuTophagosome TEthering Compound) which hijack the autophagy/lysosomal pathway for selective degradation of POIs. Excitingly, these ubiquitin independent systems would also allow degradation of large organelles, pathogenic bacteria and macromolecular protein complexes.

Macro-autophagy (Autophagy hereafter) is a fundamental cellular process regulating degradation and recycling of cellular components[9,10], also allowing the removal of bulky cytosolic cargo, such as large protein complexes, lipid droplets, portions of and whole organelles, and even bacteria that invaded the cytoplasm[11,12]. Cargo degradation is achieved by enclosure into a double-membrane vesicle (autophagosome) followed by autophagosome trafficking to the lysosome for degradation. Autophagy is an evolutionarily conserved complex process orchestrated by ~40 autophagy-related (Atg)

proteins, which include, among others, the autophagy-related ubiquitin-like modifiers (Atg8 in yeast) LC3A, LC3B, LC3C, GABARAP, GABARAPL1, and GABARAPL2 proteins (LC3/GABARAP hereafter)[13]. LC3/GABARAP recruit cargo-receptor-complexes by a short sequence motif within the receptors called the LIR (LC3-Interacting Region), mediating autophagosomal recruitment and degradation by interaction with the LDS (LIR docking site) (Fig. 1, left plot)[14–16]. The p62 LIR motif binds LC3/GABARAP proteins over this site and serves as a referential LIR motif in many research works[14]. In addition to the LDS, LC3/GABARAPs possess an additional interaction site located at the opposite face of the LDS (Fig. 1a, right plot), reminiscent of a hydrophobic binding patch present in ubiquitin. Accordingly, this site binds to several ubiquitin-interacting motifs (UIM) and was therefore named UIM docking site (UDS)[17]. Similar to the E3 ligase dependent TPD, small molecules binding to LDS and the UDS probably interferes with cargo recruitment to LC3/GABARAP proteins and could be developed into small molecule degraders by recruitment of targets to the autophagosome[18].

However, the discovery of potent LC3/GABARAP ligands has remained challenging, possibly due to the conformational plasticity of LC3/GABARAP, resulting in at least partial occlusion of the LDS[19]. First LC3A/B targeting reversible ligands such as the antibiotic Novobiocin[20], covalent lysine targeting ligands[21] as well as a number of low molecular weight fragments (summarized in Fig. 1c) have been described binding to the LDS[22], but no potent ligands nor ligands for the four remaining LC3/GABARAPs have been described. Interestingly, the first ATTECs were reported in 2019, suggesting that the autophagic

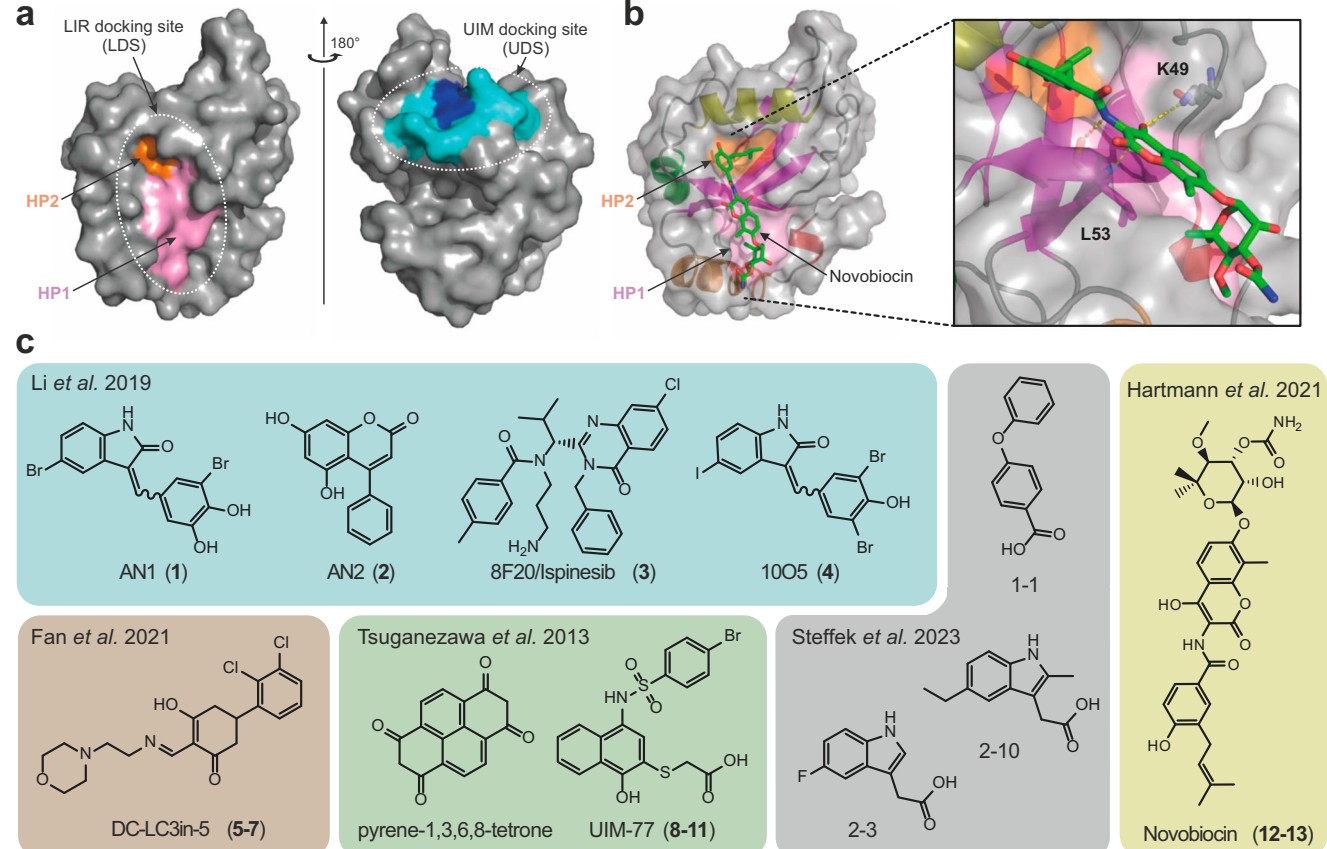

**Fig. 1 | Structural organization of human Atg8 family proteins and reported binder. a** Left panel depicting the LIR docking site (LDS) of LC3A comprised of hydrophobic pocket 1 (HP1) in pink and hydrophobic pocket 2 (HP2) in orange. Right panel displaying the UIM docking site (UDS) in blue/cyan (PDB ID 3ECI). **b** Dihydronovobiocin bound to LC3A with an enlarged panel of the binding interface by interactions towards K49 and L53 and binding to the HP2 via hydrophobic interactions (PDB ID 6TBE). **c** Chemical structures of compounds, published to bind to LC3/GABARAPs with exemplary structures shown with all structures depicted in Supplementary Fig. 1.

degradation pathway can be exploited for the design of selective degrader small molecules[8]. In this study, the authors presented a mechanism, whereby small molecules mediated the autophagosomal degradation of mutant huntingtin protein (mHTT) through LC3B recruitment. Subsequently, the identified compounds (1O5/compound 4 and 8F20/compound 3) have been used as LC3 ligands by the same research team for ATTEC design, targeting diverse proteins including the bromodomain containing 4 (BRD4)[23] and nicotinamide phosphoribosyltransferase (NAMPT)[24], suggesting a broad utility of these ligands for degrader design. Of note, compound 3 was previously published as kinesin-like mitotic motor protein inhibitor, and can also be found under the name Ispinesib/SB715992 which was described by Davis et al. in ref. 25. However, there are no selective tool compounds targeting LC3/GABARAP and acting as autophagy pathway modulators to control the proposed degradation mechanism. In addition, we found that thorough biophysical characterization, evaluation, cellular target engagement or cell-based controls for the developed ATTEC ligands were largely lacking.

Driven by our interest in the development of new degrader molecules utilizing the autophagy pathway, we rigorously evaluated current LC3 ligands (Supplementary Fig. 1) by biophysical binding assays in vitro as well as in cell lysates. Surprisingly, we were unable to detect any interaction for some of the published LC3 ligands using a comprehensive panel of assay systems suggesting that these ligands may act through alternative mechanisms. Intrigued by the concept of hijacking the autophagosomal pathway through target recruitment to LC3/GABARAP, we extensively evaluated the ligandability of the LDS by in silico screening of an in-house compound library followed by biophysical validation as well as by high-throughput crystallographic fragment screening. The campaigns revealed good ligandability of the HP2 site within the LDS, a shallow binding pocket interacting with hydrophobic residues in the LIR motif. In addition, poor accessibility of the HP1 site which interacts with aromatic residues in the LIR motif, and initial ligands targeting for UDS binding site are found, which natural binding partners remain understudied. Our data not only demonstrated the ligandability of all LC3/GABARAPs, but also presented a strategy for the development and evaluation of LDS and UDS ligands as starting points for future ATTEC development.

## Results

### Characterization of known LC3/GABARAP-binding ligands

To assess target engagement of reported LC3 ligands, we carried out diverse biophysical binding assays, including fluorescence polarization (FP), isothermal titration calorimetry (ITC) and nuclear magnetic resonance (NMR). We compiled a comprehensive set of ligands reported in the recent literature for this comparative interaction study, including all ligands used for ATTEC design (AN1 (1), AN2 (2), 8F20 (3) and 1O5 (4))[8], covalent ligands targeting the side chain amine of K49 within the LDS (compounds 5–7)[21] and four analogs of ligands and fragments that have been published to disrupt the p62:LC3 interaction (compounds 8–11)[26]. Additionally, we included Novobiocin (12) and Dihydronovobiocin (13), which we reported previously as a ligand of LC3A and LC3B[20]. We also included five LIR peptides spanning a wide affinity range as positive controls. A full list of selected LC3 ligands has been compiled in Supplementary Fig. 1 and representative ligands as well as the targeted ligand pockets are shown in Fig. 1.

Initially, we used temperature shift assays as a binding assay to evaluate small molecule interaction with the LC3/GABARAP family. However, recorded temperature shifts were relatively small, including data measured for control peptides and we therefore deemed this assay as not suitable for the detection of LC3/GABARAP ligands (Supplementary Fig. 2a). Next, we established an FP assay utilizing the p62 LIR peptide linked to a Cy5 fluorophore as a tracer molecule which interacts with all LC3/GABARAPs via the LDS (reviewed in refs. 11,14). This assay can be used to assess whether compounds interact with the

LDS of LC3/GABARAPs. Binding via alternative binding sites such as UDS cannot be monitored by this assay as this binding site does not interact with the p62 LIR peptide used. Dose-dependent titrations using all LC3/GABARAPs yielded assays with good signal-to-noise ratio and resulted in measured $K_D$ values for the tracer between 3 and 17 μM across the human Atg8 family. Thus, this displacement assay was suitable for screening and binding affinity determination of ligands in the low micromolar $K_D$ range (Supplementary Fig. 2b). We evaluated the established set of ligands (1–13) against all LC3/GABARAPs. A representative data set for LC3B is shown in Fig. 2a and all data are included in Supplementary Fig. 2c, d. Consistent with data published previously, compound 12 (Novobiocin) bound to LC3A and LC3B with highest affinity $K_I$ values of 17.4 and 48.4 μM, respectively (Fig. 2b). Next, we focused on the ligands that have been used widely for ATTEC development (1–4). The dose-dependent titrations of these compounds against LC3/GABARAP family members are shown in Fig. 2c. Surprisingly, no detectable binding to the published targets LC3A and LC3B was observed for all four ATTEC handles up to a concentration of 100 μM. However, weak interaction was detected for 1O5 (4) binding to GABARAPL2 but not to its designated target LC3B. Due to the bright color of compounds 1 and 4 and the discrepancy to literature data, we validated these results further by direct binding assays using 2D NMR titration experiments with $^{15}N$ labelled LC3B protein (Fig. 2d, e; left plots). This technique does not rely on the competition of a tracer peptide and enables the detection of allosteric LC3 binders and ligands binding outside the LDS. In agreement with our FP data, the binding of Novobiocin (12) caused large chemical shifts perturbations (CSP) within the LIR binding pocket (Fig. 2d, e). However, none of the compounds 1–4 resulted in significant CSP (comparable with Novobiocin) even at high compound concentrations in agreement with our FP binding data. Analysis of the small CSP HN resonances in the backbone, induced by 4 revealed that they are predominantly within HP2 with estimated $K_D$ values of ≥ 200 μM for LC3B (Supplementary Fig. 3).

We therefore performed further optimized biophysical analyses, using these methods for LC3A interaction with 1–4. To measure compound interaction via spectral shift assay, we chose cysteine labelling to avoid complications associated with lysine labeling due to the presence of these residues in the binding sites. After successfully setting up the assay, we were unable to reproduce the published binding data. However, lysine labeling was used in the literature in combination with high protein concentrations of 500 nM which exceeds the recommended range 10–100 fold (Supplementary Fig. 4a)[8]. Next, ITC was used as a label-independent method for binding verification. Here, in agreement with earlier experiments, Novobiocin (12) revealed binding with a $K_D$ (6.7 μM for LC3A), while titrations with AN2 (2) and 8F20 (3) did not yield significant binding heats. Additionally, we also investigated binding of AN1 (1) and 1O5 (4) as well as compound (8) by ITC, but these ligands induced protein precipitation, rendering ITC $K_D$ determination impossible (Supplementary Fig. 4b). To reproduce direct binding through FP as reported, we synthesized 8F20 (17)- and 1O5 (16)-based dye-linked tracer molecules utilizing the same linker attachment point as for ATTEC development (Supplementary Fig. 4c)[24]. Using the reported experimental setup for establishing an FP assay, we successfully reproduced the tracer-LC3 interaction. However, we were unable to obtain displacement data using the parent compound, a standard control in FP assays. The only experimental difference in our FP assay setup was the addition of 0.05% Tween-20, which is routinely used[27] to suppress unspecific binding of compounds to proteins. Under these conditions, we did not detect any binding, indicating unspecific tracer-LC3A interaction, whereas without Tween-20 some (unspecific) binding was observed (Supplementary Fig. 4d).

Even though we did not observe any binding of compounds 1–4 to LC3 proteins, we were able to measure weak interaction of 1O5 (4) with GABARAP family members (Fig. 2c). These data motivated us to

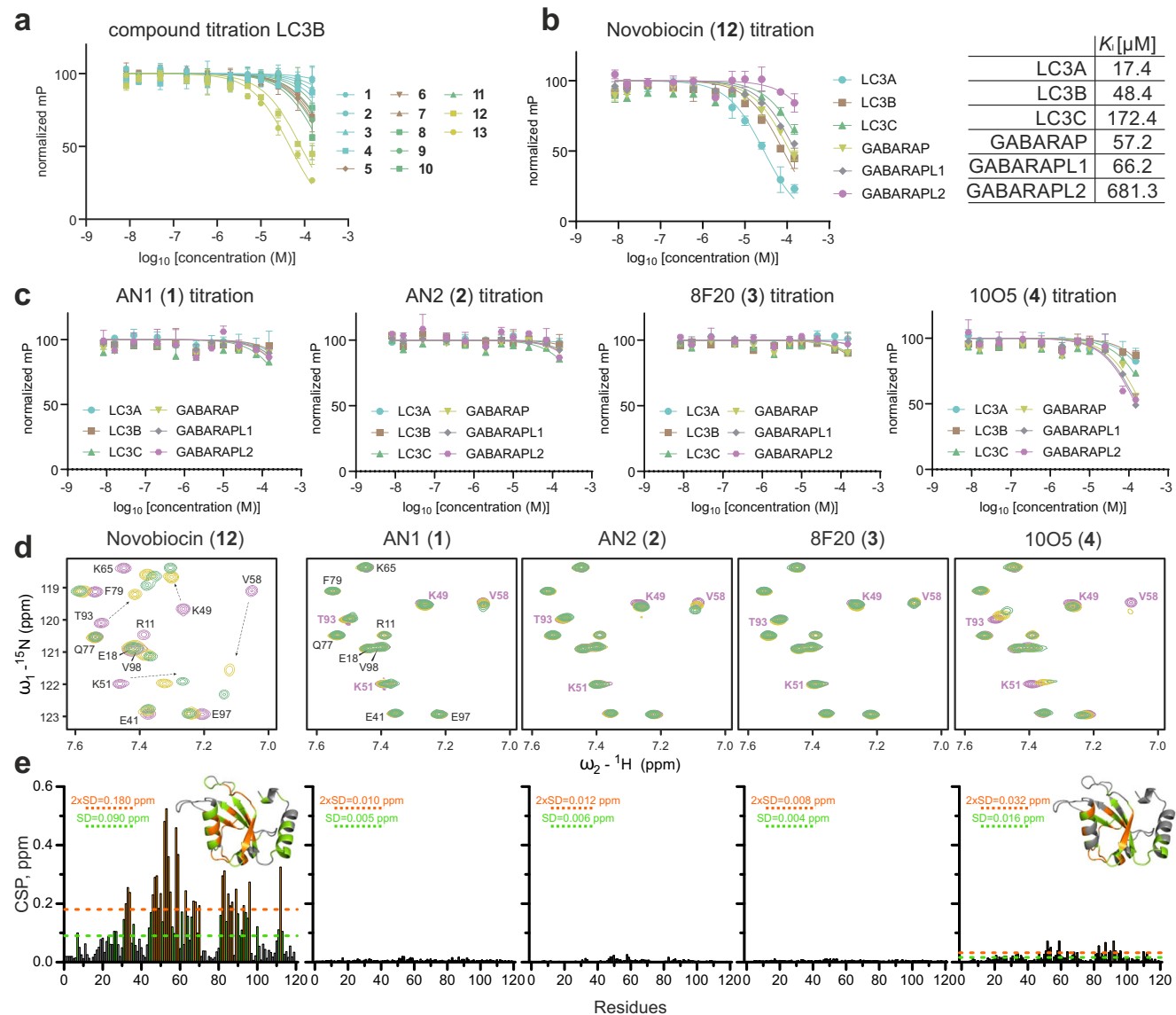

**Fig. 2 | Biophysical characterization of compound-LC3/GABARAP interactions.**
**a** Fluorescence polarization (FP) displacement assay titrations for compounds **1–13** measuring interaction with LC3B. Assays were run as technical replicates ($n = 2$) with data presented as mean values +/− SD of each data point. **b** FP data measuring the binding of Novobiocin to all six human LC3/GABARAP proteins using a p62 LIR-based tracer. Individual dose dependent titrations are depicted in Supplementary Fig. 2. Titrations were run as technical duplicates ($n = 2$) with data presented as mean values +/− SD of each data point. **c** Fluorescence polarization assay displacement curves for compound **1–4** against all LC3/GABARAPs against a p62 LIR-based tracer. Assays were run as technical duplicates ($n = 2$) with data presented as mean values +/− SD of each data point. **d** Interaction between LC3B and compounds **1–4** investigated by NMR. Representative fingerprint areas around the key K51 and V58 backbone HN resonances of 2D $^1$H-$^{15}$N correlation spectra for free LC3B

(magenta) and LC3B containing control compound **12** (recorded at 700 MHz spectrometer as [$^1$H-$^{15}$N] fHSQC experiment) and compounds **1–4** (recorded at 800 MHz spectrometer as [$^{15}$N,$^1$H] BEST-TROSY) at 1:1 (yellow) and 1:2 (green) molar ratios are shown in overlay. Mapping of backbone HN resonances on LC3B sequence and structure are depicted in Supplementary Fig. 3. **e** Left plot: chemical shifts perturbations (CSP) values, induced by **12** at molar ratio 1:2, are plotted against LC3B residue numbers and mapped on 3D-structure (insert). The light green dashed line indicates the standard deviations (SD) over all residues, the orange dashed line indicates double SD values; residues with small (CSP < SD), intermediate (SD < CSP < 2xSD) or strong (2xSD < CSP) CSP values are marked in grey, light green and orange, respectively. Other plots: compounds **1–3** induce insignificant CSP values at molar ratio 1:2, compound **4** induces small CSP around LC3B residues forming HP2 (right plot). Source data for (**a**, **b**, **c**, **e**) are provided as a Source Data file.

further investigate the interaction of compounds **1–4** with GABARAPL2 by 2D NMR, which confirmed interaction in the HP2 fingerprint area depicted in Fig. 3a, b (full analysis is shown in Supplementary Fig. 5). Indeed, we observed that only 10O5 (**4**) was able to interact with GABARAPL2 in NMR titration experiments with estimated $K_D$ values in the 15–30 μM range. Due to the weak interaction with this Atg8 family member, it is not likely that the observed degradation of mHTT was mediated by binding of 10O5 (**4**) to GABARAPL2[8].

Covalent ligands **5–7** only showed weak interaction with LC3/GABARAPs in our FP assay and due to the irreversible nature of this

interaction, the binding of these ligands might be strongly time dependent. We therefore evaluated the ligands **5–7** by ESI mass spectrometry using all LC3/GABARAP isoforms. In agreement with published data[21], we detected a mass shift corresponding to the compounds bound to LC3/GABARAPs (Fig. 3c and Supplementary Fig. 6). The mass shift corresponded to a single modification, suggesting that compounds of this class selectively form a covalent bond with recombinant LC3 proteins at K49[21]. However, investigation of covalent binding of all six LC3/GABARAP isoforms with compounds **5–7** revealed no selectivity within the human Atg8 family members,

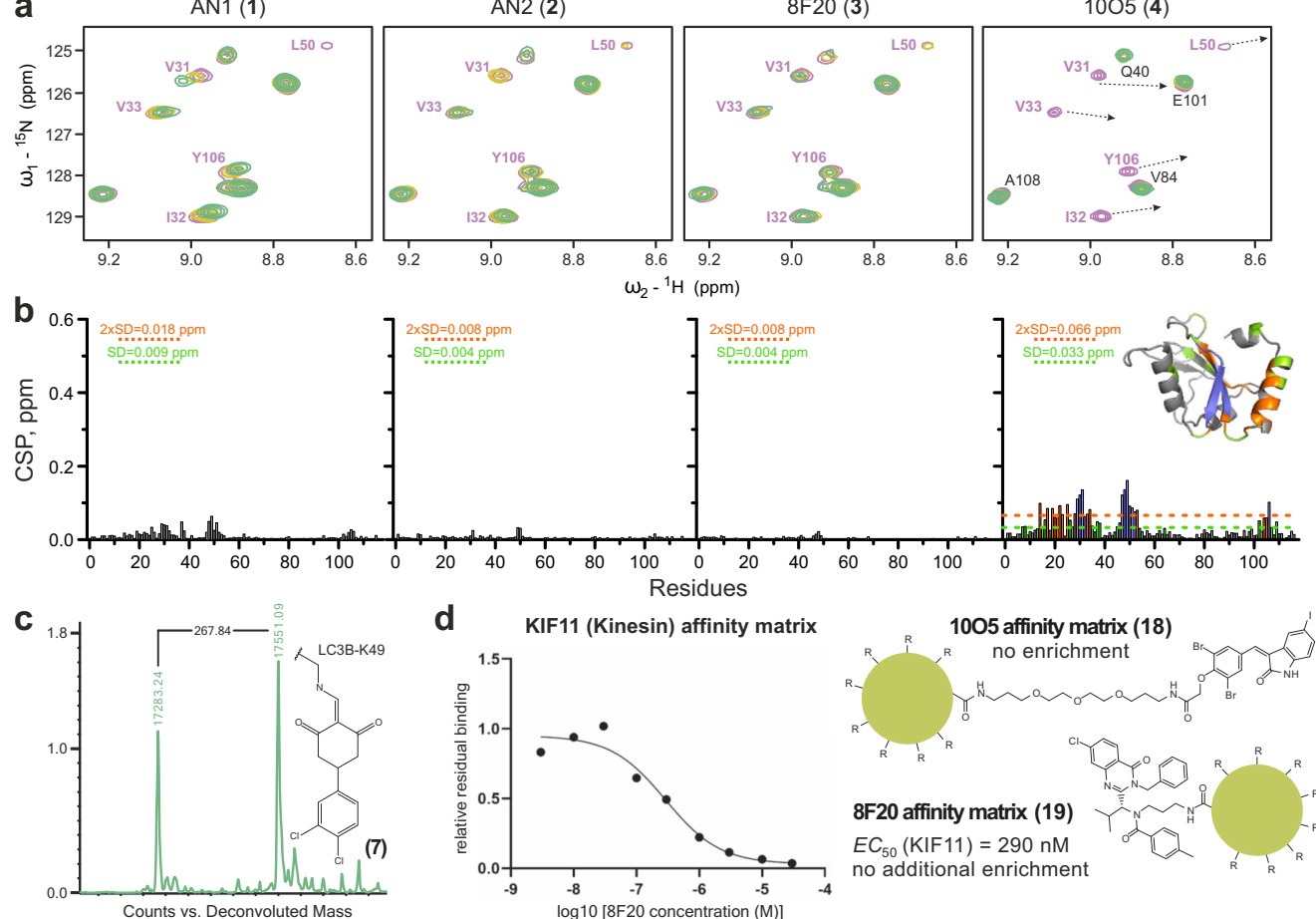

**Fig. 3 | Interactions of putative LC3 ligands measured by NMR and pull-down–MS based assays. a** Interaction between GABARAPL2 and compounds **1–4** investigated by NMR. Representative areas of GABARAPL2 [$^{15}$N,$^{1}$H] BEST-TROSY (recorded at 600 MHz spectrometer for compounds **1–3** and 900 MHz spectrometer for **4**) spectra around the key residues L50, I32 and Y106 backbone HN resonances are shown in overlay with free GABARAPL2 (magenta), and in presence of 1:1 (yellow) and 1:2 (green) molar ratio of each compound (indicated above each plot). Arrows in the plot for GABARAPL2:**4** interaction show directions of large chemical shift perturbations for the resonances which are in the intermediate exchange mode and could not be tracked until the latest titration steps, indicating the strongest interaction of these residues with GABARAPL2. Mapping of backbone HN resonances on GABARAPL2 sequence and structure and additional NMR data analysis are depicted in Supplementary Fig. 5. **b** CSP values, induced by compounds **1–4** at molar ratio 1:2, are plotted against GABARAPL2 residue numbers and mapped on 3D-structure (insert). The light green dashed line indicates the standard deviations (SD) over all residues, the orange dashed line indicates double SD values; residues with small

(CSP < SD), intermediate (SD < CSP <2xSD) or strong (2xSD <CSP) CSP values are marked in grey, light green and orange, respectively. The blue color for sequence- and 3D-mapping for compound **4** are for GABARAPL2 residues which undergo strong intermediate exchange mode (significant decrease of the resonances intensity upon titration with (**4**). **c** Exemplary mass spectrometry data expressing a mass shift of LC3B after treatment with compound **7**. Full data set for compounds **5–7** on all LC3/GABARAPs is depicted in Supplementary Fig. 6. **d** Chemoproteomic competition assays for target deconvolution of 8F20 (**3**) and 10O5 (**4**). Affinity matrices for pulldown experiments were synthesized by amide coupling yielding (**18**) and (**19**) attached to (NHS-activated) Sepharose beads. Competition experiments were performed with free compound **3** and PEG-linked compound **4** at nine concentrations and residual binding was calculated relative to a DMSO control. Of the over 4000 proteins identified, only KIF11 showed robust dose-dependent binding to 19 (EC$_{50}$ = 290 nM). Both the **18**- and **19**-based affinity matrix assays did not enrich for the reported targets LC3/GABARAP in HEK293T cell lysate. Source data for Fig. 3b, d are provided as a Source Data file.

raising the possibility of further off-targets of this compound class within the proteome, based on the reactivity of the chosen electrophile (Supplementary Fig. 6). To investigate covalent interactions of compounds **1** and **4** with LC3/GABARAP as recently reported for **1** with the E3 ligase DCAF11[28], we also studied the interaction of compound **1** and **4** with GABARAPL2 using ESI mass spectrometry, but no covalent adduct formation was detected in vitro (Supplementary Fig. 7).

Since compound **1–4** based ATTECs have been reported to induce significant target degradation in cellular assays[8,24,29,30], we were interested in possible mechanisms causing these intriguing effects. To identify possible targets of these small molecules, we used an amine-linker adduct at the same attachment point for linkers as in recently published ATTECs (Fig. 3d). For proteome-wide screening, we modified 10O5 (**4**) and 8F20 (**3**) with PEG-based linkers that can be immobilized on Sepharose beads to generate an affinity matrix for pulldown

experiments, resulting in compounds **18** and **19**. As expected, dose-dependent competition assays using the parent compound showed that the KIF11 inhibitor 8F20/Ispinesib (**3**) selectively bound to KIF11 in HEK293T lysates (EC$_{50}$ of 290 nM). No additional targets were detected for **19**, confirming excellent selectivity of this inhibitor for its designated target KIF11 (Fig. 3d). Using this pull-down assay, we therefore successfully validate interaction with the known target for Ispinesib, the kinesin-like mitotic motor protein KIF11. The only Atg8 homolog for which we detected a weak interaction, GABARAPL2, was not identified in pulled down assays using **18**, suggesting that the interaction with this Atg8 homolog was also weak in the cellular context. We also used tracers **16** and **17** to measure cellular LC3A and LC3B target engagement implementing the NanoBRET (Bioluminescent Resonance Energy Transfer) technology. BRET is a proximity assay between a fluorescent donor (in this case full length LC3A and LC3B) and a

fluorescent acceptor (the dye adducts of **3** and **4**). BRET occurs when the donor and acceptor are in proximity (<10 nm) and its fluorescence intensity is inversely proportional to the distance of the donor-acceptor pair. This technology has gained popularity as a live cell assay format for monitoring protein-protein interactions or target engagement with small molecules. As expected from our biochemical data, we observed no BRET signal using the 8F20 and 1O05 dye analogs (compounds **16** and **17**) (Supplementary Fig. 8a–c).

To further study the cellular effects of the 4 ATTEC ligands (**1**–**4**), we monitored cellular growth in a live cell imaging system. Using RPE1 and U2OS cells, cell growth was monitored in live cells over a time course of 72 h after treatment with the respective compounds. Apart from **3**, no compound caused growth inhibition at concentrations <10 μM, while compound **3** strongly suppressed cellular growth (Supplementary Fig. 8d, e) without affecting cell viability at concentrations up to 30 μM (Supplementary Fig. 8f). Since we identified KIF11 as the only proteome-wide high affinity target and given the established role of this kinesin in cell division, KIF11 inhibition by **3** might trap cells in mitosis preventing progression of cell division[31]. Therefore, we analyzed the images taken during the live cell growth assay and found an increased number of rounded cells, indicating cells in a mitotic defect, consistent with the G2/M arrest as a result of 8F20 (**3**) treatment and in agreement with the literature (Supplementary Fig. 9)[31].

### In silico identification and biophysical validation of LC3/GABARAP ligands

Intrigued by the proposed mechanism of action of ATTECs and the possible advantages over PROTACs (e.g. no complex ubiquitin transfer mechanism or higher hurdle for cells to develop ATTEC resistance), we carried out screens for the identification of LC3/GABARAP ligands, using two independent approaches. In the first approach, we initiated a virtual screening campaign by using a library of >7500 diverse in-house compounds. All the docking poses of the in silico hits were individually inspected and we collected 271 compounds for experimental validation using the developed FP assay and all LC3/GABARAPs. Experimentally confirmed hits were investigated by similarity searches, which finally led to two LC3/GABARAP ligands (Fig. 4a, b). Interestingly, both ligands contained two carboxylic acid moieties and initial SAR insights using 26 ligands of this compound class present in our collection (Supplementary Tables 1–3) revealed the importance of both carboxylic acid moieties for binding. The first hit, LY223982 (**20**), was designed targeting the leukotriene B4 receptor[32] and showed selective binding to LC3 family members (Fig. 4b, c). Interestingly, TH152[33] (**21**) displayed a $K_D$ of 2 μM for LC3A in ITC titrations and interacted with all LC3/GABARAPs in FP assays (Fig. 4b, c). Thus, TH152 represents the most potent reversible pan-LC3/GABARAP ligand reported to date.

To validate the pan-Atg8 binding activity, we characterized the interaction of TH152 with $^{15}N$ labelled LC3B and GABARAP proteins by NMR titrations (Fig. 4d–f for LC3B with full NMR data analysis shown in Supplementary Fig. 10). The NMR results revealed that LC3B and GABARAP interacted with TH152 via the LDS binding site, confirming molecular docking studies. However, due to the presence of two carboxylic acid groups the identified ligands had poor cell penetration and will therefore require optimization for ATTEC development. Since our crystallization attempts to determine a TH152:LC3B crystal structure failed, we combined our docking pose with our NMR data. Mapping of chemical shift perturbation induced by binding of TH152 on the structure of LC3B confirmed our docking model (Fig. 4f). Residues showing strongest chemical shifts after TH152 binding (marked in blue) coincided with the docking pose. Both carboxylic acid groups showed similar binding modes, interacting with arginine 70 and lysine 51, respectively (LC3B). A detailed representation of the docking pose and the NMR interaction data is shown in Supplementary Fig. 11.

### Fragment screening on LC3 using X-ray crystallography

As a second hit finding approach, we conducted a large-scale fragment screening campaign using X-ray crystallography by soaking a total of 1006 LC3B crystals with a diverse fragment library. This led to the collection of over 800 high quality diffraction datasets, which identified a total of 21 diverse hits in the binding cavities on LC3B after refinement of the structures (Fig. 5a). This set significantly complements earlier fragment and hit finding campaigns using NMR and DEL (DNA encoded library) screening[22]. Our screen confirmed that HP2 is the most ligandable binding site on LC3/GABARAP protein's surface, accommodating 10 from 21 identified fragments (Fig. 5b). Fragments such as x0145 (HP1) and x0626 (S2) offer the possibility for fragment linking. In contrast, the previously reported data[22] showed only HP2 bound ligands (Fig. 5c). The UDS-targeting fragment (x0100) allows additional targeting opportunities. Comparison with hit rates of similar protein interaction domains such as E3 ligases suggest that the HP2 pocket can accommodate a diversity of ligands, indicating a good ligandability of this site. Due to the small size of the molecules, the fragment's affinity is most likely in the high μM to mM range as already observed by Steffek and colleagues and therefore requires significant synthetic chemistry efforts to increase the affinity into a measureable and useful affinity range[22]. However, based on our success with the very limited in silico study and the fragment screening campaign, we concluded that design and development of potent LC3/GABARAP ligands for ATTECs should be feasible. Making this rich pool of hit matter available together with the established assay validation platform will allow robust validation of LC3/GABARAP ligands which may be developed to either selectively target one human Atg8 family members or to develop pan-Atg8 ligands.

## Discussion

In this study we investigated published LC3/GABARAP ligands that have been used for the development of ATTECs as well as a set of covalent and reversible LC3 ligands. Surprisingly none of the ligands used for the development of ATTECs showed measurable affinity for the proposed targets LC3/GABARAP using a diversity of biophysical methods, suggesting that reported LC3-ligand based degraders cause degradation not due to a LC3/GABARAP mediated mechanism. However, some of the sources of recombinant protein have not been described in detail, possibly leading to differences in assay performance[24,34]. The hypothesis to use LC3/GABARAP as receptors for degrader development is quite new. As a result, few tools are available that can be utilized to demonstrate that the observed degradation events indeed are associated with autophagy. In the ubiquitinbased degrader field, pathway association of PROTACs is usually accomplished using a) inactive E3 ligands such as the inactive stereoisomer of VHL or N-methylated thalidomide derivatives[35]. These tools account also for possible effect caused by POI ligand that is often a pharmacological highly active small molecule; b) proteasomal inhibitors that rescue proteasome dependent degradation events; c) inhibitors of E3 ligase activating enzymes such as neddylation inhibitors for cullin dependent E3 ligases[36] and finally a proteome wide analysis demonstrating degrader selectivity. However, for the characterization of ATTECs, autophagy pathway activators (such as mTOR inhibitors) or inhibitors (such as Bafilomycin) could be used. We strongly believe that for new ligands and new degrader mechanisms stringent community guidelines for chemical probe developments should be applied. These quality standards would comprise direct on-target engagement assays and for stronger ligands also for cell based assay systems, appropriate controls comprising inactive control molecules, drug resistant mutants and/or knock out cell lines[37–40]. Recently also first standards for covalent inhibitors and degrader molecules have been defined[41].

Our ligandability analysis, in silico and fragment screening revealed that LC3/GABARAP are ligandable and in our study initial

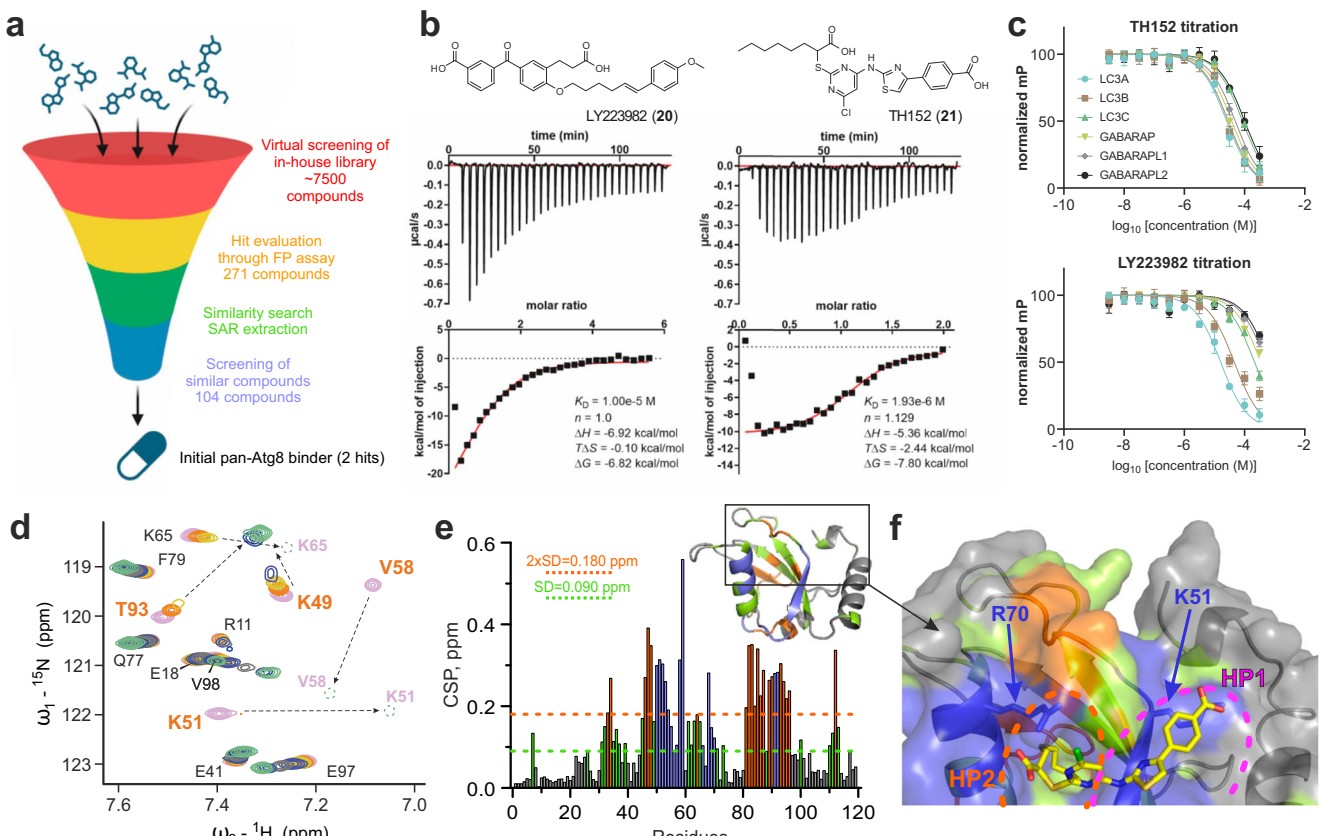

**Fig. 4 | LC3/GABARAP hit identification campaigns via virtual screening.**
**a** Schematic workflow of the virtual screening approach which was combined with biophysical hit validation. Our chemically diverse in-house library ( > 7500 compounds) was screened virtually using AutoDock[49] and SeeSAR (BioSolveIT). 271 virtual screening hits with the best docking scores were validated against all LC3/GABARAPs isoforms using our FP assay based on a p62 LIR tracer. Validated hits were used for similarity search within the in-house library was carried out using InfiniSee (BioSolveIT) and 104 similar compounds were screened again in vitro using FP assay resulting in two hits with affinity ≤ 10 µM affinity towards LC3A. Created in BioRender. Schwalm, M. (2023) BioRender.com/d80e756. **b** Structure of the two hits and corresponding ITC data (compounds **20** and **21**) measured against LC3A. **c** FP displacement assay curves using compounds **21** (upper panel) and **20** (lower panel) against a p62 LIR-based tracer for selectivity screening within the human Atg8 family proteins. Data were measured as technical triplicates with data presented as mean values +/− SD of each data point (n = 3). **d** Interaction between LC3B and compound **21** investigated by NMR. Representative areas of LC3B [$^{15}$N,$^{1}$H] BEST-TROSY spectra (recorded at 950 MHz spectrometer) around the K51 and V58 backbone HN resonances are shown in overlay with free LC3B (magenta), and in

stepwise increase of **21** molar ratios up to 1:4 (1:0.125−orange, 1:0.25−yellow, 1:0.5−light blue, 1:1−gray, 1:2−blue and 1:4−light green). Arrows show directions of large CSP for the resonances which are in the intermediate exchange mode. **e** CSP values, induced by compounds **21** at molar ratio 1:2, are plotted against LC3B residue numbers. The light green dashed line indicates the standard deviations (SD) over all residues, the orange dashed line indicates double SD values. The blue bars are for LC3B residues which undergo strong intermediate exchange mode (significant decrease of the resonances intensity upon titration with **21**). **f** Docking results from TH152 into the structure of LC3B (PDB ID 1UGM). The docked structure was subsequently color coded based on 3D mapping of the CSP values. Residues with small (CSP < SD), intermediate (SD < CSP <2xSD) or strong (2xSD <CSP) CSP values are marked in grey, light green and orange, respectively. LC3B residues which undergo strong intermediate exchange mode are marked blue, key residues K51 and R70 are indicated in stick representation. Relative positions of hydrophobic pockets HP1 (magenta) and HP2 (orange) are shown by dashed lines. More details on this NMR titration and NMR titration of **21** to the GABARAP protein are depicted in Supplementary Fig. 10. More details on the docking experiment provided in Supplementary Figs. 11, 12. Source data for (**b**, **c**, **e**) are provided as a Source Data file.

ligands with low µM potencies were reported. Interestingly, both ligands identified in our in silico study showed different selectivity profiles. While LY223982 showed selectivity towards LC3 proteins with a selectivity pattern comparable to Novobiocin, TH152 was found to bind all Atg8 family proteins with comparable affinity. Therefore, both compounds harbor the potential for further optimization to tool compounds once suitable isosteres for the required carboxylic acids moieties have been identified. A possible solution would be converting TH152 into a prodrug such as an ester which would neutralize its charge, increasing cell penetration.

Our comprehensive fragment screening study revealed several ligands that bind to the HP2 site, but also initial ligands for the HP1 and UIM sites were identified. It is however surprising that the larger HP1 site, that harbors a large aromatic amino acid side chain when liganded with LIR motifs, was not occupied by more ligands such as indoles that were present in our screening set. Interestingly, our

findings agreed with recently published fragment hits which also contained a free acid, highlighting the importance of this functional group for ligand development[22].

We hypothesize that fragment growing and linking efforts will result in more potent LC3/GABARAP ligands that could be used for the development of efficient ATTECs in the future. Loos et al. reported a study in which the authors tethered cargo to LC3 which resulted in no activation of autophagy[42]. Identification of LDS binders as autophagy handles might therefore be insufficient to trigger degradation.

Autophagosome biogenesis is a complex process involving many proteins and cellular factors. One of the key players in this process are the LC3/GABARAP proteins, which maintain a finely tuned equilibrium between their intact and lipidated forms in the cell. The perturbation of this balance by high-affinity LC3/GABARAP ligands can lead to a decrease in autophagy flux. Therefore, more research is required to gain a deeper understanding of the specific

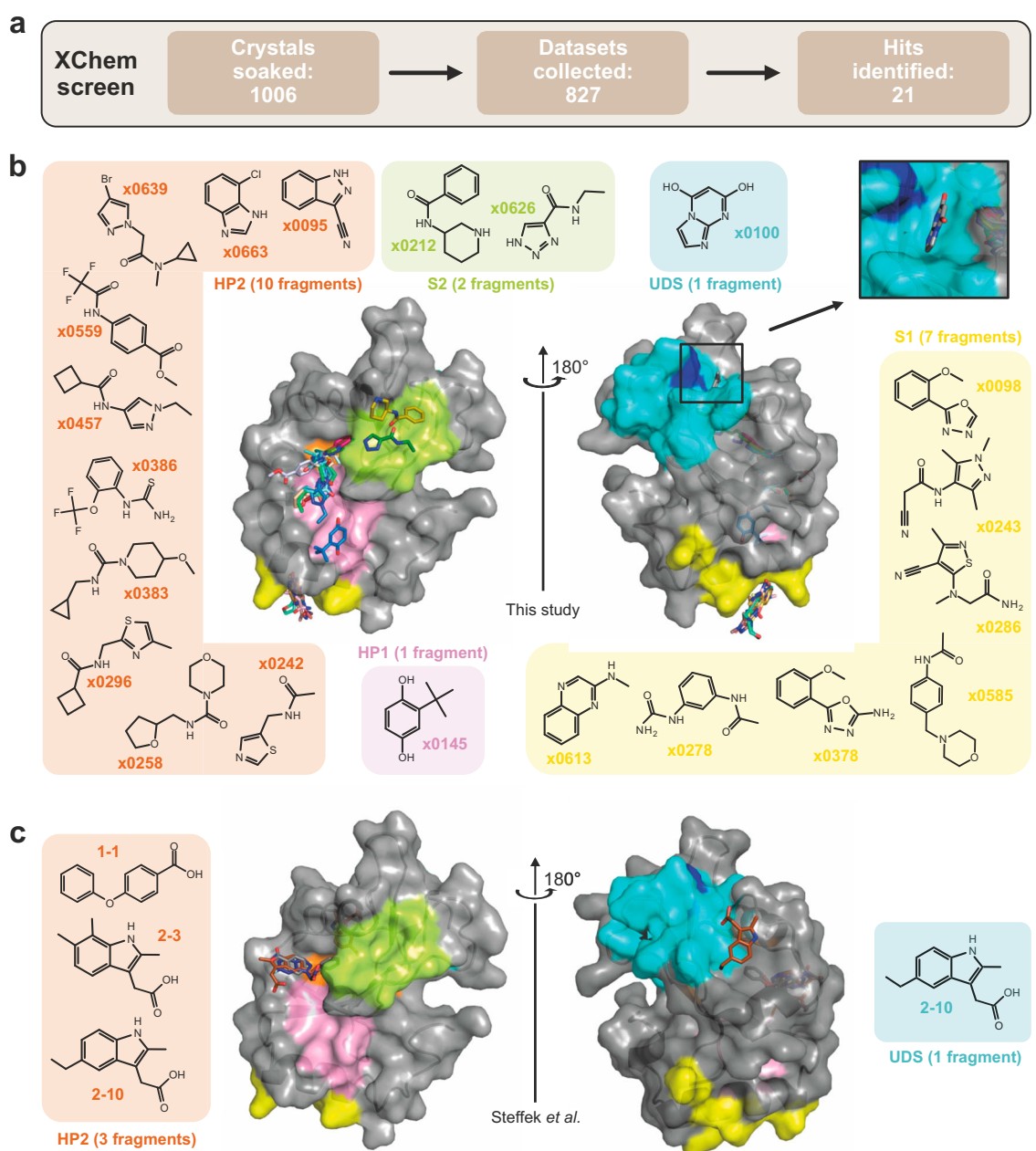

**Fig. 5 | LC3/GABARAP hit identification via X-ray crystallography fragment screening (XChem). a** Schematic workflow of the XChem screening, resulting in **21** identified hits. The hits are sorted by LC3B surface occupancy in the low part. **b** Overlay of crystal structures containing diverse fragments bound to LC3B. Bound fragments are depicted as chemical structures with arrows pointing to the binding sites: HP1 (pink), HP2 orange), UDS (cyan, key UDS residue F80 is shown blue), identified regions S1 (yellow) and S2 (light green). The insert shows the correct pose for x0100 within UDS (rotations by y45° and x20° degrees from the main plot). All structures available at protein data bank (PDB IDs 7GA8-7GA9 and 7GAA-7GAS). Exemplary electron density maps for representative binders are shown in Supplementary Fig. 13, data collection and refinement statistics are presented in the Supplementary Table 4. **c** Overlay of the three published crystal structures of LC3A containing small molecule fragments (PDB IDs 7R9W, 7R9Z and 7RA0) with bound fragments depicted by chemical structures[22].

conformational differences and dynamics of these proteins in the presence of such ligands in the cell. Consequently, in order to validate the ATTEC approach as a method of targeted protein degradation in TDP, it is necessary to undertake further experimental work.

Finally, we are certain that the design and implementation of functional ATTECs should be initiated with validated LC3/GABARAP binding compounds, which should have sufficient affinity to all or a specific protein of this subfamily. We would like to stress that the compounds and fragments we have identified in this study need further optimization to be used in cellular systems and for ATTEC

development. Thus, they should be used as chemical starting points for further development of more potent ligands.

## Methods

### Safety statement

No unexpected or unusually high safety hazards were encountered

### Chemistry

#### i. Synthesis of 1O5-based compounds

***Tert*-butyl 2-(2,6-dibromo-4-formylphenoxy)acetate.** 3,5-dibromo-4-hydroxybenzaldehyde (5.6 g, 20 mmol) was solved in anh. DMF

(60 mL). $K_2CO_3$ (5.52 g, 40 mmol) was added, and the mixture was stirred for 5 min. *Tert*-butyl 2-bromoacetate (4,68 g, 24 mmol) was stirred at ambient temperature overnight until complete consumption of starting material. The mixture was partitioned between water and ethyl acetate. The ethyl acetate layer was washed with brine, dried over $MgSO_4$, filtered, and volatiles were removed under reduced pressure yielding a pale-yellow oil which crystallized overnight. The product was used without further purification (92%) MS (ESI): m/z calc. for $[C_{13}H_{14}Br_2O_4 + Na^+]^+ = 417.06$, found = 416.85 $^1H$ NMR (500 MHz, DMSO-$d_6$) δ 9.90 (s, 1H), 8.17 (s, 2H), 4.64 (s, 2H), 1.46 (s, 9H). $^{13}C$ NMR (126 MHz, DMSO-$d_6$) δ 190.04, 190.02, 166.06, 156.20, 134.57, 133.73, 118.19, 81.86, 69.42, 27.67.

**Tert-butyl 2-(2,6-dibromo-4-((5-iodo-2-oxoindolin-3-ylidene)methyl)phenoxy)acetate.** A mixture of *tert*-butyl 2-(2,6-dibromo-4-formylphenoxy)acetate (0.606 g, 1.5 mmol), 5-iodoindolin-2-one (0.518 g, 2 mmol) were suspended in absolute ethanol (8 mL). Catalytic amounts of piperidine (0.1 eq, 0.013 g, 0,15 mmol or 15 μL) were added and the mixture was refluxed at 80 °C for 3 h. After 3 h an orange solid formed. The solid was filtered through a glass frit and rinsed with cold ethanol. The solid was collected and used in the next step without purification yielding an inseparable mixture of *E/Z* isomers. The mother liquor was evaporated *in vacuo* and purified by flash chromatography (*n*-hexanes/EtOAc, 6:1) to increase the overall yield (80%). MS (ESI): m/z calc. for $[C_{13}H_{14}Br_2O_4 + Na^+]^+ = 658,09$, found = 657.80. $^1H$ NMR (400 MHz, DMSO-$d6$) δ 10.77 (s, 1H), 8.78 (s, 1H), 8.01 (d, J = 0.7 Hz, 2H), 7.72 (d, J = 1.7 Hz, 1H), 7.60 − 7.52 (m, 3H), 6.73 (d, J = 8.2 Hz, 1H), 4.62 (d, J = 0.7 Hz, 2H), 4.59 (d, J = 0.7 Hz, 1H), 1.47 (dd, J = 2.2, 0.7 Hz, 13H). $^{13}C$ NMR (101 MHz, DMSO-$d6$) δ 166.49, 166.23, 152.93, 140.54, 138.65, 137.53, 136.19, 134.44, 133.58, 133.53, 133.38, 133.08, 130.65, 128.44, 128.03, 127.09, 126.96, 117.51, 116.84, 112.71, 111.98, 84.20, 81.78, 69.50, 69.48, 27.74, 27.72.

**2-(2,6-dibromo-4-((5-iodo-2-oxoindolin-3-ylidene)methyl)phenoxy)acetic acid (compound 14).** *Tert*-butyl 2-(2,6-dibromo-4-formylphenoxy)acetate (0.300 mg, 0.47 mmol) was solved in absolute DCM (10 mL). TFA (2 mL) was added dropwise to the solution and let stir at ambient temperature for 2 h until complete consumption of starting material. After 2 h a red solid formed which was transferred into a glass frit and rinsed witch cold DCM. The crystals were collected and dried *in vacuo* overnight (95%). $^1H$ NMR (400 MHz, DMSO-$d6$) δ 13.17 (s, 1H), 10.83 (s, 1H), 8.78 (s, 2H), 8.06−7.99 (m, 2H), 7.85 (s, 1H), 7.71 (d, J = 1.7 Hz, 1H), 7.61−7.51 (m, 2H), 6.71 (dd, J = 20.1, 8.1 Hz, 2H), 4.62 (s, 1H), 4.60 (s, 2H).

$^{13}C$ NMR (101 MHz, DMSO-$d6$) δ 168.56, 167.58, 166.49, 152.85, 142.79, 140.54, 138.64, 137.54, 136.18, 134.45, 133.60, 133.52, 133.14, 130.70, 128.45, 128.08, 127.09, 126.97, 123.04, 117.66, 116.96, 112.70, 111.98, 84.21, 83.87, 68.86, 68.83.

**Tert-butyl (1-(2,6-dibromo-4-((5-iodo-2-oxoindolin-3-ylidene)methyl)phenoxy)−2-oxo-7,10,13-trioxa-3-azahexadecan-16-yl)carbamate (compound 15).** A mixture of 2-(2,6-dibromo-4-((5-iodo-2-oxoindolin-3-ylidene)methyl)phenoxy)acetic acid (0.316 g, 0.55 mmol), *tert*-butyl (3-(2-(2-(3-aminopropoxy)ethoxy)ethoxy)propyl)carbamate (0.192 g, 0.6 mmol) were solved in anhy. DMF (18 mL). PyAOP (0.342 g, 0.65 mmol) and DIPEA (0,091 g, 0,71 mmol or 125 μL) were added to the mixture and let stir for 1.5 h at ambient temperature. The crude mixture was evaporated *in vacuo* and purified directly via reverse phase column chromatography ($H_2O$/ACN) (79%). MS (ESI): m/z calc. for $[C_{13}H_{14}Br_2O_4 + Na^+]^+ = 904,4$ found = 904.00 $^1H$ NMR (500 MHz, Methylene Chloride-$d2$) δ 9.27 (s, 1H), 8.51 (s, 1H), 7.85−7.73 (m, 2H), 7.55−7.47 (m, 1H), 7.39−7.31 (m, 1H), 7.25 (s, 1H), 6.69 (dd, J = 25.4, 8.2 Hz, 1H), 5.17−4.97 (m, 1H), 4.56 (s, 1H), 4.52 (s, 1H), 3.60 (d, J = 5.9 Hz, 6H), 3.57−3.43 (m, 9H), 3.16 (q, J = 6.4 Hz, 2H),

1.88 (td, J = 6.3, 3.3 Hz, 2H), 1.70 (p, J = 6.2 Hz, 2H). $^{13}C$ NMR (126 MHz, Methylene Chloride-$d2$) δ 167.54, 167.47, 156.48, 153.39 (d, J = 39.1 Hz), 140.71, 139.54, 138.62, 136.74, 134.33, 134.14, 134.01, 132.22, 128.97, 118.70, 117.82, 113.11, 112.51, 79.13, 71.81, 71.73, 71.04, 71.02, 71.01, 70.99, 70.96, 70.71, 70.67, 70.07, 70.01, 69.93, 69.89, 54.43, 54.22, 54.00, 53.78, 53.57, 39.04, 37.77, 37.74, 30.31, 29.82, 29.79, 28.74.

**N-(1-(2,6-dibromo-4-((5-iodo-2-oxoindolin-3-ylidene)methyl)phenoxy)−2-oxo-7,10,13-trioxa-3-azahexadecan-16-yl)−3-(5,5-difluoro-7-(1H-pyrrol-2-yl)−5H-4l4,5l4-dipyrrolo[1,2-c:2',1'-f][1,3,2]diazaborinin-3-yl)propenamide (compound 16).** *Tert*-butyl (1-(2,6-dibromo-4-((5-iodo-2-oxoindolin-3-ylidene)methyl)phenoxy)−2-oxo-7,10,13-trioxa-3-azahexadecan-16-yl)carbamate (0.025 g, 0.029 mmol) was charged into a flask and solved in anhy. DCM (1 mL). TFA (0.7 mL) was added and the solution for stirred for 1 h until complete consumption of starting material. Toluene (2 mL) was added and the solution was evaporated *in vacuo* and used directly in the next step without purification. The crude was solved in anhy. DMF (1 mL) and the flask was wrapped in tin foil. Py-BODIPY-NHS ester (0.011 g, 0.026 mmol) and DIPEA (0.06 mM, 11 μL) were added to the solution and stirred for 2 h at ambient temperature. The crude was afterwards purified by prep. HPLC ($H_2O$/ACN with 0.1% TFA) to provide the title compound (80%). MS (HRMS): m/z calc. for $[C_{43}H_{44}BBr_2F_2IN_6O_7 + Na^+]^+ = 1113,0744$, found = 1113,06312. $^1H$ NMR (400 MHz, DMSO-$d6$) δ 11.40 (s, 1H), 10.84 (s, 1H), 8.79 (s, 2H), 8.17 (dt, J = 11.8, 5.8 Hz, 2H), 8.03 (dd, J = 4.9, 1.2 Hz, 2H), 7.90 (t, J = 5.6 Hz, 1H), 7.86 (s, 1H), 7.57 (td, J = 8.2, 1.7 Hz, 2H), 7.43 (s, 1H), 7.38 − 7.32 (m, 3H), 7.27 (td, J = 2.7, 1.3 Hz, 1H), 7.16 (d, J = 4.5 Hz, 1H), 7.01 (d, J = 4.0 Hz, 1H), 6.69 (d, J = 8.1 Hz, 1H), 6.37−6.29 (m, 3H), 4.45 (s, 1H), 4.43 (s, 2H), 3.55−3.43 (m, 16H), 3.25 (q, J = 6.7, 5.0 Hz, 5H), 3.11 (dd, J = 12.9, 6.9 Hz, 6H), 1.73 (td, J = 6.7, 3.2 Hz, 3H), 1.64 (q, J = 6.6 Hz, 3H), 1.25 (d, J = 6.8 Hz, 4H). $^{13}C$ NMR (101 MHz, DMSO-$d6$) δ 171.29, 166.97, 166.64, 156.43, 153.24, 150.69, 141.04, 139.12, 138.06, 137.41, 136.65, 134.87, 133.98, 133.76, 133.48, 132.87, 128.94, 127.55, 127.22, 126.59, 124.86, 123.37, 119.82, 117.90, 117.53, 116.60, 112.48, 111.99, 84.70, 71.46, 70.25, 70.10, 70.00, 68.86, 68.55, 36.56, 36.34, 34.39, 29.81, 29.64, 29.49, 29.18, 24.54, 22.56.

**N-(3-(2-(2-(3-aminopropoxy)ethoxy)ethoxy)propyl)−2-(2,6-dibromo-4-((5-iodo-2-oxoindolin-3-ylidene)methyl)phenoxy)acetamide (TFA salt) (compound 18).** *Tert*-butyl (1-(2,6-iodo-2-oxoindolin-3-ylidene)methyl)phenoxy)−2-oxo-7,10,13-trioxa-3-azahexadecan-16-yl)carbamate (0.021 g, 0.024 mmol) was charged into a flask and solved in anhy. DCM (1 mL). TFA (0.6 mL) was added and the solution for stirred for 1 h until complete consumption of starting material. Toluene (2 mL) was added and the solution was evaporated and dried *in vacuo* overnight yielding the title compound as TFA salt.

MS (ESI): m/z calc. for $[C_{43}H_{44}BBr_2F_2IN_6O_7 + H^+]^+ = 782.28$ found = 782.05

## ii. Synthesis of 8F2O (Ispinesib)-based compounds

**(R)-N-(1-amino-12-oxo-3,6,9-trioxa-13-azahexadecan-16-yl)-N-(1-(3-benzyl-7-chloro-4-oxo-3,4-dihydroquinazolin-2-yl)−2-methylpropyl)−4-methylbenzamide (compound 19).** A mixture of (R)-N-(3-aminopropyl)-N-(1-(3-benzyl-7-chloro-4-oxo-3,4-dihydroquinazolin-2-yl)−2-methylpropyl)−4-methylbenzamide (Ispinesib (bought from MedChemExpress), 80 mg, 150 μmol), (7-Azabenzotriazol-1-yloxy)tripyrrolidinophosphonium hexafluorophosphate (123 mg, 216 μmol), 2,2-dimethyl-4-oxo-3,8,11,14-tetraoxa-5-azaheptadecan-17-oic acid (52 mg, 162 μmol) and *N,N*-diisopropylethylamine (40 μL, 232 μmol) in anh. DMF (5 mL) was stirred at ambient temperature. After 1 h, the mixture was partitioned between water and ethyl acetate. The ethyl

acetate layer was washed with brine, dried over MgSO$_4$, filtered, and volatiles were removed under reduced pressure. The residue was dissolved in DCM/TFA (3/1, 8 mL) and stirred for 1 h. After 1 h, all volatiles were removed under reduced pressure to provide the title compound (100 mg, 90%) which was used in the next step without further purification. MS (ESI): m/z calc. for [M + H$^+$]$^+$ = 720.34, found = 720.30.

**(R)-N-(1-(3-benzyl-7-chloro-4-oxo-3,4-dihydroquinazolin-2-yl)−2-methylpropyl)-N-(1-(5,5-difluoro-7-(1H-pyrrol-2-yl)−5H−5l4,6l4-dipyrrolo[1,2-c:2′,1′-f][1,3,2]diazaborinin-3-yl)−3,16-dioxo-7,10,13-trioxa-4,17-diazaicosan-20-yl)−4-methylbenzamide (compound 17).**
A mixture of (R)-N-(1-amino-12-oxo-3,6,9-trioxa-13-azahexadecan-16-yl)-N-(1-(3-benzyl-7-chloro-4-oxo-3,4-dihydroquinazolin-2-yl)−2-methylpropyl)−4-methylbenzamide (compound **19**, 15 mg, 21 μM), 2,5-dioxopyrrolidin-1-yl 3-(5,5-difluoro-7-(1H-pyrrol-2-yl)−5H−5λ$^4$,6λ$^4$-dipyrrolo[1,2-c:2′,1′-f][1,3,2]diazaborinin-3-yl)propanoate (8.5 mg, 20 μM) and N,N-diisopropylethylamine (8.6 μL, 50 μM) in anh. DMF (0.3 mL) was stirred at ambient temperature for 2 h and afterwards purified by prep. HPLC (H$_2$O/ACN with 0.1% TFA) to provide the title compound (18 mg, 88%). $^1$H NMR (500 MHz, DMSO-$d_6$): δ 11.50 (s, 1H), 8.23 (d, J = 8.6 Hz, 1H), 7.81 (d, J = 2.1 Hz, 1H), 7.79 (s, 1H), 7.66 (dd, J = 8.6, 2.1 Hz, 1H), 7.46 (s, 1H), 7.44−7.39 (m, 2H), 7.38 (d, J = 4.6 Hz, 2H), 7.36 (d, J = 7.6 Hz, 2H), 7.34−7.27 (m, 2H), 7.28−7.19 (m, 7H), 7.01 (d, J = 4.0 Hz, 1H), 6.45 (d, J = 3.9 Hz, 1H), 6.37 (dt, J = 4.2, 2.2 Hz, 1H), 5.88 (d, J = 16.2 Hz, 1H), 5.54 (d, J = 10.5 Hz, 1H), 5.05 (d, J = 16.3 Hz, 1H), 3.57 (t, J = 5.4 Hz, 2H), 3.56−3.50 (m, 3H), 3.49−3.39 (m, 5H), 3.27 (q, J = 10.5, 8.9 Hz, 5H), 3.15 (t, J = 7.5 Hz, 3H), 2.96 (q, J = 5.5 Hz, 2H), 2.73 (dq, J = 10.8, 6.4 Hz, 1H), 2.33 (s, 3H), 2.03 (q, J = 6.8 Hz, 2H), 1.35−1.14 (m, 2H), 0.89 (d, J = 6.7 Hz, 3H), 0.87−0.78 (m, 1H), 0.47 (d, J = 6.3 Hz, 3H). $^{13}$C NMR (126 MHz, DMSO): δ 171.99, 170.21, 169.48, 168.30, 161.13, 155.24, 152.40, 150.96, 147.21, 139.52, 138.71, 137.44, 136.69, 133.77, 133.08, 133.01, 128.91, 128.71, 128.67, 128.04, 127.45, 126.71, 126.42, 126.13, 125.90, 124.41, 122.78, 119.96, 119.10, 118.07, 118.02, 117.96, 116.07, 111.80, 69.67, 69.64, 69.60, 69.44, 66.67, 66.64, 58.99, 45.18, 42.32, 35.83, 35.77, 29.33, 25.47, 22.99, 20.89, 19.48, 18.16. HRMS (MALDI): m/z calc. for [M+Na$^+$]$^+$ = 1053.4383, found = 1053.4377

**Protein expression and purification for biophysical assays:** LC3A$_{1-120}$, LC3B$_{1-120}$, LC3C$_{1-126}$, GABARAP$_{1-116}$, GABARAPL1$_{1-117}$ and GABARAPL2$_{1-117}$ were cloned into the pNIC28-Bsa4 vector using restriction sites Lic5 and Lic3 and expressed as a recombinant fusion protein incorporating a His6 and TEV cleavage site at the N-terminus. E. coli Rosetta cells were cultured in Terrific Broth (TB) at 37 °C until an OD600 of 1.0 was reached. The culture was then cooled to 18 °C and allowed to reach an OD600 of 2.5. Protein expression was induced by the addition of 0.5 mM isopropyl β-D-1-thiogalactopyranoside (IPTG) and the protein was allowed to express overnight. Cells were harvested (Beckman centrifuge, via centrifugation at 6000 g at 4 °C) and lysed by sonication (SONICS vibra cell, 5 s on-, 10 s off cycle using a total of 30 min) in the presence of DNase I (Roche, Basel, CH) and cOmplete EDTA-free protease inhibitor (Roche, Basel, CH), and recombinant protein was purified using Ni-NTA-affinity chromatography in Purification buffer (30 mM 4-(2-hydroxyethyl)−1-piperazineethanesulfonic acid; pH 7.5 (HEPES), 500 mM NaCl, 5 % glycerol, 0.5 mM tris(2-carboxyethyl) phosphine (TCEP) and 30 mM Imidazole) and elution was carried out using Purification buffer including additional 300 mM Imidazole. The eluted proteins were dialyzed overnight into gel filtration buffer (30 mM HEPES pH 7.5, 250 mM NaCl, 5 % glycerol and 0.5 mM TCEP) while the expression tag was cleaved using 1 mg tobacco etch virus (TEV) protease. The cleaved protein was passed through a HiLoad® 26/600 Superdex® 75 pg (GE Healthcare) size exclusion chromatography column and the resulting pure protein was stored in gel filtration buffer, flash frozen in liquid nitrogen and subsequently stored at −80 °C for further experiments.

**Protein purification for X-ray crystallography:** Human LC3B$_{1-120}$ construct cloned into the pNIC28-Bsa4 vector was transformed into E. coli Rosetta(DE3) competent cells and expressed in TB medium by overnight induction with 0.2 mM IPTG (OD600 = 2.5). Cells were harvested by centrifugation, resuspended in buffer A (30 mM HEPES pH 7.5 @ 4 °C, 500 mM NaCl, 0.5 mM TCEP, 10 mM Imidazole, and 5% Glycerol), and lysed by sonication on ice. The soluble fraction was collected by centrifugation at 21,000 g for 40 min. The fraction was with 4 ml Ni-NTA beads (pre-equilibrated with lysis buffer) for batch binding on ice for 1 h. The beads were washed with buffer B (30 mM HEPES pH 7.5 @ 4 °C, 500 mM NaCl, 0.5 mM TCEP, 30 mM Imidazole, and 5% Glycerol) and eluted with buffer C (30 mM HEPES pH 7.5 @ 4 °C, 500 mM NaCl, 0.5 mM TCEP, 300 mM Imidazole, and 5% Glycerol). Protein in the eluted fraction was treated with TEV protease overnight while dialyzing against (30 mM HEPES pH 7.5 @ 4 °C, 300 mM NaCl, 0.5 mM TCEP, and 5% Glycerol) to cleave the His-tag. The dialyzed mixture was passed through 4 ml Ni-NTA beads, flow-through was collected, concentrated, and injected into GE Superdex 75 16/600 Prep grade column pre-equilibrated with SEC buffer (30 mM HEPES pH 7.5 @ 4 °C, 100 mM NaCl, 0.5 mM TCEP, and 5% Glycerol). The peak was collected and concentrated to 22.5 mg/ml.

**Isothermal Titration Calorimetry (ITC):** ITC experiments were performed using a NanoITC instrument (TA Instruments, New Castle, USA) at 25 °C in gel filtration buffer (50 mM Na$_2$HPO$_4$ pH = 7.0, 100 mM NaCl and 0.5 mM TCEP). To minimize nonspecific dilution heat effects, DMSO concentrations in the protein and compound samples were matched. 500 μM inhibitors dissolved in gel filtration buffer (in syringe) was titrated into purified LC3/GABARAPs at a concentration of 25 μM (in the reaction cell). For this protocol, the chamber was pre-equilibrated with the protein, and the test compounds were titrated while continuously measuring the rate of exothermic heat evolution. The heat of binding was integrated, corrected, and fitted to an independent single-binding site model based on the manufacturer's instructions, from which thermodynamic parameters (ΔH and TΔS), equilibrium association and dissociation constants (K$_A$ and K$_D$, respectively), and stoichiometry (n) were calculated using TA Instruments NanoAnalyze software. Data were displayed using GraphPad Prism 9.3.

**Temperature shift assay (TSA):** Purified proteins were buffered in TSA buffer (25 mM HEPES pH 7.5, 500 mM NaCl) and were assayed in a 384-well plate (Thermo, #BC3384) with a final protein concentration of 20 μM in 10 μL final assay volume. Inhibitors were added in excess to a final concentration of 40 μM, using an ECHO 550 acoustic dispenser (Labcyte). As a fluorescent probe, SYPRO-Orange (Molecular Probes) was used at 5x final concentration. Filters for excitation and emission were set to 465 nm and 590 nm, respectively. The temperature was increased from 25 °C with 3 °C/min to a final temperature of 99 °C, while scanning, using the QuantStudio5 (Applied Biosystems). Data was analyzed using Boltzmann-equation in the Protein Thermal Shift software (Applied Biosystems). Samples were measured in technical triplicates.

**Affinity determination using spectral shift mode on Dianthus:** To determine dissociation constants, the Dianthus instrument (NanoTemper Technologies GmbH, Germany) was used to monitor molecular interactions. Spectral shift assays monitor changes in the emission spectrum of extrinsically fluorescently labelled LC3A by measuring the fluorescence intensity at two specific emission wavelengths and expressed as the Ratio 670 nm/650 nm. Protein was labelled with RED-maleimide 2nd Generation (cat# MO-L014; NanoTemper Technologies GmbH). Labeling was carried out following the manufacturer protocol, using a 3:1 ratio dye:protein. Labeled LC3A with degree of labeling of 0.7 (as determined by UV-VIS absorbance spectroscopy). Protein was purified in 30 mM HEPES, 100 mM NaCl, 0.05% Tween, pH 7.5. For determination of the K$_D$ with spectral shift assays, a 16-point affinity measurement was performed in 10 mM

Na$_2$HPO$_4$/KH$_2$PO$_4$, 0.05% Tween-20, 5 % DMSO, pH 7.4 with a maximum ligand concentration of 500 μM. Measurements were performed in spectral shift mode with an LED excitation power of 100%. Data were analyzed using the DI.Screening Analysis Software (v.2.0.4) (Nano-Temper Technologies GmbH, Germany) and quality criteria values including Δ Ratio, Signal-to-Noise-Ratio, Fit Saturation and K$_D$ values were determined.

**Fluorescence polarization assay (FP assay):** For the tracer displacement assays, the fluorescently labeled p62 LIR probe (SDNSSGGDDDWTHLSSK-Cy5) was diluted to (30 nM) in assay buffer (50 mM HEPES pH 7.5, 150 mM NaCl, 5 % glycerol, 1 mM TCEP and 0.05% TWEEN20) in a black 384-well flat bottom plate (Greiner Bio-One, #784076) and purified LC3/GABARAPs were titrated in a concentration range from 55 μM to 600 pM. After 1 h incubation at room temperature, fluorescence polarization was measured with polarized excitation wavelength of 590 nm and filtered emission wavelength of 675 nm, respectively, using a PHERAstar plate reader (BMG Labtech). Resulting data was plotted using GraphPad Prism 9.3 software and analyzed using a nonlinear fit to calculate the probe IC$_{50}$. For competition assays, 30 nM probe was added to assay buffer containing 4 μM LC3/GABARAPs. Compounds were titrated from 20 μM to 20 nM using an ECHO 550 acoustic dispenser (Labcyte) incubated for 1 h at room temperature and subsequent read out as described above. Data was plotted in GraphPad Prism 9.3 and analyzed using a nonlinear fit for IC$_{50}$ determination. K$_I$ calculation was performed using the Nikolovska-Coleska formula[43]. A detailed step-by-step protocol for this assay is described in Schwalm et al. [44].

**Covalent compound screening:** For screening of covalent modification, compounds (5–7 and 1) were tested against all LC3/GABARAPs. For LC-MS experiments, 50 μM of protein was used together with 100 μM of compound. The reaction was incubated for 90 min at room temperature and stopped by a 1:30 dilution in H$_2$O with 0.1 % formic acid. Samples were measured, using an Agilent 6230 TOF LC/MS. Data was evaluated using the BioConfirm B.08.00 software.

**NanoBRET cellular target engagement assay:** Constructs contained the cDNA of full-length LC3A and LC3B cloned in frame with an N-terminal NanoLuc-fusion. Plasmids were transfected into HEK293T (ATCC; CRL-11268) cells using FuGENE HD (Promega, E2312) and proteins were allowed to express for 20 h. 10O5 and 8F20-based tracers were titrated to the protein as depicted in Supplementary Fig. 4d. For competition experiments, 1 μM of the tracers was pipetted into white 384-well plates (Greiner 781 207) using an Echo 550 acoustic dispenser (Labcyte) containing LC3A/LC3B expressing transfected cells at a density of $2.5 \times 10^5$ cells/mL in Opti-MEM without phenol red (Life Technologies). The system was allowed to equilibrate for 2 h at 37 °C and 5% CO$_2$ prior to BRET measurements. To measure BRET, Nano-BRET NanoGlo Substrate + Extracellular NanoLuc Inhibitor (Promega, N2540) was added as per the manufacturer's protocol, and filtered luminescence was measured on a PHERAstar plate reader (BMG Labtech) equipped with a luminescence filter pair (450 nm BP filter (donor) and 610 nm LP filter (acceptor)). Competitive displacement data were then plotted using GraphPad Prism 9.3 software using a normalized 3-parameter curve fit. A detailed step-by-step protocol for this assay is described in Schwalm. [45].

**Preparation of Affinity Matrix 10O5 and 8F20:** Compounds 18 and 19 (1 μM) were linked to DMSO-washed NHS-activated ( ~ 20 μM/mL beads) sepharose beads (1 mL) and triethylamine (20 μL) in DMSO (2 mL) on an end-over-end shaker overnight at RT in the dark. Aminoethanol (50 μL) was then added to inactivate the remaining NHS-activated carboxylic acid groups. After 16 h the beads were washed with 10 mL DMSO and 30 mL EtOH to yield an affinity matrix of 10O5 and 8F20, respectively which were stored at 4 °C in EtOH. Successful immobilization was controlled by LC-MS and Kaiser-test[46].

**Preparation of cell lysates for affinity pulldown assays:** HEK293 cells (ATCC; CRL-11268) were cultured in DMEM (PAN Biotech). All media were supplemented with 10% FBS (PAN Biotech) and cells were internally tested for Mycoplasma contamination. Cells were lysed in lysis buffer (0.8% Igepal, 50 mM Tris-HCl pH 7.5, 5% glycerol, 1.5 mM MgCl$_2$, 150 mM NaCl, 1 mM Na$_3$VO$_4$, 25 mM NaF, 1 mM DTT and supplemented with protease inhibitors (SigmaFast, Sigma) and phosphatase inhibitors (prepared in-house according to Phosphatase inhibitor cocktail 1, 2 and 3 from Sigma-Aldrich)). The protein amount of cell lysates was determined by Bradford assay and adjusted to a concentration of 5 mg/mL[46].

**Competition pulldown assays:** For the selectivity profiling of free Compound 18 and 19, lysates from HEK293 cells were adjusted to 5 mg/mL protein concentration (0.4% Igepal). Then, 0.5 mL lysate was pre-incubated with 10 doses of the compounds (DMSO vehicle, 3 nM, 10 nM, 30 nM, 100 nM, 300 nM, 1000 nM, 3000 nM, 10000 nM, 30000 nM) for 1 h at 4 °C in an end-over-end shaker, followed by incubation with 18 μL of the affinity matrix 10O5 or 8F20 for 30 min at 4 °C in an end-over-end shaker[46]. The experiment was performed once at each concentration ($n = 1$).

The beads were washed (1 × 1 mL of lysis buffer without inhibitors and only 0.4% Igepal, 2 × 2 mL of lysis buffer without inhibitors and only 0.2% Igepal, and 3x lysis buffer without Igepal). The remaining proteins were denatured in in 40 μl of 8 M urea buffer (with 10 mM DTT in 40 mM Tris-HCl, pH 7.4), alkylated by adding 5 μl of 55 mM chloroacetamide and digested by adding 30 μl of 10 ng/μl Trypsin solution. Resulting peptides were desalted on a C18 filter plate (Sep-Pak® tC18 μElution Plate, Waters), vacuum dried and stored at −20 °C until LC-MS/MS measurement.

**LC-MSMS measurement of (competition) pulldown assays:** Peptides were analyzed via LC-MS/MS on a Dionex Ultimate3000 nano HPLC coupled to an Orbitrap Fusion Lumos mass spectrometer, operated via the Thermo Scientific Xcalibur software. Peptides were loaded on a trap column (100 μm × 2 cm, packed in house with Reprosil-Gold C18 ODS-3 5 μm resin, Dr. Maisch, Ammerbuch) and washed with 5 μL/min solvent A (0.1% formic acid in HPLC grade water) for 10 min. Peptides were then separated on an analytical column (75 μm × 40 cm, packed in house with Reprosil-Gold C18 3 μm resin, Dr. Maisch, Ammerbuch) using a 50 min gradient ranging from 4–32% solvent B (0.1% formic acid, 5% DMSO in acetonitrile) in solvent A (0.1% formic acid, 5% DMSO in HPLC grade water) at a flow rate of 300 nL/min.

The mass spectrometer was operated in data dependent mode, automatically switching between MS1 and MS2 spectra. MS1 spectra were acquired over a mass-to-charge (m/z) range of 360–1300 m/z at a resolution of 60,000 (at m/z 200) in the Orbitrap using a maximum injection time 50 ms and an automatic gain control (AGC) target value of 4e5. Up to 12 peptide precursors were isolated (isolation width of 1.2 Th, maximum injection time of 75 ms, AGC value of 2e5), fragmented by HCD using 25 % 30% normalized collision energy (NCE) and analyzed in the Orbitrap at a resolution of 15,000. The dynamic exclusion duration of fragmented precursor ions was set to 30 s[46].

**Competition pulldown assay protein identification and quantification:** Protein identification and quantification was performed using MaxQuant (v 1.6.1.0[47]. by searching the LC-MS/MS data against all canonical protein sequences as annotated in the Swissprot reference database (v03.12.15, 20193 entries, downloaded 22.03.2016) using the embedded search engine Andromeda. Carbamidomethylated cysteine was set as fixed modification and oxidation of methionine and N-terminal protein acetylation as variable modifications. Trypsin/P was specified as the proteolytic enzyme and up to two missed cleavage sites were allowed. Precursor tolerance was set to 10 ppm and fragment ion tolerance to 20 ppm. The minimum length of amino acids was set to seven and all data were adjusted to 1% PSM and 1% protein FDR. Label-free quantification[47] and match between runs was enabled[46].

**Competition pulldown assay data analysis:** Relative residual binding of proteins to the affinity matrix was calculated based on the protein intensity ratio relative to the DMSO control for every single inhibitor concentration. $EC_{50}$ values were derived from a four-parameter log-logistic regression using an internal R script that utilizes the 'drc' package in R. Targets of the inhibitors were annotated manually. A protein was considered a target or interactor of a target if the resulting binding curve showed a sigmoidal curve shape with a dose dependent decrease of binding to the beads. Additionally, the number of unique peptides and MSMS counts per condition were taken into account.

**Cell growth assay:** Cell confluence (phase) from RPE1 (hTERT RPE-1, ATCC CLR_4000) or U2OS (ATCC, HTB-96) cell lines were monitored over time with the IncuCyte S3 (Sartorius, Germany) in 384-well plates in 50 ul DMEM media, supplemented with 10% Fetal Bovine Serum (FBS) 1% penicillin/Streptomycin and incubated 24 h before treatment. Experiment was performed as described in Cano-Franco et al. [48]. 50 μl of media containing either 2× final concentration of indicated compounds in or control compounds (final conc.: 0.1% DMSO, 250 nM Torin1, 200 ng/mL Bafilomycin) were added and images were taken every two hours over 72 h. Cell confluence is represented as % of area covered by cells. Each data point represents the average ratio or confluence obtained from three individual wells of the plate.

**Virtual screening:** Our compound library of ~7500 compounds were virtually screened against LC3A and GABARAP (PDB IDs 6TBE, 4XC2) utilizing SeeSAR (BioSolveIT) and in-house software based upon AutoDock-GPU [49]. The binding site definitions used in SeeSAR was D23, Q24, I27, Q30, H31, K34, V37, I39, L51, K53, K55, F56, L57, V58, P59, V62, Q66, L67, I70, I71, R74, F112 for LC3A and D17, K20, I32, K24, Y25, R28, V31, V33, L44, K46, K48, Y49, L50, V51, P52, L55, Q59, F60, L63, I64, R67, F104 for GABARAP. The parameters were set to the generation of 3 poses per molecule, standard clash tolerance and the chair conformation as the allowed ring conformation. After structural alignment of GABARAP to LC3A, the following grid parameters were used in our virtual screening efforts with AutoDock-GPU: 0.375 spacing, (48, 42, 62) as numbers of grid points, (9.088, 19.070, −0.591) as center coordinates, defined in the (x, y, z) directions. Default settings were used as the docking parameters in AutoDock-GPU. Resulting poses were sorted by estimated affinity (SeeSAR) and free energy (Autodock) [49] results and filtered by molecular mass with a 750 Da cut-off. 271 compounds displaying best docking scores were subjected to in-vitro hit validation.

**Similarity search:** Validated hits were included in a Tanimoto-based similarity search via the SpaceLight chemical space exploration tool within the infiniSee suite (BioSolveIT). The parameters were set to a maximum number of results of 60 per query, minimum similarity of 0.10 and the ECFP4 fingerprint was chosen. After removal of already validated hits, 104 additional similars were subjected to in vitro analysis.

**NMR experiments:** Prior to measurements, LC3B, GABARAP and GABARAPL2 proteins were equilibrated with buffer containing 25 mM HEPES pH=7.0, 100 mM NaCl, 5% D2O and 0.15 mM DSS as internal reference. All NMR experiments were performed at a sample temperature of 298 K (24.85 °C) on cryogenic probes equipped with Bruker Avance spectrometers operating at proton frequencies of 600, 700, 800, 900, and 950 MHz. All NMR spectra were acquired and processed with Bruker TopSpin software (versions 3.6.5, 2.1, 3.6.2 and 4.3.0, respectively). The NMR spectra were analyzed with the Sparky 3.114 software (University of California, San Francisco, USA). For NMR titration experiments, selected compounds were titrated to 75 μM $^{15}$N-labeled LC3B, to 50 μM $^{13}$C,$^{15}$N-labelled GABARAP and to 25 μM $^{13}$C,$^{15}$N-labelled GABARAPL2 proteins (in standard 5 mm tube, total sample volume 600 μL) to molar ratios of 1:1 and 1:2 (protein:compounds). To achieve reliable calculation of $K_D$ values, more

titration points were performed for 1O5 (molar ratios 1:1, 1:2 and 1:8 by LC3B; 1:0.5, 1:1, 1:2 and 1:4 by GABARAPL2) and TH152 (molar ratios 1:0.125, 1:0.25, 1:0.5, 1:1, 1:2 and 1:4 for both LC3B and GABARAP), proportional to the compounds or complexes solubility. 2D $^1$H-$^{15}$N correlation spectra ([$^1$H-$^{15}$N] fHSQC [50] for novobiocin, [$^{15}$N,$^1$H] BEST-TROSY [51] for other compounds) were recorded at each titration point. HN backbone resonance assignments were transferred from the BMRB entries 5958, 5058 and 18827 (describing resonances of LC3B, GABARAP and GABARAPL2, respectively) and adopted to the current experimental conditions as reported in Stadel et al. [52]. CSP values, Δδ, were calculated for each individual backbone amide group using the formula $\Delta\delta = \sqrt{\frac{(0.2\Delta\delta N)^2 + (\Delta\delta HN)^2}{2}}$ according to the recent guidelines [53].

**Crystallization:** Initial crystallization hits were obtained by sitting drop vapor diffusion in SwissCi 3-drops plates using a series of commercially available coarse screens. Best hits were obtained in JCSG+ (Hampton Research, USA). Several rounds of optimization were done to meet the conditions required for XChem data collection (high resolution and reproducibility). The test crystals diffracted consistently around 2 Å and as high as 1.36 Å. The selected crystallization condition for further work consisted of 36% PEG 8000 and 0.1 M sodium acetate pH 4.7. For the fragment screening at XChem, the crystals were grown on-site using sitting drop vapor diffusion and the selected condition. The DSI poised compounds library had a stock concentration of 500 mM in DMSO. For the first round, the crystals were soaked at a 20% DMSO final concentration (equivalent to 100 mM of fragment concentration). Those conditions that failed, were re-soaked at 10% DMSO final concentration (equivalent to 50 mM of fragment concentration). No additional cryo-protectant was used. A summary of the assay, library, screen and post-screen analysis can be found in SI Table 5.

**Fragment Screening and Structure Solution:** A total of 808 fragments from the DSI poised library [54] (stocks dissolved in DMSO) were transferred to the LC3B crystallization drops using an ECHO liquid handler (20% final DMSO concentration) and soaked for 3 h before harvesting. Data was collected at the Diamond light source beamline I04-1. A total of 827 datasets were collected (including apo crystals), most of which diffracted to about 2 Å.

Data processing was performed using the automated XChem Explorer pipeline [55]. Fragment hits were identified using the PanDDA algorithm [56], followed by visual inspection. Refinement was performed using REFMAC [57].

## Reporting summary

Further information on research design is available in the Nature Portfolio Reporting Summary linked to this article.

## Data availability

The structural biology data generated in this study are available in the PDB under accession codes 8Q53 (truncated human Microtubule-associated proteins 1 A/1B light chain 3B (MAP1LC3B) in apo form), 7GAU (ground-state model of MAP1LC3B), 7GA8 (MAP1LC3B in complex with Z1198158918), 7GA9 (MAP1LC3B in complex with Z1198177230), 7GAA (MAP1LC3B in complex with Z1198233191), 7GAB (MAP1LC3B in complex with Z1255402624), 7GAC (MAP1LC3B in complex with Z1456069604), 7GAD (MAP1LC3B in complex with Z1667545918), 7GAE (MAP1LC3B in complex with Z1688504114), 7GAF (MAP1LC3B in complex with Z183352334), 7GAG (MAP1LC3B in complex with Z198195770), 7GAH (MAP1LC3B in complex with Z2033637875), 7GAI (MAP1LC3B in complex with Z212122838), 7GAJ (MAP1LC3B in complex with Z285233820), 7GAK (MAP1LC3B in complex with Z287121492), 7GAL (MAP1LC3B in complex with Z291279160), 7GAM (MAP1LC3B in complex with Z373768900), 7GAN (MAP1LC3B in complex with Z56767614), 7GAO (MAP1LC3B in complex with Z57450788), 7GAP (MAP1LC3B in complex with

Z728939702), 7GAQ (MAP1LC3B in complex with Z755044716), 7GAR (MAP1LC3B in complex with Z820676436), 7GAS (MAP1LC3B in complex with Z952016136) and 7GAU (ground-state model of MAP1LC3B). The mass spectrometry data for the affinity matrix experiments within this study are available in the MassIVE database under accession code MSV000093528 [https://doi.org/10.25345/C5804XW3S] (Chemoproteomic competition pulldown assays with immobilized drug molecules and free drug molecules in cell lysates). The remaining data generated in this study are provided in the Supplementary Information/Source Data file. Source data are provided with this paper.

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

## Acknowledgements

Compounds **1–4** were kindly provided by Akinori Toita and Misaki Homma (Takeda). M.P.S., A.K., F.A.G., N.B., T.H., V.M., C.L., J.D., L.B., K.S., S.M., A.S., S.K. and V.V.R. are grateful for support by the Structural Genomics Consortium (SGC), a registered charity (no: 1097737) that receives funds from Bayer AG, Boehringer Ingelheim, Bristol Myers Squibb, Genentech, Genome Canada through Ontario Genomics Institute, EU/EFPIA/OICR/McGill/KTH/Diamond Innovative Medicines Initiative 2 Joint Undertaking [EUbOPEN grant 875510], Janssen, Merck KGaA, Pfizer and Takeda and by the German Cancer Research Center DKTK and the Frankfurt Cancer Institute (FCI). M.P.S. is funded by the Deutsche Forschungsgemeinschaft (DFG, German Research Foundation), CRC1430 (Project-ID 424228829). S.L. and B.K. are funded by the Deutsche Forschungsgemeinschaft (DFG) SFB1309 (ID: 325871075, project ID: 401883058). The authors would like to acknowledge the Diamond Light Source for access to the fragment screening facility XChem, for usage of DSi-Poised library and for beam time on beamline I04-1 under proposal LB29658. S.C.-F., L.B., A.S. and S.K. are grateful for support by the BMBF program PROXIDRUGs. PROXIDRUGs as part of the initiative "Clusters4Future" is funded by the Federal Ministry of Education and Research BMBF (03ZU1109XX). Figure 4a was created in BioRender: Schwalm, M. (2024) https://BioRender.com/d80e756. We appreciate support of Prof. Volker Dötsch providing access to NMR equipment and for helping with these experiments.

## Author contributions

Figures were prepared by MPS who also drafted the manuscript which was edited by V.V.R. and S.K. M.P.S., J.D., A.K., F.A.G., J.H., S.C.-F., F.L., N.B., S.L., I.J., V.M., C.L., D.F., P.G.M., C.W.E.T. and L.B. contributed experimental data. Scientific supervision by T.H., N.B.P.A., F.v.D., A.S., B.K., S.M., K.S., S.K., E.P. and V.V.R.

## Funding

## Competing interests

The authors declare no competing interests.

## Additional information

¹Institute for Pharmaceutical Chemistry, Department of Biochemistry, Chemistry and Pharmacy, Goethe University Frankfurt, Max-von-Laue-Straße 9, 60438 Frankfurt, Germany. ²Structural Genomics Consortium, Buchmann Institute for Molecular Life Sciences, Goethe University Frankfurt, Max-von-Laue-Straße 15, 60438 Frankfurt, Germany. ³German Cancer Consortium (DKTK) / German Cancer Research Center (DKFZ), DKTK site Frankfurt-Mainz, 69120

Heidelberg, Germany. [4]Institute for Biophysical Chemistry, Department of Biochemistry, Chemistry and Pharmacy, Goethe University Frankfurt, Max-von-Laue-Straße 9, 60438 Frankfurt, Germany. [5]Fraunhofer Institute for Translational Medicine and Pharmacology ITMP, Theodor-Stern-Kai 7, 60596 Frankfurt, Germany. [6]Institute of Biochemistry II (IBC2), Faculty of Medicine, Goethe University, Theodor-Stern-Kai 7, 60590 Frankfurt am Main, Germany. [7]Buchmann Institute for Molecular Life Sciences (BMLS), Goethe University, Max-von-Laue-Straße 15, 60438 Frankfurt am Main, Germany. [8]Chair of Proteomics and Bioanalytics, TUM School of Life Sciences, Technical University of Munich, 85354 Freising, Germany. [9]NanoTemper Technologies GmbH, Flößergasse 4, 81369 Munich, Germany. [10]Diamond Light Source Ltd., Harwell Science and Innovation Campus, Didcot OX11 0QX, UK.
✉e-mail: knapp@pharmchem.uni-frankfurt.de; rogov@pharmchem.uni-frankfurt.de

