## [Peer Review file · Nature Communications]

Critical assessment of LC3/GABARAP ligands used for degrader development and ligandability of LC3/GABARAP binding pockets

Corresponding Author: Dr Vladimir Rogov

Version 0:

Reviewer comments:

Reviewer #1

(Remarks to the Author)

The Authors explore small molecule approaches toward the identification of autophagosome tethering compounds (ATTECs), a fairly novel and challenging field of research for potential drug development. Together with performing both in silico and x-ray based screenings, the Authors perform a number of biophysical assays to characterize previously published compounds, reporting that for the majority of them it was not possible to confirm the proposed mechanism of action. I found the data produced to validate the previously reported binders to be very compelling and useful. That is very interesting data and surely helpful to other researchers in the field, beside providing a more broad perspective on drug discovery campaigns in general.

The overall quality of the manuscript and the data provided is sufficient for being published, but there are some issues that I feel should be addressed prior to that.

My concern is regarding the structure-based work, where more details would have been very welcome. Also, there is some degree of disconnection between the different approaches pursued under this effort. The initial in-silico screening was instrumental to the identification of the first round of novel hits, generating what looks like an innovative scaffold that identified at least one important pharmacophoric feature (the presence of the two carboxylic groups). The affinity of these hits is also interesting because it is comparable to that of known and well-established hits.

However, it is not clear how the two hits were identified from the 173 compounds. Where these the only active molecules identified or just the best/most promising (and if so, according to which criterion)? Either way this information should be reported. Also, it would be great if the structures of the other 171 virtual hits would be made available, since that would be very informative to provide context to the results obtained. Without any details about the content of the (proprietary?) library, it is very difficult to evaluate the data generated from the screening. It would be very helpful to have more information about the content of such library, even if just by providing statistics about physicochemical properties of the molecules contained in it (MW ranges, HB count, clogP, etc.). This is relevant also because in the methods it is reported that a relatively high MW cutoff was used to filter out compounds. How that affects the outcome of the screenings is difficult to estimate without knowing the overall properties of the libraries, in light of the issues associated with all docking scoring functions.

Also, these hits were experimentally validated using 15N NMR, which provided a compelling support to the in-silico hypothesis, "confirming molecular modelling of these interactions", as stated in the manuscript. However, the structure of the molecular docking results was not provided, preventing the reader to assess the implications of these statements. Even more important, by providing this information, it would be more easy to connect the subsequent crystallographic fragment screening. In fact, it would be very interesting to see potential overlaps between the placement of chemical moieties of the first hits in the models with the x-ray hits. This lacking data is diminishing and conflicting with the following statement in the manuscript: "Based on our success with the very limited in silico study and the fragment screening campaign, we concluded that design and development of potent LC3/GABARAP ligands for ATTECs should be feasible."

Being this the major innovation of the work, I think it should be properly addressed, because it would increase the impact and the value of the manuscript.

Finally, the details about the setup of the virtual screenings are not sufficient. The Authors should add more information about the experimental setup such as size and location of the binding sites used for docking. Values of docking scores, (or their ranges, at least?) should be provided, too. Was the scoring scheme a consensus scoring (SeeSAR +AUTODOCK) or sequential scoring (first SeeSAR then AUTODOCK)? These details are essential to support at least a minimum degree of reproducibility. In the methods it is listed that 271 compounds were selected, which might be a typo?

Overall, while the results are compelling, a more critical and in-deep analysis of them would have been welcome, especially in light of the structural work performed. There is a perceived unbalance toward the validation of the known binders/inhibitors at the expenses of the innovation. The amount of critical discussion of the new results is relatively scarce, which is unexpected given that there are positive results to report.

Reviewer #2

(Remarks to the Author)

The paper titled 'Targeting LC3/GABARAP for degrader development and autophagy modulation' by Schwalm, M, et al. uncovers issues in current LC3/GABARAP-targeting compounds and generates novel LC3 fragments for the development of future ATTEC-based therapeutics. Using a wide range of biophysical techniques, currently available ATTEC ligands are screened for binding and selectivity at canonical and non-canonical binding sites. Finding that available ligands display poor/no binding capabilities, the authors use virtual and experimental fragment screening to generate scaffolds for future ATTEC development. Overall, this work provides value to the ATTEC field by correcting the existing literature and reinforcing the potential for development of legitimate LC3 ligands. The principal weakness is the perfunctory presentation of the results and lack of cheminformatic analysis; as such, the manuscript serves as a multi-mode screening report that confirms a recent report from the Genentech group (ref 22) showing that Atg8 proteins are legitimate targets for small molecule ligand discovery. Follow-up studies will be required to establish the utility of identified hit compounds. Below are specific comments to improve the manuscript.

1. The paper pursues several goals. Breaking the results up into sections with subheadings (e.g. "Validation of known ATTEC ligands", "LC3 Virtual Fragment Screening", etc.) would improve clarity. Likewise, the 2nd paragraph of the introduction should be split into two or more paragraphs for readability.
2. The discussion should be amended with a cheminformatic SAR analysis of the fragments and compounds identified in this study, as well as a comparison with the results from Steffek et al (ref 22).
3. English usage could be improved. For example, there are missing or misused articles (e.g. 'the') throughout the text.
4. Line 213: change "effecting" to "affecting"
5. Typo on line 285: ATTECT

Reviewer #3

(Remarks to the Author)

This paper reports the biophysical characterization of compounds which have previously been reported to bind LC3/GABARAP proteins, including some that have been used to generate autophagy tethering compounds (ATTECs). Fluorescence polarization and protein NMR indicate that many of these compounds do not interact with LC3/GABARAP in vitro, and the authors therefore cast doubt as to their suitability for use in ATTECs. They then report their own virtual screen and crystallography screen to identify ligands that bind to LC3/GABARAP.

The thorough biophysical characterization of reported LC3/GABARAP ligands is a useful addition to the field and highlights the need for caution when using published compounds. However, the new hits identified from the ligand screens reported here are not fully characterized, and no effort is made to show that these could be used for development of ATTECs. Indeed, the authors themselves note that 'the identified ligands might be characterized by poor cell penetration and might therefore require optimization for ATTEC development'. This limits the potential impact and also does not correlate with the title of the paper.

Key strengths

- The field of induced proximity has rapidly expanded in recent years and novel molecules in this area are frequently reported, but sometimes without the necessary rigorous characterization. This paper addresses this by characterizing reported ligands that have been used to generate ATTECs.
- A variety of biophysical techniques are used to characterize previously published ligands, including fluorescence polarization, MST, NMR, mass spectrometry and ITC.

Key weaknesses

- The title and abstract do not accurately represent the content of this paper. The title 'Targeting LC3/GABARAP for degrader development and autophagy modulation' implies that the paper will report molecules that do this, but in fact, the data indicate that existing molecules reported to target LC3/GABARAP may not act in this way, and the new ligands identified are not used in degrader development or shown to modulate autophagy. In addition, the abstract states 'Here, we present diverse comprehensively validated ligands for future ATTEC development', but the new ligands are not comprehensively validated, nor is there any evidence that they could be used for future ATTEC development. The crystallography hits are not confirmed with biophysical or biochemical data, and only two virtual screening hits are confirmed to bind by ITC and NMR, binding LC3/GABARAP weakly with K_d 1.9-10 μ M. It is not clear why the authors use several assays to investigate

previously reported ligands, but do not follow the same level of characterization for their own hit compounds to fully validate them.

- Biophysical characterization indicates that the reported compounds 1-4 do not significantly bind LC3/GABARAP in vitro, but no attempts are made to replicate the previously reported cellular activity of these compounds. It would be beneficial to know whether these compounds have any effect on autophagy in the authors' hands, and therefore could act by alternative mechanisms, or whether they are inactive.
- It is not clear how many biological replicates have been performed. For example, the figure legends state 'Data were collected in technical triplicates with error bars expressing the SD (n=3).' Does n=3 refer to the technical triplicates or does this mean biological replicates? At least two independent experiments should be performed to enable accurate conclusions to be drawn, and this needs to be stated clearly.
- An FP assay is established using Cy5-labelled p62-LIR peptide as the probe, but it is not explained where this peptide binds on LC3, and it needs to be acknowledged that a lack of activity in this assay does not necessarily mean that the compounds do not bind to LC3/GABARAP as they could bind non-competitively at a different binding site on the protein.
- Most experiments reported here do not show compound binding to LC3/GABARAP, in contrast to previously published work. However, in this paper the authors labelled cysteine instead of lysine residues, when the data would be more comparable and the conclusions stronger if the same conditions were used.
- It is odd to mention thermal shift assays (line 122) when no interpretation is made of this data and the curves are not included anywhere. If the assay is deemed unsuitable and is not used for compound analysis, then it does not need to be reported.
- Line 91, the authors state 'no tool compounds such as autophagy pathway inhibitor are available', however, several autophagy related inhibitors are available such as 3-methyladenine, bafilomycin A, chloroquine and can be used in the field.
- Line 189 states 'To investigate covalent interactions of compounds 1 and 4 with LC3/GABARAP as recently reported for 1 with the E3 ligase DCAF11, we also studied the interaction of compound 1 and 4 with GABARAPL2 using ESI mass spectrometry, where no covalent modification was detected in vitro.' – reference to the data missing.
- Line 196 states 'For proteome-wide screening, we modified 10O5 (4) and 8F20 (3) with PEG-based linkers that can be immobilized on sepharose beads to generate an affinity matrix for target pulldown, resulting in compounds 18 and 19. As expected, dose-dependent competition assays showed that the KIF11 inhibitor 8F20/Ispinesib (3) selectively bound to KIF11 in HEK293T lysates (EC₅₀ of 290 nM). No additional targets were detected for 19 confirming excellent selectivity of this inhibitor for KIF11 (Fig. 3d).' Proteomics data is not shown anywhere in the paper. Fig 3d only shows competition data for compound 18 interacting with KIF11; how can it be stated that 'No additional targets were detected' if only one target has been monitored?
- 'Druggable' is frequently used instead of 'ligandable', for example, line 248 'Our screen confirmed that HP2 is the most druggable binding site on 248 LC3/GABARAP proteins surface, accommodating 10 from 21 identified fragments'. This is not true as the fragments have not been confirmed to bind by orthogonal techniques, and binding does not mean that the site is druggable (functional).
- Figure legends should be more specific and contain more information. For example, all FP data is described as 'Fluorescence polarization assay curves', but it appears that these are displacement curves so the probe should be stated each time.
- Has the data been corrected for probe background fluorescence? Fig 4B states 'Structural representation and corresponding ITC data for the two screening hits (compounds 20 and 21)', what proteins were used here? Fig 4D does not state what each of the different colored spectra correspond to.
- Insufficient data are provided on the virtual screen – where were the compounds docked on the proteins? Does the NMR data shown in Fig 4D, E agree with the initial docking site?

Some experimental details are not clear:

- Chemistry experimental methods are given for compounds 19, 17, 18, 16 (in a random order), but schemes are missing, and it is not stated where the molecules used in this paper came from.
- Experimental method 'Affinity determination using spectral shift mode on Dianthus', it is not clear what data/assay this refers to, is it MST?
- 'Fluorescence polarization assay (FP assay): For the complementation assay' – what is the complementation assay, should it be displacement assay?
- 'Covalent compound screening: For screening of covalent compounds, 46 compounds were tested against all LC3/GABARAPs, purified as described above' – what are these 46 compounds? Fewer compounds were discussed in the text. Does 'purified as described above' refer to the compounds or the proteins?
- Crystallization – What protein construct was used?
- Fragment screening and structure solution – what concentrations were the compounds soaked at? What cryo protectant was used?

Minor points

- FP graphs should use a shorter y axis scale (e.g. 0 - 120mP) to make the data easier to see.
- Fig2D the yellow and green spectra are very similar in color; it would be better to use a different color for one of them.
- Compound 16 and 17 structures are buried in the supplementary and difficult to find, they should be referenced more clearly when mentioned in the text.

Reviewer #4

(Remarks to the Author)

In this manuscript, Schwalm et al. focus on LC3/GABARAP-targeting ligands, characterising previously developed molecules and identifying novel chemical scaffolds. Protein recruitment to LC3/GABARAP through bifunctional small molecules, named autophagosome tethering compounds (ATTECs), has been previously explored as one of the different approaches to induce targeted protein degradation.

The authors set out to validate existing LC3/GABARAP ligands used in ATTECs design, testing target engagement with different biophysical methods. The results reveal weak binding (in some cases no binding) of existing ATTEC ligands to LC3/GABARAP, highlighting the need to further characterise the cellular mechanism of action of ATTECs. In line with this observation, a recent study (Xue et al) shows that arylidene-indolinones, previously reported as LC3B binders, mediate protein degradation via DCAF11.

In this context, it is certainly essential to perform a thorough characterisation of widely used chemical probes. However, in some parts of the manuscript, the authors seem to undermine any measured binding with recombinant LC3/GABARAP as weak or insignificant, although it is not yet clear what binding affinities are necessary to induce autophagy via ATTECs.

Before publication, the authors should address the following points:

- Some examples of paragraphs where the authors should consider rephrasing statements dismissing observed binding:
 - oLines 136-139, referring to Fig 2c and supplementary figure 2D;
 - oLines 144-145: “none of the compounds 1-4 resulted in significant CSP even at higher compound concentrations in agreement with our FP binding data.” Supplementary figure 3 shows some shifts for compound (4). Also, please improve the layout/quality of this figure, as well as supplementary Fig. 5, as it is really difficult to look at the NMR spectra.
 - oLines 169-178: despite observing micromolar binding of (4) with GABARAP family members, the authors define it again as not significant.
 - oLines 186-187: “raising the possibility of further off-targets within the proteome, based on the reactivity of these compounds”. The authors should not make this claim without supporting data.
- MST experiments: the compound labelling in supplementary figure 4a should be corrected. Did the authors test (12) or (13) or any of the peptides using MST? do they show binding?
- Lines 155-157: “Novobiocin (12) revealed binding with a KD (6.7 μ M for LC3A), while titrations with AN2 (2) and 8F20 (3) did not yield significant binding heats”. These ITC experiments suggest no binding or weaker binding (not measurable at the concentrations tested). Dong et al. were able to measure binding of (3) using ITC (Kd \sim 12 μ M) at higher protein and compound concentrations. Did the authors try performing the ITC titration in those conditions?
- Regarding compounds (4) and (1): are they tested as E/Z mixture? was the mixture purified in previous papers? is one of the two isomers more active?
Tracer (16) is instead represented as pure isomer in Supplementary figure 4C. If the (4) E/Z isomers bind differently, this will result in lower effective concentration of the ‘active’ isomer and thus affect its ability to displace the tracer. Additionally, in comparison to tracer (17), (16) contains an α,β -unsaturated carbonyl portion next to the Py-BODIPY moiety, is that a drawing mistake?
- Supplementary fig. 4D: did the authors try lower tween concentrations? or, in the absence of tween, displacement with one of their validated ligands (Novobiocin or peptides)?
- Lines 196-206: This section describing pull-down experiments requires some clarifications (also in the corresponding methods section) and additional controls. The authors should show a positive control (e.g. using a peptide or one of the better binders) to demonstrate that it is possible to measure enrichment of LC3/GABARAP with this setup. Also, what does it mean that there was no enrichment “using 18 (up to 30 μ M)” if the compound is conjugated to sepharose beads?
- It would be helpful to include a description of the NanoBRET assay in the main text. The authors say that there is a moderate BRET increase but no displacement. It would be helpful to further confirm this by testing displacement with one of the stronger binders.
- Supplementary fig. 10 - line 196: “Crystallized fragments bound to LC3A” contradicts the rest of the text, where it’s LC3B (e.g. page 39 line 738)
- Did the authors try to soak/co-crystallise any of the previously described ligands with LC3/GABARAP? Some of the compounds showing weak binding in this manuscript could potentially be useful starting points for optimisation, similarly to the newly identified fragments.

Reviewer #5

(Remarks to the Author)

The manuscript by Schwalm et al. “Targeting LC3/GABARAP for degrader development and autophagy modulation” is an impressive dualistic exercise where on one hand the authors rigorously test published ATTECs (autophagosome tethering compounds) for binding to LC3/GABARAP family proteins, and on the other hand perform in silico docking and large scale crystallographic fragment screening to identify a group of compounds they validate as ligands for future ATTEC development. Perhaps surprisingly, and certainly disappointingly, the authors find that the published ATTECs do not bind at

all (almost all tested compounds), or too weakly, to LC3/GABARAPs for a specific biological effect to be explained. This is a very important finding to be communicated pointing to a too poor characterization and binding validation effort done in the published work on ATTECs: This is clearly damaging to the field as such and cries for better evaluation of such papers before they are accepted for publication. In the second part of their paper the authors use in silico docking and large scale crystallographic fragment screening to identify a number of compounds that do bind to the LC3/GABARAPs and also identify two new binding surfaces. A conclusion is also that it is the hydrophobic pocket 2 (HP2) that is the most druggable in the LIR Docking Site (LDS) while HP1 interacting with the aromatic residue of the core LIR is much less accessible for the compounds.

This is clearly an important paper with thoroughly performed experiments using an impressive array of assays to validate binding both in vitro and in cells in addition to structure explorations techniques like NMR and x-ray crystallography. The "negative" half of the paper, showing that published ATTEC compounds basically do not bind to LC3/GABARAPs, deserves to be communicated widely in a very visible journal. The "positive" half of the paper stops when it starts to become exciting. A number of candidate compounds are presented that can be used to develop new ATTECs are identified, but we do not see any of these taken to the next level and we are left wondering: Is ATTECs at all a viable approach? For the paper to be acceptable for publication in Nature Communications the authors need to go further with one or more of the compounds they have identified and test in a proof of principle setting if they can see targeted degradation using such a compound equipped with a target recognition part connected with a linker to the LC3/GABARAP interacting compound.

Reviewer #6

(Remarks to the Author)

In the paper titled 'Targeting LC3/GABARAP for degrader development and autophagy modulation' by Martin P. Schwalm et al. The authors shed light on the principles of autophagosome tethering compounds (ATTECs) based on the target recruitment to LC3/GABARAP. The researchers assessed the published ATTEC ligands using biophysical techniques including FP, ITC, and NMR. The authors note their surprise that the literature identified ligands do in fact not bind to their intended target LC3. They then go forth to use their world leading expertise to identify a number of small molecule ligands through virtual screening and high-throughput fragment via X-ray crystallography. Presenting a diverse comprehensively validated series of ligands for future ATTEC development.

The present study is well designed, and the authors have given a comprehensive insight into the binding mechanism of ATTEC compounds. The results of this study is of great use to the drug discovery community not only from the ligands for future ATTEC development but also from reagents produced in the LC3/GABARAP systems and experimental methods. I would suggest this study for publication with some minor alterations and points of discussion.

Minor Points:

1) When debunking a previously reported study it is difficult to be completely resolute in biophysical data, as a number of factors, such as PTM and binding partners may play a roll in the published ATTEC ligands that are hard to replicate in a reconstituted cell-free/cellular manner. Although the studies data is strong and very compelling, I would suggest a sentence to this extent.

2) In a pleasant way, it is hard to read the publication and not ask "what's next?" I assume this is out of the scope of this study and would be a major addition to the publication.

i) Building on the positive outcome from the virtual screen could the authors possibly resolve a structure of TH152 bound to LC3B

ii) Could the assays built within the invalidation of the published ATTEC ligands be used against the crystallographic output, even if they are likely to be weak binders?

Version 1:

Reviewer comments:

Reviewer #1

(Remarks to the Author)

Some of the issues raised previously have been addressed, but there are still some outstanding issues regarding the discussion of the in silico methods used, which affect the quality of the results and their reproducibility.

While it is well-received that the Authors provided clarifications in the document for the reviewers, most (if not all of them) are essential for the reader, and they should be in the manuscript.

For example, the manuscript still lacks details about the following:

- chemical properties of the library used
- details of the similarity search (Tanimoto similarity of "what versus what"? what cutoff?)
- details of the site(s) used for the molecular dockings
- identity of the residues included
- search parameters used for the virtual screening (if "default" were used, it should be explicitly stated)
- detailed comparison between residues identified in the dockings versus the experimental validation

The recommendation to provide more critical discussion of the hits found and how these can be used for driving further

research in this area (an issue that apparently other reviewers raised as well).

Reviewer #2

(Remarks to the Author)

The revised manuscript satisfies the concerns I raised with the original version. I note the following grammatical corrections in the discussion section:

line 292 - 'events indeed are' (or 'event indeed is')

line 324 - insert comma after 'changes' (I think; this sentence is awkward)

Reviewer #3

(Remarks to the Author)

This manuscript details the characterization of ligands targeting LC3/GABARAP proteins. This includes a thorough assessment of literature ligands targeting LC3/GABARAP that have been used in ATTECs, autophagosome targeting compounds, identifying key concerns with literature reference compounds and their proposed mechanism of action, due to their inability to engage LC3/GABARAP proteins in vitro. The authors then provide additional chemical matter, identified through in silico and fragment screening, which bind to multiple sites on LC3 proteins and may provide new starting points for the development of ATTECs.

Whilst a thorough characterisation of literature ATTEC ligands is provided and is necessary for the community, there is concern regarding the inclusion of two separate 'stories' – ATTEC characterisation and novel screening strategies against LC3 – within the same manuscript. Without confirmation of ATTEC mechanism of action, and the validation of novel ligand activity, the ATTEC characterisation alone should be published to avoid muddying the literature in an already problematic space.

Major points

Without confirmation that the ATTEC proposed mechanism can be achieved in cells, the fragments and in silico compounds are of limited use, as per the statement in the manuscript that LC3 binders alone are insufficient to trigger autophagy. As these ligands are not potent enough to be used in the development of ATTECs, the autophagy-induced degradation mechanism could be confirmed using a HaloTag-LC3 construct and a HaloTag-ligand to a known protein of interest or other tag.

Readers may incorrectly use these compounds, particularly 20 and 21, without further development or characterisation. We would recommend the inclusion of an example of a developed molecule from a fragment that exhibits measurable potency following the proposed fragment growing/linking strategy to highlight the plausibility of this approach, and to emphasise the need for compound optimisation prior to ATTEC generation. Without this, this part of manuscript should be removed for publication at a later date. At the very least, it needs to be included in text that KDs could not be determined for the fragments due to weak binding, as stated in the response to reviewers. If the screening data is still included, some closer views of hotspots (eg the HP2 site) for preliminary SAR would also be useful for the reader.

Minor points

Was ITC attempted to identify the KD of the fragments?

Page 6 – the p62:LC3 interaction is clearly important, resulting in the use of the p62 LIR peptide, but the context of this interaction within autophagy/the cell is not clearly outlined in the introduction

Page 9 – the point regarding KIF11 was a little confusing. The fact that compound 3 is an inhibitor of KIF11 had not been previously introduced; recommend providing additional context to this, perhaps when 3 is introduced to the manuscript, for reader clarity. Furthermore, is there any evidence eg in databases that show it would be expected to be able to identify LC3 or GABARAP proteins by proteomics (eg tryptic peptides, known expression levels in cell line)?

Page 14 line 324 – “by affine LC3/GABARAPs ligands” – the word affine is misused in this sentence.

Figure 2e – the light green text is challenging to read, it is suggested to make this a little darker

Figure 4b – could the authors comment on the relative enthalpy change for the two compounds; the maximum for TH152 is significantly smaller than for LY223962 (and novobiocin, in supplementary 4b) and more comparative to the inactive compounds shown in the supplementary? Is this due to compound solubility?

Figure 4f – the pink amino acid labelling is challenging to see against the background of the blue/grey. Recommend making the background lighter/more transparent and/or changing the pink colour of the ligand.

There seems to be no compound 14 or 15, have these numbers been missed in the manuscript, or are some additional compounds not referenced in text?

Reviewer #4

(Remarks to the Author)

The authors have addressed most of the points that were raised. However, one major concern pointed out in response to reviewer #3, is the number of independent experiments performed (e.g. ITCs with n=1).

In addition, a few minor points are listed below:

The chemical structure with the α,β -unsaturated carbonyl moiety (Py-BODIPY) in Supplementary figure 4C has not been corrected.

Regarding supplementary figure 4a - the authors include the positive control for the measurements (p62-LIR) only in the data for reviewers. It would be helpful to include this in the published manuscript. Was this performed in the same conditions as supplementary figure 4a? Following the comment and response to reviewer #3, does n=2 here refer to independent experiments or technical replicates?

Also, in the supplementary figure 4 legend, "lover table" is maybe "lower table"?

In the discussion section, given that some weak binding has been measured for some compounds, I would suggest modifying the sentence "none of the ligands used for the development of ATTECs interacted with LC3/GABARAP", specifying that this refers to profiles of selectivity/affinity expected for chemical probes.

Reviewer #5

(Remarks to the Author)

The authors have answered the single critical comment I had in a satisfactory manner.

Reviewer #6

(Remarks to the Author)

Reviewing the amendments to the paper titled 'Targeting LC3/GABARAP for degrader development and autophagy modulation' by Martin P. Schwalm et al. As before the study is well designed, and the authors have responded to my previous minor suggestions adequately. The results of this study are of great use to the drug discovery community not only for future ATTEC development but also from reagents produced in the LC3/GABARAP systems and experimental methods.

I would still be keen to see some other experiments/points of note, more out of interest than anything.

1. Although TH152 struggles to be cell permeable potentially due to the two carboxylic acids, or likely high protein binding are you able to show target engagement with the NanoBRET with lysed cells? The idea is you would have an additional measure, although likely/expected to be weaker due to the likely high protein binding of TH152?

2. Or further this point of target engagement looking for some extra evidence, could you add a -PEG-linker to TH152 and look at this in your affinity matrix pull down experiments?

I would suggest this study for publication, but still unsure in this journal tier in the absence of an example of ligand in use.

We would like to thank the reviewers for their insightful comments and constructive criticism. We have now revised the manuscript considering all issues that have been raised. Below we provide a point-by-point response to each comment as well as an outline of change we made in the revised manuscript

Reviewer #1:

The Authors explore small molecule approaches toward the identification of autophagosome tethering compounds (ATTECs), a fairly novel and challenging field of research for potential drug development. Together with performing both in silico and x-ray based screenings, the Authors perform a number of biophysical assays to characterize previously published compounds, reporting that for the majority of them it was not possible to confirm the proposed mechanism of action. I found the data produced to validate the previously reported binders to be very compelling and useful. That is very interesting data and surely helpful to other researchers in the field, besides providing a broader perspective on drug discovery campaigns in general. The overall quality of the manuscript and the data provided is sufficient for being published, but there are some issues that I feel should be addressed prior to that.

Response: We thank the reviewer for this summary and the encouraging comments.

My concern is regarding the structure-based work, where more details would have been very welcome. Also, there is some degree of disconnection between the different approaches pursued under this effort. The initial in-silico screening was instrumental to the identification of the first round of novel hits, generating what looks like an innovative scaffold that identified at least one important pharmacophoric feature (the presence of the two carboxylic groups). The affinity of these hits is also interesting because it is comparable to that of known and well-established hits.

However, it is not clear how the two hits were identified from the 173 compounds. Where these the only active molecules identified or just the best/most promising (and if so, according to which criterion)? Either way this information should be reported. Also, it would be great if the structures of the other 171 virtual hits would be made available, since that would be very informative to provide context to the results obtained. Without any details about the content of the (proprietary?) library, it is very difficult to evaluate the data generated from the screening. It would be very helpful to have more information about the content of such library, even if just by providing statistics about physicochemical properties of the molecules contained in it (MW ranges, HB count, clogP, etc.). This is relevant also because in the methods it is reported that a relatively high MW cutoff was used to filter out compounds. How that affects the outcome of the screenings is difficult to estimate without knowing the overall properties of the libraries, in light of the issues associated with all docking scoring functions.

Response: We thank the reviewer for taking a close look at the virtual screening approach. To answer this question, we re-evaluated the virtual screening hits and updated the hits in Figure 4. The 271 most promising compounds (called virtual screening hits in the manuscript) from the virtual screening were validated in vitro. Of these, TH152 and LY223982 showed significant displacement and were therefore selected for further validation of their interaction with LC3. The selection of these hits was based on the best scores in the respective programme (estimated affinity for SeeSAR and estimated free energy of binding for Autodock) and our selection cut-off was also based on the screening capacity in the laboratory.

The distribution of physicochemical properties of the screened compound library is shown in Reviewer Response (RR) RR_Figure 1. A 750 Da cut-off was used to filter out large library compounds, as such molecules often give rise to high estimated free energies in Autodock, probably due to large entropic contributions and the large number of intramolecular interactions.

RR_Figure 1: Physicochemical properties of our in-house library computed with rdkit

Also, TH152 and LY223982 were experimentally validated using ^{15}N NMR, which provided a compelling support to the in-silico hypothesis, "confirming molecular modelling of these interactions", as stated in the manuscript.

RR_Figure 2: Docking poses of TH152 to LC3A (PDB ID 6TBE)

As an example, the obtained docking poses of TH152 are depicted in RR_figure 2. Additionally, we used these poses to compare them with the interaction regions determined through NMR and were able to show overlap between the experimental and in silico methods. The results of this comparison are now depicted in Figure 4 F, the new SI Figure 11 and are now discussed in the main text of the paper.

Being this the major innovation of the work, I think it should be properly addressed, because it would increase the impact and the value of the manuscript. Finally, the details about the setup of the virtual screenings are not sufficient. The Authors should add more information about the experimental setup such as size and location of the binding sites used for

docking. Values of docking scores, (or their ranges, at least?) should be provided, too. Was the scoring scheme a consensus scoring (SeeSAR +AUTODOCK) or sequential scoring (first SeeSAR then AUTODOCK)? These details are essential to support at least a minimum degree of reproducibility In the methods it is listed that 271 compounds were selected, which might be a typo?

Response: The 271 compounds were selected on the basis of their *in silico* score in one of the two docking programmes used. Thus, only these 271 compounds were evaluated in the FP experiments. Furthermore, in a second screening experiment, we selected 104 compounds that showed structural similarity to TH152 and LY223982, as these compound hits were part of a SAR series available in the laboratory. RR_Figure 3 shows the distribution of docking scores.

RR_Figure 3: Distribution of docking scores (estimated free Energy calculated by AutoDockGPU and estimated affinity by SeeSar). The compounds used for *in vitro* validation after filtering are highlighted in purple, the remaining compounds are depicted in green.

RR Figure 4: Structures and binding sites (HP1+HP2) used in virtual screening (left: LC3A, PDB ID 6TBE, right: GABARAP, PDB ID 4XC2)

Overall, while the results are compelling, a more critical and in-deep analysis of them would have been welcome, especially in light of the structural work performed. There is a perceived unbalance toward the validation of the known binders/inhibitors at the expenses of the innovation. The amount of critical discussion of the new results is relatively scarce, which is unexpected given that there are positive results to report.

Response: We have expanded our Discussion section to improve the balance and provide a broader overview of the prospects for rational design of ATTECs. We believe that the number of chemically diverse hits identified in the high throughput fragment screen provides an indication of the ligandability of the binding site. However, the fragment interactions were too weak to allow an accurate assessment of their affinities. The HP1 pocket appeared to be most accessible to many structurally diverse fragments, with only one fragment identified for the HP1 pocket that bound to the edge of the pocket. This was certainly surprising and highlights a potential problem with the accessibility of this binding site (as suggested in J Cell Biochem. 2023 Feb 13. doi: 10.1002/jcb.30380. We have expanded the discussion of the fragment data in the revised version of the paper.

Reviewer #2:

The paper titled 'Targeting LC3/GABARAP for degrader development and autophagy modulation' by Schwalm, M, et al. uncovers issues in current LC3/GABARAP-targeting compounds and generates novel LC3 fragments for the development of future ATTEC-based therapeutics. Using a wide range of biophysical techniques, currently available ATTEC ligands are screened for binding and selectivity at canonical and non-canonical binding sites. Finding that available ligands display poor/no binding capabilities, the authors use virtual and experimental fragment screening to generate scaffolds for future ATTEC development. Overall, this work provides value to the ATTEC field by correcting the existing literature and reinforcing the potential for development of legitimate LC3 ligands.

Response: We thank the reviewer for these comments.

The principal weakness is the perfunctory presentation of the results and lack of cheminformatic analysis; as such, the manuscript serves as a multi-mode screening report that confirms a recent report from the Genentech group (ref 22) showing that Atg8 proteins are legitimate targets for small molecule ligand discovery. Follow-up studies will be required to establish the utility of identified hit compounds. Below are specific comments to improve the manuscript.

1. The paper pursues several goals. Breaking the results up into sections with subheadings (e.g. "Validation of known ATTEC ligands", "LC3 Virtual Fragment Screening", etc.) would improve clarity. Likewise, the 2nd paragraph of the introduction should be split into two or more paragraphs for readability.

Response: We agree with the suggestion of the reviewer and re-structured the result section as suggested. We also included the suggested subheadings and improved the Introduction.

2. The discussion should be amended with a cheminformatic SAR analysis of the fragments and compounds identified in this study, as well as a comparison with the results from Steffek et al (ref 22).

Response: We have now added a more comprehensive SAR analysis in the Discussion section, as well as a comparison with the fragments reported by Steffek et al.

3. English usage could be improved. For example, there are missing or misused articles (e.g. 'the') throughout the text.

4. Line 213: change "effecting" to "affecting"

5. Typo on line 285: ATTECT

Response: We thank the reviewer for highlighting these typos. They have been corrected as suggested by the reviewer.

Reviewer #3:

This paper reports the biophysical characterization of compounds which have previously been reported to bind LC3/GABARAP proteins, including some that have been used to generate autophagy tethering compounds (ATTECs). Fluorescence polarization and protein NMR indicate that many of these compounds do not interact with LC3/GABARAP in vitro, and the authors therefore cast doubt as to their suitability for use in ATTECs. They then report their own virtual screen and crystallography screen to identify ligands that bind to LC3/GABARAP.

The thorough biophysical characterization of reported LC3/GABARAP ligands is a useful addition to the field and highlights the need for caution when using published compounds. However, the new hits identified from the ligand screens reported here are not fully characterized, and no effort is made to show that these could be used for development of ATTECs. Indeed, the authors themselves note that 'the identified ligands might be characterized by poor cell penetration and might therefore require optimization for ATTEC development'. This limits the potential impact and also does not correlate with the title of the paper.

Response: The current paper was mainly focussed on the validation of the published ATTEC systems. These data have been published in high impact journals and as a new design strategy for degrader development it has received significant attention in the community with several follow-up papers that were however all from the same research team. We are aware that the data on ATTECs are "negative data" and as such less exciting. But we know that the ATTEC system has been unsuccessfully reproduced by many labs. We believe therefore that sharing these data will be important for the community of scientists working on degraders. A second aspect that we wanted to address was the druggability and the chemistry needed to for a rational ATTEC development. We therefore provided the fragment screening data, evaluation

of other published LC3 ligands as well as a panel of assay system that can be used for LC3 ligand development. Considering the small size of the screened in silico library the hit finding campaign was very successful. In fact, the two identified in silico hits are the most potent LC3 ligands known to date. The development of potent ligands for LC3 that would be suitable for ATTEC development will require the synthesis and evaluation of many hundred compounds (and as a consequence 2-3 years of experimental work). We therefore did not include LC3 ligand development into this paper as we wanted to share these data early. We also think that the large body of structural data on LC3 fragments will allow the chemical biology community to contribute to the ligand development process.

Key strengths

- *The field of induced proximity has rapidly expanded in recent years and novel molecules in this area are frequently reported, but sometimes without the necessary rigorous characterization. This paper addresses this by characterizing reported ligands that have been used to generate ATTECs.*
- *A variety of biophysical techniques are used to characterize previously published ligands, including fluorescence polarization, MST, NMR, mass spectrometry and ITC.*

Response: We thank the reviewer for this accurate summary and recognition of our efforts.

Key weaknesses

- *The title and abstract do not accurately represent the content of this paper. The title 'Targeting LC3/GABARAP for degrader development and autophagy modulation' implies that the paper will report molecules that do this, but in fact, the data indicate that existing molecules reported to target LC3/GABARAP may not act in this way, and the new ligands identified are not used in degrader development or shown to modulate autophagy. In addition, the abstract states 'Here, we present diverse comprehensively validated ligands for future ATTEC development', but the new ligands are not comprehensively validated, nor is there any evidence that they could be used for future ATTEC development. The crystallography hits are not confirmed with biophysical or biochemical data, and only two virtual screening hits are confirmed to bind by ITC and NMR, binding LC3/GABARAP weakly with K_d 1.9-10 μ M. It is not clear why the authors use several assays to investigate previously reported ligands, but do not follow the same level of characterization for their own hit compounds to fully validate them.*

Response: We agree with the reviewer that the title need to be changed to reflect better the presented data. We change therefore the title to: **Critical assessment of LC3/GABARAP ligands used for degrader development and ligandability of LC3/GABARAP binding pockets**. We also edited the abstract and deleted statements that current ligands can be used for ATTEC development. We tried to measure affinities of the crystallized fragments. However, the identified fragments are still too weak to measure accurate affinity data by SPR and other technologies. The identified in silico hits were however comprehensively characterized and considering all ligands published for LC3/GABARAP, they represent the most attractive starting points for development of potent ligands. As the intension of the ligands and fragments together with their binding modes was to serve as chemical starting points, we did not fully characterize them.

Biophysical characterization indicates that the reported compounds 1-4 do not significantly bind LC3/GABARAP in vitro, but no attempts are made to replicate the previously reported cellular activity of these compounds. It would be beneficial to know whether these compounds have any effect on autophagy in the authors' hands, and therefore could act by alternative mechanisms, or whether they are inactive.

Response: We were very concerned that the ligands used for ATTEC development might induce general toxicity in cell lines, as they are based on drugs used in oncology (such as the kinesin inhibitor 1005) and the isoxazole scaffold often used in the development of multi-targeted kinase inhibitors. However, we have now investigated compounds 1-4 in autophagy flux assays using the RPE-1 GFP-LC3B-RPF-LC3 Δ G reporter cell line (single clone) as

reported in [PMID:37558952]. We found that 10O5 and AN2 (RR_Figure 5) did not induce autophagy in this assay system, but AN1 and 8F20 showed significant autophagy activation (RR_Figure 6). The authors of the first manuscript reporting these ligands [PMID: 3166698] did not report autophagy induction by 8F20 and AN1. However, due to the potent activity of 8F20 on KIF11, which will have profound effects on cell trafficking and growth, and DCAF11 (AN1, [PMID: 38036533]), we find it unlikely that the effects on autophagy are due to LC3/GABARAP modulation. Due to the lack of target (LC3/GABARAP) binding *in vitro* and the unclear mechanisms of autophagy induction, we only present these data here for the purposes of this review but have excluded the data in the manuscript.

RR figure 5: Autophagy flux measurement for 10O5 and AN2. No induction of autophagy or effects (right panels, assessed by Incucyte (Sartorius) life cell imaging) on cell growth was observed.

RR figure 6: Autophagy flux measurement for 8F20 and AN1 show changes in autophagy flux upon compound treatment. However, also profound effects on cell growth were observed using life cell imaging.

It is not clear how many biological replicates have been performed. For example, the figure legends state 'Data were collected in technical triplicates with error bars expressing the SD (n=3)'. Does n=3

refer to the technical triplicates or does this mean biological replicates? At least two independent experiments should be performed to enable accurate conclusions to be drawn, and this needs to be stated clearly.

Response: We have now included the number of replicates and technical duplicates in the figure legends. However, NMR and ITC titrations were not repeated due to the large amount of protein material and instrument time required for these experiments.

An FP assay is established using Cy5-labelled p62-LIR peptide as the probe, but it is not explained where this peptide binds on LC3, and it needs to be acknowledged that a lack of activity in this assay does not necessarily mean that the compounds do not bind to LC3/GABARAP as they could bind non-competitively at a different binding site on the protein.

Response: We now describe the binding of the p62 LIR peptide in the text. We have also included a statement that compounds that are not LIR competitive would not be detected by this assay. However, the additional tracer-independent assays (spectral shift assay – analogue of commonly used MST, ITC, SPR and NMR) allow us to exclude binding to other sites in LC3/GABARAP.

Most experiments reported here do not show compound binding to LC3/GABARAP, in contrast to previously published work. However, in this paper the authors labelled cysteine instead of lysine residues, when the data would be more comparable and the conclusions stronger if the same conditions were used.

Response: LC3/GABARAP proteins contain a number of Lys residues (LC3A - 9, LC3B - 10, LC3C - 9, GABARAP - 12, GABARAPL1 - 13, GABARAPL2 - 12). 2 lysines in LC3s (K49 and K51 in LC3A) and 2 lysines of GABARAPs (K46 and K48) are located within the LDS binding site and chemical modification of these could interfere with the binding of peptides/compounds. More specifically, Fan et al. showed that covalent modification of K49 (by molecules even smaller than a fluorescent tag) was sufficient to suppress peptide binding [PMID: 34590387]. We therefore consider labelling at cysteine residues a more prudent approach developing this assay format. We would like to mention that the authors of the paper reporting the development of ATTECs used 500 nM protein in their MST assay. This is a protein concentration about 1-2 magnitudes higher than usually used in MST assays. In addition, lysine labelling might lead to incorporation of several labels and signal quenching. As there is only one cysteine present, we avoided this complication. Finally, we confirmed our findings using label free methods (ITC, NMR, FP) which resulted in similar conclusions.

Line 91, the authors state 'no tool compounds such as autophagy pathway inhibitor are available', however, several autophagy related inhibitors are available such as 3-methyladenine, bafilomycin A, chloroquine and can be used in the field.

Response: We agree with the reviewer and have corrected the text to read "tool compounds based on direct LC3/GABARAP binding" instead of "autophagy inhibitors/enhancers" in general. This statement has been corrected in the text of the main manuscript.

Line 189 states 'To investigate covalent interactions of compounds 1 and 4 with LC3/GABARAP as recently reported for 1 with the E3 ligase DCAF11, we also studied the interaction of compound 1 and 4 with GABARAPL2 using ESI mass spectrometry, where no covalent modification was detected in vitro.' – reference to the data missing.

Response: We apologise for this omission. We have added a corresponding figure with the data (Supplementary Figure 7) and referenced it in the text.

Line 196 states 'For proteome-wide screening, we modified 1005 (4) and 8F20 (3) with PEG-based linkers that can be immobilized on sepharose beads to generate an affinity matrix for target pull-down, resulting in compounds 18 and 19. As expected, dose-dependent competition assays showed that the KIF11 inhibitor 8F20/Ispinesib (3) selectively bound to KIF11 in HEK293T lysates (EC50 of 290 nM). No

additional targets were detected for 19 confirming excellent selectivity of this inhibitor for KIF11 (Fig. 3d).’ Proteomics data is not shown anywhere in the paper. Fig 3d only shows competition data for compound 18 interacting with KIF11; how can it be stated that ‘No additional targets were detected’ if only one target has been monitored?

The complete proteomics data is now accessible via the MassIVE database (Accession code: MSV000093528, Reviewer password: Schwalm-Rogov). Figure 3d shows the only one robustly identified dose-dependent competition curve from both experiments. This indicates that KIF11 is a specific binder of 19. The reported targets of 18 and 19, GABARAP and LC3 (MAP1LC3), were not enriched in the pull-down proteomics readout. This argues for a low binding affinity of the probes to the reported targets in this highly sensitive MS-based readout. Based on the reviewer's comments, we therefore rephrased the section describing these data:

Figure legend 3D:

Chemoproteomic competition assays for target deconvolution of 8F20 (**3**) and 10O5 (**4**). Affinity matrices for pulldown experiments were synthesised via amide coupling, generating (**18**) and (**19**) attached to (NHS-activated) Sepharose beads. Competition experiments were performed with free compound **3** and PEG-linked compound **4** at nine concentrations and residual binding was calculated relative to a DMSO control. Of the over 4000 proteins identified, only KIF11 showed robust dose-dependent binding to **19** (EC₅₀ = 290 nM). Both the **18**- and **19**-based affinity matrix assays did not enrich for the reported targets LC3/GABARAP in HEK293T cell lysate.

Main text:

The only Atg8 homolog for which we detected a weak interaction, GABARAPL2, was not pulled down by **18** up to a compound concentration of 30 μM, suggesting that the interaction with this Atg8 homolog was also weak in the cellular context. [...]

‘Druggable’ is frequently used instead of ‘ligandable’, for example, line 248 ‘Our screen confirmed that HP2 is the most druggable binding site on 248 LC3/GABARAP proteins surface, accommodating 10 from 21 identified fragments’. This is not true as the fragments have not been confirmed to bind by orthogonal techniques, and binding does not mean that the site is druggable (functional).

Response: We agree with this comment and have changed the text as suggested. We avoid the term druggability when referring to the fragments and use 'ligandability' instead.

Figure legends should be more specific and contain more information. For example, all FP data is described as ‘Fluorescence polarization assay curves’, but it appears that these are displacement curves so the probe should be stated each time.

Response: Thanks a lot for pointing this out. We changed the figure legends accordingly.

Has the data been corrected for probe background fluorescence?

Response: All depicted FP assay data were normalized and contain a correction for the background fluorescence.

Fig 4B states ‘Structural representation and corresponding ITC data for the two screening hits (compounds 20 and 21)’, what proteins were used here? Fig 4D does not state what each of the different colored spectra correspond to.

Response: We provide now information on the protein (LC3A) and color code to the figure captions (1:0.125 – orange, 1:0.25 – yellow, 1:0.5 – green, 1:1 – cyan, 1:2 – blue and 1:4 - magenta).

Insufficient data are provided on the virtual screen – where were the compounds docked on the proteins? Does the NMR data shown in Fig 4D, E agree with the initial docking site?

Response: We docked to the well-characterised LDS site that has already been targeted with novobiocin (PDB ID 6TBE). At the request of reviewer 1, we have included the docking poses and the binding site description in RR_Figure 1 and RR_Figure 4, respectively. Indeed, the binding site indicated in Figure 4E was consistent with the binding site used for docking. We have now included panel F in Figure 4, where we have highlighted the residues that showed significant chemical shift perturbations in our NMR titration. The two carboxylic acid groups present in both screening hits shared a similar binding mode in which this moiety interacted tightly with an arginine and a lysine residue. This binding mode explained the loss of affinity after substitution of the acid group.

Some experimental details are not clear: Chemistry experimental methods are given for compounds 19, 17, 18, 16 (in a random order), but schemes are missing, and it is not stated where the molecules used in this paper came from.

Response: We have described the compounds in the order in which they are mentioned in the manuscript. As the compounds used to generate the affinity matrix were precursors of the previously mentioned tracers, these compounds have a higher number. We have updated the Methods section by adding schemes and corresponding compounds in the text.

Experimental method ‘Affinity determination using spectral shift mode on Dianthus’, it is not clear what data/assay this refers to, is it MST?

Response: All experiments were done with the spectral shift method (PMID: 35171002) using fluorescently labelled protein(s). We clarified this in the relevant section of the Materials and Methods.

‘Fluorescence polarization assay (FP assay): For the complementation assay’ – what is the complementation assay, should it be displacement assay?

Response: Indeed, we performed displacement assays. Thank you for spotting this error.

‘Covalent compound screening: For screening of covalent compounds, 46 compounds were tested against all LC3/GABARAPs, purified as described above’ – what are these 46 compounds? Fewer compounds were discussed in the text. Does ‘purified as described above’ refer to the compounds or the proteins?

Response: We initially screened 46 analogues of published covalent LC3 ligands. However, we concluded that ligands that did not interact with LC3 were of limited use and now only include the validation data for compounds 5-7 and additional data for compound 1. We have removed the statement "purified as described above", which referred to the protein used, as the purification has been described earlier in the paper.

Crystallization – What protein construct was used?

Response: We used the protein expression construct LC3B1-120 in the T7 vector pNIC28-Bsa4. This protein has been successfully crystallized in our group previously. Information about the construct has been added to the method section.

Fragment screening and structure solution – what concentrations were the compounds soaked at? What cryo protectant was used?

Response: For fragment screening, the DSI compound library had a stock concentration of 500 mM in DMSO. For the first round of data collection, the crystals were soaked at 20% of the starting concentration (100 mM). Under these conditions, the high DMSO concentration (20%) damaged some crystals, and a second screening campaign was run at 50 mM fragment

concentration. No additional cryoprotectant was used. We have now included these experimental details in the Materials and Methods section.

Minor points

FP graphs should use a shorter y axis scale (e.g. 0 - 120mP) to make the data easier to see.

Response: We adjusted all graphs accordingly using the 0-120 mP scale.

Fig2D the yellow and green spectra are very similar in color; it would be better to use a different color for one of them.

Response: We now changed the colour to a brighter yellow to make the differences clearer.

Compound 16 and 17 structures are buried in the supplementary and difficult to find, they should be referenced more clearly when mentioned in the text.

Response: As compound 16 and compound 17 were only briefly discussed, we decided to publish the structures of these compounds in the SI only.

Reviewer #4:

In this manuscript, Schwalm et al. focus on LC3/GABARAP-targeting ligands, characterising previously developed molecules and identifying novel chemical scaffolds. Protein recruitment to LC3/GABARAP through bifunctional small molecules, named autophagosome tethering compounds (ATTECs), has been previously explored as one of the different approaches to induce targeted protein degradation. The authors set out to validate existing LC3/GABARAP ligands used in ATTECs design, testing target engagement with different biophysical methods. The results reveal weak binding (in some cases no binding) of existing ATTEC ligands to LC3/GABARAP, highlighting the need to further characterise the cellular mechanism of action of ATTECs. In line with this observation, a recent study (Xue et al) shows that arylidene-indolinones, previously reported as LC3B binders, mediate protein degradation via DCAF11. In this context, it is certainly essential to perform a thorough characterisation of widely used chemical probes. However, in some parts of the manuscript, the authors seem to undermine any measured binding with recombinant LC3/GABARAP as weak or insignificant, although it is not yet clear what binding affinities are necessary to induce autophagy via ATTECs.

Before publication, the authors should address the following points:

Some examples of paragraphs where the authors should consider rephrasing statements dismissing observed binding: Lines 136-139, referring to Fig 2c and supplementary figure 2D;

Response: We have reworded some of the statements. The reviewer pointed out that the affinity required for ATTEC development is not known. This is certainly true, but if we do not see an interaction up to 100 μ M against the designated targets, affinities in the mM range would certainly not justify the claim of an LC3-dependent mechanism. We also believe that it is not clear that an autophagy-based degradation mechanism would be catalytic rather than stoichiometric, as the cargo is encapsulated for degradation and new cargo cannot be recruited in later autophagy cycles. However, these experiments would have to wait for confirmed and probably more potent Atg8 ligands.

Lines 144-145: "none of the compounds 1-4 resulted in significant CSP even at higher compound concentrations in agreement with our FP binding data." Supplementary figure 3 shows some shifts for compound (4). Also, please improve the layout/quality of this figure, as well as supplementary Fig. 5, as it is difficult to look at the NMR spectra.

Response: Indeed, titration of compound 4 into LC3B induces some CSP values for LC3B resonances, but these are significantly smaller compared to those for novobiocin (K_D to LC3B is ~ 10 μM), indicating a weak interaction. We have amended the section describing these data to reflect this. We have also revised the supplementary NMR figures (3, 5 and 10), which we hope will improve the presentation of the data.

oLines 169-178: despite observing micromolar binding of (4) with GABARAP family members, the authors define it again as not significant.

Response: As discussed above, titration of 1005 (4) into GABARAPL2 induced prominent CSP for several GABARAPL2 resonances, but the affinities were too weak to be determined by NMR or other biophysical methods. We therefore defined interactions in the >50 μM range as weak and, given the nature of the ligands, unlikely to be selective.

oLines 186-187: "raising the possibility of further off-targets within the proteome, based on the reactivity of these compounds". The authors should not make this claim without supporting data.

Response: We have reworded this section. The interactions of compound 5-7 are not selective within the LC3/GABARAP subfamily, probably due to the high reactivity of this electrophile. We have not assessed proteome-wide selectivity and describe this interaction therefore as unselective inhibition of human Atg8 family proteins.

MST experiments: the compound labelling in supplementary figure 4a should be corrected. Did the authors test (12) or (13) or any of the peptides using MST? do they show binding?

We have corrected the labelling of all compounds in Supplementary Figure 4a and included positive controls. RR_Figure 7 shows an example of a TRIC (temperature related intensity change) experiment (analogue of MST) showing the binding of LC3A protein to the canonical p62 LIR peptide resulting in a K_D of 2 μM , in agreement with literature values.

Ligand	Highest Ligand Conc. [mM]	Buffer	Target Conc. [nM]	K_D [mM]	ΔF_{norm} [%]	S/N Ratio	Saturation
p62-LIR	0.051	10 mM Tris 0.05% Tween	5	0.00184	-9.4	9.4	96.5 %

RR_figure 7: Binding data of LC3A:p62 LIR interaction using the TRIC technology. The table shown in the lower panel provides all experimental parameters as well as determined K_D values.

Lines 155-157: “Novobiocin (12) revealed binding with a K_D (6.7 μ M for LC3A), while titrations with AN2 (2) and 8F20 (3) did not yield significant binding heats”. These ITC experiments suggest no binding or weaker binding (not measurable at the concentrations tested). Dong et al. were able to measure binding of (3) using ITC (K_D ~12 μ M) at higher protein and compound concentrations. Did the authors try performing the ITC titration in those conditions?

Response: In fact, Dong et al. included ITC data showing apparent binding. However, the authors used 1.5 mM protein concentration. We have not been able to achieve such a high protein concentration. In addition, the compound has limited solubility, making it unlikely, in our opinion, that the heat effects are due to binding heats. We speculate that the high protein concentrations resulted in significant dilution heats which were not corrected in the blank titrations. However, as we have not been able to reproduce these data, we can only speculate about the nature of the published titration data.

Regarding compounds (4) and (1): are they tested as E/Z mixture? was the mixture purified in previous papers? is one of the two isomers more active? Tracer (16) is instead represented as pure isomer in Supplementary figure 4C. If the (4) E/Z isomers bind differently, this will result in lower effective concentration of the ‘active’ isomer and thus affect its ability to displace the tracer. Additionally, in comparison to tracer (17), (16) contains an α,β -unsaturated carbonyl portion next to the Py-BODIPY moiety, is that a drawing mistake?

Response: We tried to purify the isomers as we suspected that a single isomer was the active form. Unfortunately, even after isolating the individual isomers, the purified compound quickly isomerised. This is a known property of the oxindole scaffold. The HPLC trace is shown in RR_Figure 8. The single isomer shown in Figure 4C has now been corrected along with the α,β -unsaturated carbonyl moiety. We thank the reviewer for pointing out this error.

RR_figure 8: Analysis of the E/Z isomerization by HPLC.

• *Supplementary fig. 4D: did the authors try lower tween concentrations? or, in the absence of tween, displacement with one of their validated ligands (Novobiocin or peptides)?*

Response: Based on previous experiments we knew that the use of Tween was important in suppressing unspecific binding. However, the experiment (protein titration and displacement) was initially performed to reproduce the titration reported by Dong et al. in the absence of Tween. Under these conditions, we confirmed the tracer binding reported by these authors. However, we routinely perform displacement experiments to control for unspecific binding and found no displacement using the parent ligands of the tracers. To avoid unspecific binding, we used 0.05% Tween, which suppressed unspecific interactions and the apparent FP signal detected in Tween-free buffer. As we concluded that the compound-based tracers did not monitor specific bind to the protein, we did not perform further FP experiments with these tracers. Due to the lack of small molecule tracers, we subsequently established the p62 LIR-based fluorescence polarisation assay.

• *Lines 196-206: This section describing pull-down experiments requires some clarifications (also in the corresponding methods section) and additional controls. The authors should show a positive control (e.g. using a peptide or one of the better binders) to demonstrate that it is possible to measure enrichment of LC3/GABARAP with this setup. Also, what does it mean that there was no enrichment “using 18 (up to 30uM)” if the compound is conjugated to Sepharose beads?*

Response: We apologise for not describing the assay well enough. We performed this assay in a ligand displacement format, i.e. immobilised proteins binding to the biotin adducts were eluted with the original ligands. We agree that a positive control would be necessary to prove that the probes do not bind to LC3/GABARAP in the competition assay setup. However, the successful pulldown and competition of the reported target KIF11 confirmed that the assay performed well in principle by identifying the known target of this inhibitor. We have attempted to clarify the description of the pull-down assay by amending the figure legend and the section describing these data in the manuscript.

• *It would be helpful to include a description of the NanoBRET assay in the main text. The authors say that there is a moderate BRET increase but no displacement. It would be helpful to further confirm this by testing displacement with one of the stronger binders.*

Response: We introduced the assay in the revised version and modified this section: BRET is a proximity assay between a fluorescent donor (in this case full length LC3A and LC3B) and a fluorescent acceptor (the dye adducts of 3 and 4). BRET occurs when the donor and acceptor are in proximity (<10 nm) and its fluorescence intensity is inversely proportional to the distance of the donor-acceptor pair. This technology has gained popularity as a live cell assay format for monitoring protein-protein interactions or target engagement with small molecules. As expected from our biochemical data, we observed no BRET signal using the 8F20 and 10O5 dye analogues (compounds 16 and 17) (Supplementary Figure 8a-c).

• *Supplementary fig. 10 - line 196: “Crystallized fragments bound to LC3A” contradicts the rest of the text, where it’s LC3B (e.g. page 39 line 738)*

Response: Thanks for spotting this error. We used LC3B for all crystallization studies and corrected this error.

• *Did the authors try to soak/co-crystallise any of the previously described ligands with LC3/GABARAP? Some of the compounds showing weak binding in this manuscript could potentially be useful starting points for optimisation, similarly to the newly identified fragments.*

Response: We used compounds 1, 4, 9, 11, LY22398, TH152 and TH200 in soaking experiments, but only obtained apo structures or weakly diffracting crystals. To support further studies on rational ligand design, we now include our docking model and highlighted residues with significant chemical shift perturbations in NMR (panel F in Figure 4). As discussed above,

the COOH group of our docking hits showed a similar binding mode, interacting with an arginine and a lysine residue.

Reviewer #5:

The manuscript by Schwalm et al. "Targeting LC3/GABARAP for degrader development and autophagy modulation" is an impressive dualistic exercise where on one hand the authors rigorously test published ATTECs (autophagosome tethering compounds) for binding to LC3/GABARAP family proteins, and on the other hand perform in silico docking and large scale crystallographic fragment screening to identify a group of compounds they validate as ligands for future ATTEC development. Perhaps surprisingly, and certainly disappointingly, the authors find that the published ATTECs do not bind at all (almost all tested compounds), or too weakly, to LC3/GABARAPs for a specific biological effect to be explained. This is a very important finding to be communicated pointing to a too poor characterization and binding validation effort done in the published work on ATTECs: This is clearly damaging to the field as such and cries for better evaluation of such papers before they are accepted for publication. In the second part of their paper the authors use in silico docking and large scale crystallographic fragment screening to identify a number of compounds that do bind to the LC3/GABARAPs and also identify two new binding surfaces. A conclusion is also that it is the hydrophobic pocket 2 (HP2) that is the most druggable in the LIR Docking Site (LDS) while HP1 interacting with the aromatic residue of the core LIR is much less accessible for the compounds.

Response: We thank the reviewer for this summary.

This is clearly an important paper with thoroughly performed experiments using an impressive array of assays to validate binding both in vitro and in cells in addition to structure explorations techniques like NMR and x-ray crystallography. The "negative" half of the paper, showing that published ATTEC compounds basically do not bind to LC3/GABARAPs, deserves to be communicated widely in a very visible journal. The "positive" half of the paper stops when it starts to become exciting. A number of candidate compounds are presented that can be used to develop new ATTECs are identified, but we do not see any of these taken to the next level and we are left wondering: Is ATTECs at all a viable approach? For the paper to be acceptable for publication in Nature Communications the authors need to go further with one or more of the compounds they have identified and test in a proof of principle setting if they can see targeted degradation using such a compound equipped with a target recognition part connected with a linker to the LC3/GABARAP interacting compound.

Response: We agree with the reviewer that validation with LC3/GABARAP is still lacking. However, we wanted to avoid using non-optimal compounds to develop an LC3/GABARAP-based degrader. The *in silico* hits are probably the most potent ligands currently known, but the COOH group prevents efficient cell penetration. We have developed a series of compounds based on the novobiocin hit with the central coumarin scaffold (PMID: 33769048), but even these ligands are still weak (>20 μM), suggesting that high concentrations will be required to achieve target engagement in cells. The use of high concentration ligands is the main reason why many chemical biology studies report inaccurate phenotypic responses. It is therefore our aim to develop the first ligands that meet the criteria of chemical probes (with on-target potencies better than 100 nM). This will require the synthesis and evaluation of hundreds of compounds. As the field of degraders is developing rapidly, we believe it is important to share negative data to prevent research groups from developing ATTECs based on current ligands. To date, 29 manuscripts have been published using ATTEC ligands. We therefore believe that the publication of negative data is important for the advancement of the field.

Reviewer #6:

In the paper titled 'Targeting LC3/GABARAP for degrader development and autophagy modulation' by Martin P. Schwalm et al. The authors shed light on the principles of autophagosome tethering compounds (ATTECs) based on the target recruitment to LC3/GABARAP. The researchers assessed

the published ATTEC ligands using biophysical techniques including FP, ITC, and NMR. The authors note their surprise that the literature identified ligands do in fact not bind to their intended target LC3. They then go forth to use their world leading expertise to identify a number of small molecule ligands through virtual screening and high-throughput fragment via X-ray crystallography. Presenting a diverse comprehensively validated series of ligands for future ATTEC development. The present study is well designed, and the authors have given a comprehensive insight into the binding mechanism of ATTEC compounds. The result of this study is of great use to the drug discovery community not only from the ligands for future ATTEC development but also from reagents produced in the LC3/GABARAP systems and experimental methods. I would suggest this study for publication with some minor alterations and points of discussion.

Response: We thank the reviewer for encouraging us to publish this work.

Minor Points:

1) When debunking a previously reported study it is difficult to be completely resolute in biophysical data, as a number of factors, such as PTM and binding partners may play a roll in the published ATTEC ligands that are hard to replicate in a reconstituted cell-free/cellular manner. Although the studies data is strong and very compelling, I would suggest a sentence to this extent.

Response: We agree with the reviewer that PTMs or other components may increase ligand affinity. We have therefore included MS pull-down experiments and cellular BRET-based assays in our ligand characterisation. As suggested, we have added a sentence in the Discussion section highlighting the modulation of target binding in the context of the cellular environment.

2) In a pleasant way, it is hard to read the publication and not ask “what’s next?” I assume this is out of the scope of this study and would be a major addition to the publication.

Response: This is quite right. We have also discussed the issue of developing ATTECs in our response to reviewer 5. We strongly believe that a potent and well validated LC3(GABARAP) ligand would be required for further studies and these activities are ongoing in our laboratory.

i) Building on the positive outcome from the virtual screen could the authors possibly resolve a structure of TH152 bound to LC3B

Response: Following the identification of TH152 as a potential ligand, we immediately initiated crystallisation approaches on LC3B, LC3A and GABARAPL2 proteins. However, co-crystallisation did not yield protein crystals and soaking experiments resulted in poorly diffracting crystals. However, we have now included the docking model confirmed by our NMR data in the revised manuscript.

ii) Could the assays built within the invalidation of the published ATTEC ligands be used against the crystallographic output, even if they are likely to be weak binders?

Response: We used the developed FP assay as well as SPR to determine the binding constants of the identified fragments and their closest homologs. However, the ligands identified in the crystallographic fragment screening were still too weak for accurate determination of K_D s. We estimate the affinities to be in the high micromolar or even millimolar range. The data are summarised in RR_Figure 9.

All compounds were tested in the FP displacement assay, where fluorescence-labelled p62 LIR peptide was displaced from LC3B and GABARAPL2 (RR_Figure 9). The data was normalised to p62 LIR displacement at 100 μ M to yield a relative readout. Hence, compounds below the threshold set by the p62 LIR could be potential binders.

RR_Figure 9: FP values normalised to p62 LIR binding at 100 μM concentration. The actual compound concentration (500 or 1000 μM) is indicated on each plot. The FP values above the normalisation (red line) did not provide data that allowed the K_D to be determined.

As follow up, the compounds were subjected to SPR measurements and screened in a one-shot format at 250 μM . Only 2 compounds (F15 and F17) gave data that allowed us to estimate K_D values in the high micromolar or low mM K_D range.

We thank the editor and the reviewer for their insightful comments and constructive criticism. We have now revised the manuscript to address all of the issues raised. A detailed description of the changes we have made is provided below.

Reviewer #1:

Some of the issues raised previously have been addressed, but there are still some outstanding issues regarding the discussion of the in-silico methods used, which affect the quality of the results and their reproducibility. While it is well-received that the Authors provided clarifications in the document for the reviewers, most (if not all of them) are essential for the reader, and they should be in the manuscript.

For example, the manuscript still lacks details about the following:

- chemical properties of the library used*
- details of the similarity search (Tanimoto similarity of "what versus what"? what cutoff?)*
- details of the site(s) used for the molecular dockings*
- identity of the residues included*
- search parameters used for the virtual screening (if "default" were used, it should be explicitly stated)*
- detailed comparison between residues identified in the dockings versus the experimental validation*

The recommendation to provide more critical discussion of the hits found and how these can be used for driving further research in this area (an issue that apparently other reviewers raised as well).

Response: Thank you for the valuable feedback. Since the algorithms used in the software are non-deterministic, we were initially unsure of their relevance. However, we agree that the definitions of the binding sites and the corresponding constraints on the search space could be useful to the readers. We have therefore included a detailed explanation of these aspects in the method section of the manuscript. We also wrote a discussion on the properties of the hits identified by *in silico* screening as well as the fragments identified by protein crystallography and we rearrange the supplemental figure describing the site used for docking which details the residues included (Supplementary Fig. S11). The analysis of the chemical properties of the library used are now presented in a new Supplementary Fig. S12. As the number of molecules similar to the query implicitly determines the similarity threshold, no hard limit was defined.

Reviewer #2:

The revised manuscript satisfies the concerns I raised with the original version. I note the following grammatical corrections in the discussion section:

line 292 - 'events indeed are' (or 'event indeed is')

line 324 - insert comma after 'changes' (I think; this sentence is awkward)

Response: We would like to thank this reviewer for the detailed work and for spotting many typos and errors. All of the above suggested corrections have been included in the revised version. The entire paragraph starting at line 324 has been revised:

"Autophagosome biogenesis is a complex process involving many proteins and cellular factors. One of the key players in this process are the LC3/GABARAP proteins, which maintain a finely tuned equilibrium between their intact and lipidated forms in the cell. Perturbation of this balance by high-affinity LC3/GABARAP ligands can lead to a decrease in autophagy flux. Therefore, further research is required to gain a deeper understanding of the specific conformational differences and dynamics of these proteins in the presence of such ligands in the cell."

Reviewer #3:

This manuscript details the characterization of ligands targeting LC3/GABARAP proteins. This includes a thorough assessment of literature ligands targeting LC3/GABARAP that have been used in ATTECs, autophagosome targeting compounds, identifying key concerns with literature reference compounds and their proposed mechanism of action, due to their inability to engage LC3/GABARAP proteins in vitro. The authors then provide additional chemical matter, identified through in silico and fragment screening, which bind to multiple sites on LC3 proteins and may provide new starting points for the development of ATTECs.

Whilst a thorough characterisation of literature ATTEC ligands is provided and is necessary for the community, there is concern regarding the inclusion of two separate 'stories' – ATTEC characterisation and novel screening strategies against LC3 – within the same manuscript. Without confirmation of ATTEC mechanism of action, and the validation of novel ligand activity, the ATTEC characterisation alone should be published to avoid muddying the literature in an already problematic space.

Response: We thank the reviewers for these comments. However, we consider the assessment of LC3/GABARAP ligandability an important aspect of the manuscript. It was not our intention to only re-evaluate current ligands and ATTECs but to provide a tool box of assay and chemical starting points for the development of such tools. At this point we would like to stress that the experimental setup leading to the acquisition of close to 1000 crystal structures for routine structure determination of fragments is not trivial and a considerable effort. The identified fragments map the chemical space for ligand development and provide insights into the ligandability of potential binding pockets. It was for instance surprising that the largest binding pocket in the LIR binding site that accommodates the tryptophane sidechain is not addressable despite the large number of indole moieties in the fragment set. This information is highly valuable for drug design and the assessment of interaction hotspots on the LC3/GABARAP surface.

Major points: Without confirmation that the ATTEC proposed mechanism can be achieved in cells, the fragments and in silico compounds are of limited use, as per the statement in the manuscript that LC3 binders alone are insufficient to trigger autophagy. As these ligands are not potent enough to be used in the development of ATTECs, the autophagy-induced degradation mechanism could be confirmed using a HaloTag-LC3 construct and a HaloTag-ligand to a known protein of interest or other tag.

Response: We tested diverse principles to artificially tether proteins of interest to LC3 including using a GFP-LC3 fusion and GFP-targeting DARPINS fused to either DARPINS or nanobodies targeting a protein of interest. However, such systems potentially fail due to a low basal autophagy flux caused by the fusion partners triggering inhibition of LC3/GABARAP activation. Thus, it is likely that overexpression of LC3 as a fusion protein changes the natural turnover leading to induced proximity of the protein of interest to LC3 but no membrane anchorage of LC3 to the phagophore. Because of this, we focused here on the discovery of small molecule ligands.

Readers may incorrectly use these compounds, particularly 20 and 21, without further development or characterisation. We would recommend the inclusion of an example of a developed molecule from a fragment that exhibits measurable potency following the proposed fragment growing/linking strategy to highlight the plausibility of this approach, and to emphasise the need for compound optimisation prior to ATTEC generation. Without this, this part of manuscript should be removed for publication at a later date. At the very least, it needs to be included in text that K_Ds could not be determined for the fragments due to weak binding, as stated in the response to reviewers. If the screening data is still included, some closer views of hotspots (eg the HP2 site) for preliminary SAR would also be useful for the reader.

Response: We revised the paper stressing that the current ligands (fragments and in silico hits) are not suitable as inhibitors in cellular assays nor for ATTEC development. We hope that we highlight these limitations sufficiently in the current version. We also discussed that charged moieties such as the carboxylic acid groups in the in silico hits most likely prevent efficient cell penetration

Minor points: Was ITC attempted to identify the K_D of the fragments?

Response: The fragments were only assessed by FP and SPR which indicated affinities in the mid to high micromolar K_D region. ITC experiments are therefore not feasible for measuring fragment affinities.

Page 6 – the p62:LC3 interaction is clearly important, resulting in the use of the p62 LIR peptide, but the context of this interaction within autophagy/the cell is not clearly outlined in the introduction

Response: Indeed, p62 LIR is the prototypical LIR motif, its interaction with LC3B drove the development of the LIR concept (Ichimura et al, J Biol Chem. 2008; doi:10.1074/jbc.M802182200) and it is therefore widely known in the field of autophagy. Therefore, we decided not to focus on definitions of p62 LIR, but to cite some of the numerous research and review articles considering it. Nevertheless, we extend the Introduction (page 3) to refer implicitly to a review that describes in details the p62 LIR motif and its interactions with LC3/GABARAP proteins.

Page 9 – the point regarding KIF11 was a little confusing. The fact that compound 3 is an inhibitor of KIF11 had not been previously introduced; recommend providing additional context to this, perhaps when 3 is introduced to the manuscript, for reader clarity. Furthermore, is there any evidence eg in databases that show it would be expected to be able to identify LC3 or GABARAP proteins by proteomics (eg tryptic peptides, known expression levels in cell line)?

Response: We introduced now the KIF11 inhibitor in the introduction on page 4 and added a reference.

Page 14 line 324 – “by affine LC3/GABARAPs ligands” – the word affine is misused in this sentence.

Response: We thank the reviewer for highlighting this. We have restructured this paragraph to provide more clear statement.

Figure 2e – the light green text is challenging to read, it is suggested to make this a little darker

Response: We have changed the green colour in all the relevant figures: Fig. 2e, Fig. 3b and Supplementary Fig. 10g to make the text, lines and bars for $CSP > 1xSD < 2xSD$ more visible.

Figure 4b – could the authors comment on the relative enthalpy change for the two compounds; the maximum for TH152 is significantly smaller than for LY223962 (and novobiocin, in supplementary 4b) and more comparative to the inactive compounds shown in the supplementary? Is this due to compound solubility?

Response: Indeed, the enthalpy change upon interaction between protein and ligand is mostly proportional to the strength of non-covalent interactions (van der Waals, hydrogen bonds, electrostatics) relative to those existing with the solvent. The enthalpies we observed in our ITC experiments for novobiocin as control (-4 kcal/mol), LY223982 (-7 kcal/mol) and TH152 (-5 kcal/mol), indicate the contribution of polar interactions such as intramolecular hydrogen bonds and charge interaction. Compounds were chemically stable. Novobiocin was measured in reverse titration (protein titrated into ligand) due to solubility limits.

Figure 4f – the pink amino acid labelling is challenging to see against the background of the blue/grey. Recommend making the background lighter/more transparent and/or changing the pink colour of the ligand.

Response: We have adjusted Fig. 4f and the related Supplementary Fig. 11 to provide readers a better view on the TH152 docking pose and the comparison of the predicted binding mode with the NMR results.

There seems to be no compound 14 or 15, have these numbers been missed in the manuscript, or are some additional compounds not referenced in text?

Response: Compounds 14 and 15 were intermediates in the chemical synthesis of compounds 16 and 18. We therefore mention compounds 14 in 15 only in the methods section of the manuscript.

Reviewer #4:

The authors have addressed most of the points that were raised. However, one major concern pointed out in response to reviewer #3, is the number of independent experiments performed (e.g. ITCs with $n=1$).

It is unusual repeating ITC experiments due to the high protein requirements of this technology. As the K_D/K_A determination is based on 30 data points, the errors are typically estimated based on the error of the fit. However, to provide evidence of the reproducibility of the ITC data we repeated the titration of LY223982 3 times. We obtained comparable thermodynamic results for all 3 measurements:

Replicate, NR	K_D [M]	Stoichiometry, n	Enthalpy, [kCal/mol]	Entropy, [Cal/(mol*K)]
1	1,12E-05	1,038	-11,64	-16,4
2	1,14E-05	1,021	-9,814	-10,29
3	1,00E-05	1,011	-10,66	-12,88

We would like to highlight that we validated all binding data in diverse orthogonal assays including FP and NMR confirming the reported binding affinities of all ligands

In addition, a few minor points are listed below:

The chemical structure with the α,β -unsaturated carbonyl moiety (Py-BODIPY) in Supplementary figure 4C has not been corrected.

Response: We have edited Supplementary Fig. 4c which displays now the correct chemical structure of tracer **16**.

Regarding supplementary figure 4a - the authors include the positive control for the measurements (p62-LIR) only in the data for reviewers. It would be helpful to include this in the published manuscript. Was this performed in the same conditions as supplementary figure 4a? Following the comment and response to reviewer #3, does $n=2$ here refer to independent experiments or technical replicates?

Response: We would like to keep the control measurements with p62 LIR containing peptide in the Rebuttal Letter only – this experiment was done at exactly the same conditions but with another technology (TRIC), which will lead to excessive details and will shift the focus from the real results. Additionally, all data within Rebuttal Letter will be available for the readers.

Also, in the supplementary figure 4 legend, “lover table” is maybe “lower table”?

Response: Thanks for spotting this error. This has been corrected.

In the discussion section, given that some weak binding has been measured for some compounds, I would suggest modifying the sentence “none of the ligands used for the development of ATTECs interacted with LC3/GABARAP”, specifying that this refers to profiles of selectivity/affinity expected for chemical probes.

Response: We have modified the Discussion section inserting the term “measurable affinity” considering the limitations of the used technologies: “Surprisingly none of the ligands used for the development of ATTECs showed measurable affinity for LC3/GABARAP using a diversity of biophysical methods, suggesting that reported LC3 based degraders cause degradation not due to a LC3/GABARAP mediated mechanism.

Reviewer #5:

The authors have answered the single critical comment I had in a satisfactory manner.

Response: We thank this reviewer for supporting this publication and detailed comments on the originally submitted manuscript.

Reviewer #6:

Reviewing the amendments to the paper titled ‘Targeting LC3/GABARAP for degrader development and autophagy modulation’ by Martin P. Schwalm et al. As before the study is well designed, and the authors have responded to my previous minor suggestions adequately. The results of this study are of great use to the drug discovery community not only for future ATTEC development but also from reagents produced in the LC3/GABARAP systems and experimental methods.

I would still be keen to see some other experiments/points of note, more out of interest than anything.

- 1. Although TH152 struggles to be cell permeable potentially due to the two carboxylic acids, or likely high protein binding are you able to show target engagement with the NanoBRET with lysed cells? The idea is you would have an additional measure, although likely/expected to be weaker due to the likely high protein binding of TH152?*

Response: We did not success designing a tracer for cellular nanoBRET assays. However, we don't think that the two identified changed molecules are cell penetrant. We hope that our invitro data that covers ITC, FP, NMR and SPR based assays would sufficiently confirm the binding activity of these ligands.

- 2. Or further this point of target engagement looking for some extra evidence, could you add a - PEG-linker to TH152 and look at this in your affinity matrix pull down experiments?*

Response: We have not been concerned about the selectivity of the current hit compound. The parent compound has been developed targeting a GPCR. It is therefore likely that also its derivatives will have some additional activities. We hope that the SAR provided on this series will facilitate further development of this scaffold into a potent chemical probe.

Letter to reviewer commenting on issues raised on the first submission of the paper

We would like to thank the reviewers for their insightful comments and constructive criticism. We have now revised the manuscript considering all issues that have been raised. Below we provide a point-by-point response to each comment as well as an outline of change we made in the revised manuscript

Reviewer #1:

The Authors explore small molecule approaches toward the identification of autophagosome tethering compounds (ATTECs), a fairly novel and challenging field of research for potential drug development. Together with performing both in silico and x-ray based screenings, the Authors perform a number of biophysical assays to characterize previously published compounds, reporting that for the majority of them it was not possible to confirm the proposed mechanism of action. I found the data produced to validate the previously reported binders to be very compelling and useful. That is very interesting data and surely helpful to other researchers in the field, besides providing a broader perspective on drug discovery campaigns in general. The overall quality of the manuscript and the data provided is sufficient for being published, but there are some issues that I feel should be addressed prior to that.

Response: We thank the reviewer for this summary and the encouraging comments.

My concern is regarding the structure-based work, where more details would have been very welcome. Also, there is some degree of disconnection between the different approaches pursued under this effort. The initial in-silico screening was instrumental to the identification of the first round of novel hits, generating what looks like an innovative scaffold that identified at least one important pharmacophoric feature (the presence of the two carboxylic groups). The affinity of these hits is also interesting because it is comparable to that of known and well-established hits.

However, it is not clear how the two hits were identified from the 173 compounds. Where these the only active molecules identified or just the best/most promising (and if so, according to which criterion)? Either way this information should be reported. Also, it would be great if the structures of the other 171 virtual hits would be made available, since that would be very informative to provide context to the results obtained. Without any details about the content of the (proprietary?) library, it is very difficult to evaluate the data generated from the screening. It would be very helpful to have more information about the content of such library, even if just by providing statistics about physicochemical properties of the molecules contained in it (MW ranges, HB count, clogP, etc.). This is relevant also because in the methods it is reported that a relatively high MW cutoff was used to filter out compounds. How that affects the outcome of the screenings is difficult to estimate without knowing the overall properties of the libraries, in light of the issues associated with all docking scoring functions.

Response: We thank the reviewer for taking a close look at the virtual screening approach. To answer this question, we re-evaluated the virtual screening hits and updated the hits in Figure 4. The 271 most promising compounds (called virtual screening hits in the manuscript) from the virtual screening were validated in vitro. Of these, TH152 and LY223982 showed significant displacement and were therefore selected for further validation of their interaction with LC3. The selection of these hits was based on the best scores in the respective programme (estimated affinity for SeeSAR and estimated free energy of binding for Autodock) and our selection cut-off was also based on the screening capacity in the laboratory.

The distribution of physicochemical properties of the screened compound library is shown in Reviewer Response (RR) RR_Figure 1. A 750 Da cut-off was used to filter out large library compounds, as such molecules often give rise to high estimated free energies in Autodock, probably due to large entropic contributions and the large number of intramolecular interactions.

RR_Figure 1: Physicochemical properties of our in-house library computed with rdkit

Also, TH152 and LY223982 were experimentally validated using ¹⁵N NMR, which provided a compelling support to the in-silico hypothesis, "confirming molecular modelling of these interactions", as stated in the manuscript.

RR_Figure 2: Docking poses of TH152 to LC3A (PDB ID 6TB E)

As an example, the obtained docking poses of TH152 are depicted in RR_figure 2. Additionally, we used these poses to compare them with the interaction regions determined through NMR and were able to show overlap between the experimental and in silico methods. The results of this comparison are now depicted in Figure 4 F, the new SI Figure 11 and are now discussed in the main text of the paper.

Being this the major innovation of the work, I think it should be properly addressed, because it would increase the impact and the value of the manuscript.

Finally, the details about the setup of the virtual screenings are not sufficient. The Authors should add more information about the experimental setup such as size and location of the binding sites used for docking. Values of docking scores, (or their ranges, at least?) should be provided, too. Was the scoring scheme a consensus scoring (SeeSAR +AUTODOCK) or sequential scoring (first SeeSAR then AUTODOCK)? These details are essential to support at least a minimum degree of reproducibility. In the methods it is listed that 271 compounds were selected, which might be a typo?

Response: The 271 compounds were selected on the basis of their *in silico* score in one of the two docking programmes used. Thus, only these 271 compounds were evaluated in the FP experiments. Furthermore, in a second screening experiment, we selected 104 compounds that showed structural similarity to TH152 and LY223982, as these compound hits were part of a SAR series available in the laboratory. RR_Figure 3 shows the distribution of docking scores.

RR_Figure 3: Distribution of docking scores (estimated free Energy calculated by AutoDockGPU and estimated affinity by SeeSAR). The compounds used for *in vitro* validation after filtering are highlighted in purple, the remaining compounds are depicted in green.

RR Figure 4: Structures and binding sites (HP1+HP2) used in virtual screening (left: LC3A, PDB ID 6TBE, right: GABARAP, PDB ID 4XC2)

Overall, while the results are compelling, a more critical and in-deep analysis of them would have been welcome, especially in light of the structural work performed. There is a perceived unbalance toward the validation of the known binders/inhibitors at the expenses of the innovation. The amount of critical discussion of the new results is relatively scarce, which is unexpected given that there are positive results to report.

Response: We have expanded our Discussion section to improve the balance and provide a broader overview of the prospects for rational design of ATTECs. We believe that the number of chemically diverse hits identified in the high throughput fragment screen provides an indication of the ligandability of the binding site. However, the fragment interactions were too weak to allow an accurate assessment of their affinities. The HP1 pocket appeared to be most accessible to many structurally diverse fragments, with only one fragment identified for the HP1 pocket that bound to the edge of the pocket. This was certainly surprising and highlights a potential problem with the accessibility of this binding site (as suggested in J Cell Biochem. 2023 Feb 13. doi: 10.1002/jcb.30380. We have expanded the discussion of the fragment data in the revised version of the paper.

Reviewer #2:

The paper titled 'Targeting LC3/GABARAP for degrader development and autophagy modulation' by Schwalm, M, et al. uncovers issues in current LC3/GABARAP-targeting compounds and generates novel LC3 fragments for the development of future ATTEC-based therapeutics. Using a wide range of biophysical techniques, currently available ATTEC ligands are screened for binding and selectivity at canonical and non-canonical binding sites. Finding that available ligands display poor/no binding capabilities, the authors use virtual and experimental fragment screening to generate scaffolds for future ATTEC development. Overall, this work provides value to the ATTEC field by correcting the existing literature and reinforcing the potential for development of legitimate LC3 ligands.

Response: We thank the reviewer for these comments.

The principal weakness is the perfunctory presentation of the results and lack of cheminformatic analysis; as such, the manuscript serves as a multi-mode screening report that confirms a recent report from the Genentech group (ref 22) showing that Atg8 proteins are legitimate targets for small molecule ligand discovery. Follow-up studies will be required to establish the utility of identified hit compounds. Below are specific comments to improve the manuscript.

1. The paper pursues several goals. Breaking the results up into sections with subheadings (e.g. "Validation of known ATTEC ligands", "LC3 Virtual Fragment Screening", etc.) would improve clarity. Likewise, the 2nd paragraph of the introduction should be split into two or more paragraphs for readability.

Response: We agree with the suggestion of the reviewer and re-structured the result section as suggested. We also included the suggested subheadings and improved the Introduction.

2. The discussion should be amended with a cheminformatic SAR analysis of the fragments and compounds identified in this study, as well as a comparison with the results from Steffek et al (ref 22).

Response: We have now added a more comprehensive SAR analysis in the Discussion section, as well as a comparison with the fragments reported by Steffek et al.

3. English usage could be improved. For example, there are missing or misused articles (e.g. 'the') throughout the text.

4. Line 213: change "effecting" to "affecting"

5. Typo on line 285: ATTECT

Response: We thank the reviewer for highlighting these typos. They have been corrected as suggested by the reviewer.

Reviewer #3:

This paper reports the biophysical characterization of compounds which have previously been reported to bind LC3/GABARAP proteins, including some that have been used to generate autophagy tethering compounds (ATTECs). Fluorescence polarization and protein NMR indicate that many of these compounds do not interact with LC3/GABARAP in vitro, and the authors therefore cast doubt as to their suitability for use in ATTECs. They then report their own virtual screen and crystallography screen to identify ligands that bind to LC3/GABARAP.

The thorough biophysical characterization of reported LC3/GABARAP ligands is a useful addition to the field and highlights the need for caution when using published compounds. However, the new hits identified from the ligand screens reported here are not fully characterized, and no effort is made to show that these could be used for development of ATTECs. Indeed, the authors themselves note that 'the identified ligands might be characterized by poor cell penetration and might therefore require optimization for ATTEC development'. This limits the potential impact and also does not correlate with the title of the paper.

Response: The current paper was mainly focussed on the validation of the published ATTEC systems. These data have been published in high impact journals and as a new design strategy for degrader development it has received significant attention in the community with several follow-up papers that were however all from the same research team. We are aware that the data on ATTECs are "negative data" and as such less exciting. But we know that the ATTEC system has been unsuccessfully reproduced by many labs. We believe therefore that sharing these data will be important for the community of scientists working on degraders. A second aspect that we wanted to address was the druggability and the chemistry needed to for a rational ATTEC development. We therefore provided the fragment screening data, evaluation of other published LC3 ligands as well as a panel of assay system that can be used for LC3 ligand development. Considering the small size of the screened in silico library the hit finding

campaign was very successful. In fact, the two identified in silico hits are the most potent LC3 ligands known to date. The development of potent ligands for LC3 that would be suitable for ATTEC development will require the synthesis and evaluation of many hundred compounds (and as a consequence 2-3 years of experimental work). We therefore did not include LC3 ligand development into this paper as we wanted to share these data early. We also think that the large body of structural data on LC3 fragments will allow the chemical biology community to contribute to the ligand development process.

Key strengths

- *The field of induced proximity has rapidly expanded in recent years and novel molecules in this area are frequently reported, but sometimes without the necessary rigorous characterization. This paper addresses this by characterizing reported ligands that have been used to generate ATTECs.*
- *A variety of biophysical techniques are used to characterize previously published ligands, including fluorescence polarization, MST, NMR, mass spectrometry and ITC.*

Response: We thank the reviewer for this accurate summary and recognition of our efforts.

Key weaknesses

- *The title and abstract do not accurately represent the content of this paper. The title 'Targeting LC3/GABARAP for degrader development and autophagy modulation' implies that the paper will report molecules that do this, but in fact, the data indicate that existing molecules reported to target LC3/GABARAP may not act in this way, and the new ligands identified are not used in degrader development or shown to modulate autophagy. In addition, the abstract states 'Here, we present diverse comprehensively validated ligands for future ATTEC development', but the new ligands are not comprehensively validated, nor is there any evidence that they could be used for future ATTEC development. The crystallography hits are not confirmed with biophysical or biochemical data, and only two virtual screening hits are confirmed to bind by ITC and NMR, binding LC3/GABARAP weakly with K_d 1.9-10 μ M. It is not clear why the authors use several assays to investigate previously reported ligands, but do not follow the same level of characterization for their own hit compounds to fully validate them.*

Response: We agree with the reviewer that the title need to be changed to reflect better the presented data. We change therefore the title to: **Critical assessment of LC3/GABARAP ligands used for degrader development and ligandability of LC3/GABARAP binding pockets.** We also edited the abstract and deleted statements that current ligands can be used for ATTEC development. We tried to measure affinities of the crystallized fragments. However, the identified fragments are still too weak to measure accurate affinity data by SPR and other technologies. The identified in silico hits were however comprehensively characterized and considering all ligands published for LC3/GABARAP, they represent the most attractive starting points for development of potent ligands. As the intension of the ligands and fragments together with their binding modes was to serve as chemical starting points, we did not fully characterize them.

Biophysical characterization indicates that the reported compounds 1-4 do not significantly bind LC3/GABARAP in vitro, but no attempts are made to replicate the previously reported cellular activity of these compounds. It would be beneficial to know whether these compounds have any effect on autophagy in the authors' hands, and therefore could act by alternative mechanisms, or whether they are inactive.

Response: We were very concerned that the ligands used for ATTEC development might induce general toxicity in cell lines, as they are based on drugs used in oncology (such as the kinesin inhibitor 1005) and the isoxazole scaffold often used in the development of multi-targeted kinase inhibitors. However, we have now investigated compounds 1-4 in autophagy flux assays using the RPE-1 GFP-LC3B-RPF-LC3 Δ G reporter cell line (single clone) as reported in [PMID:37558952]. We found that 1005 and AN2 (RR_Figure 5) did not induce autophagy in this assay system, but AN1 and 8F20 showed significant autophagy activation

(RR_Figure 6). The authors of the first manuscript reporting these ligands [PMID: 3166698] did not report autophagy induction by 8F20 and AN1. However, due to the potent activity of 8F20 on KIF11, which will have profound effects on cell trafficking and growth, and DCAF11 (AN1, [PMID: 38036533]), we find it unlikely that the effects on autophagy are due to LC3/GABARAP modulation. Due to the lack of target (LC3/GABARAP) binding *in vitro* and the unclear mechanisms of autophagy induction, we only present these data here for the purposes of this review but have excluded the data in the manuscript.

RR_figure 5: Autophagy flux measurement for 10O5 and AN2. No induction of autophagy or effects (right panels, assessed by Incucyte (Sartorius) life cell imaging) on cell growth was observed.

RR_figure 6: Autophagy flux measurement for 8F20 and AN1 show changes in autophagy flux upon compound treatment. However, also profound effects on cell growth were observed using life cell imaging.

It is not clear how many biological replicates have been performed. For example, the figure legends state 'Data were collected in technical triplicates with error bars expressing the SD (n=3).' Does n=3 refer to the technical triplicates or does this mean biological replicates? At least two independent

experiments should be performed to enable accurate conclusions to be drawn, and this needs to be stated clearly.

Response: We have now included the number of replicates and technical duplicates in the figure legends. However, NMR and ITC titrations were not repeated due to the large amount of protein material and instrument time required for these experiments.

An FP assay is established using Cy5-labelled p62-LIR peptide as the probe, but it is not explained where this peptide binds on LC3, and it needs to be acknowledged that a lack of activity in this assay does not necessarily mean that the compounds do not bind to LC3/GABARAP as they could bind non-competitively at a different binding site on the protein.

Response: We now describe the binding of the p62 LIR peptide in the text. We have also included a statement that compounds that are not LIR competitive would not be detected by this assay. However, the additional tracer-independent assays (spectral shift assay – analogue of commonly used MST, ITC, SPR and NMR) allow us to exclude binding to other sites in LC3/GABARAP.

Most experiments reported here do not show compound binding to LC3/GABARAP, in contrast to previously published work. However, in this paper the authors labelled cysteine instead of lysine residues, when the data would be more comparable and the conclusions stronger if the same conditions were used.

Response: LC3/GABARAP proteins contain a number of Lys residues (LC3A - 9, LC3B - 10, LC3C - 9, GABARAP - 12, GABARAPL1 - 13, GABARAPL2 - 12). 2 lysines in LC3s (K49 and K51 in LC3A) and 2 lysines of GABARAPs (K46 and K48) are located within the LDS binding site and chemical modification of these could interfere with the binding of peptides/compounds. More specifically, Fan et al. showed that covalent modification of K49 (by molecules even smaller than a fluorescent tag) was sufficient to suppress peptide binding [PMID: 34590387]. We therefore consider labelling at cysteine residues a more prudent approach developing this assay format. We would like to mention that the authors of the paper reporting the development of ATTECs used 500 nM protein in their MST assay. This is a protein concentration about 1-2 magnitudes higher than usually used in MST assays. In addition, lysine labelling might lead to incorporation of several labels and signal quenching. As there is only one cysteine present, we avoided this complication. Finally, we confirmed our findings using label free methods (ITC, NMR, FP) which resulted in similar conclusions.

Line 91, the authors state 'no tool compounds such as autophagy pathway inhibitor are available', however, several autophagy related inhibitors are available such as 3-methyladenine, bafilomycin A, chloroquine and can be used in the field.

Response: We agree with the reviewer and have corrected the text to read "tool compounds based on direct LC3/GABARAP binding" instead of "autophagy inhibitors/enhancers" in general. This statement has been corrected in the text of the main manuscript.

Line 189 states 'To investigate covalent interactions of compounds 1 and 4 with LC3/GABARAP as recently reported for 1 with the E3 ligase DCAF11, we also studied the interaction of compound 1 and 4 with GABARAPL2 using ESI mass spectrometry, where no covalent modification was detected in vitro.' – reference to the data missing.

Response: We apologise for this omission. We have added a corresponding figure with the data (Supplementary Figure 7) and referenced it in the text.

Line 196 states 'For proteome-wide screening, we modified 1005 (4) and 8F20 (3) with PEG-based linkers that can be immobilized on sepharose beads to generate an affinity matrix for target pull-down, resulting in compounds 18 and 19. As expected, dose-dependent competition assays showed that the KIF11 inhibitor 8F20/Ispinesib (3) selectively bound to KIF11 in HEK293T lysates (EC50 of 290 nM). No additional targets were detected for 19 confirming excellent selectivity of this inhibitor for KIF11 (Fig.

3d).’ *Proteomics data is not shown anywhere in the paper. Fig 3d only shows competition data for compound 18 interacting with KIF11; how can it be stated that ‘No additional targets were detected’ if only one target has been monitored?*

The complete proteomics data is now accessible via the MassIVE database (Accession code: MSV000093528, Reviewer password: Schwalm-Rogov). Figure 3d shows the only one robustly identified dose-dependent competition curve from both experiments. This indicates that KIF11 is a specific binder of 19. The reported targets of 18 and 19, GABARAP and LC3 (MAP1LC3), were not enriched in the pull-down proteomics readout. This argues for a low binding affinity of the probes to the reported targets in this highly sensitive MS-based readout. Based on the reviewer's comments, we therefore rephrased the section describing these data:

Figure legend 3D:

Chemoproteomic competition assays for target deconvolution of 8F20 (**3**) and 10O5 (**4**). Affinity matrices for pulldown experiments were synthesised via amide coupling, generating (**18**) and (**19**) attached to (NHS-activated) Sepharose beads. Competition experiments were performed with free compound **3** and PEG-linked compound **4** at nine concentrations and residual binding was calculated relative to a DMSO control. Of the over 4000 proteins identified, only KIF11 showed robust dose-dependent binding to **19** (EC₅₀ = 290 nM). Both the **18**- and **19**-based affinity matrix assays did not enrich for the reported targets LC3/GABARAP in HEK293T cell lysate.

Main text:

The only Atg8 homolog for which we detected a weak interaction, GABARAPL2, was not pulled down by **18** up to a compound concentration of 30 µM, suggesting that the interaction with this Atg8 homolog was also weak in the cellular context. [...]

‘Druggable’ is frequently used instead of ‘ligandable’, for example, line 248 ‘Our screen confirmed that HP2 is the most druggable binding site on 248 LC3/GABARAP proteins surface, accommodating 10 from 21 identified fragments’. This is not true as the fragments have not been confirmed to bind by orthogonal techniques, and binding does not mean that the site is druggable (functional).

Response: We agree with this comment and have changed the text as suggested. We avoid the term druggability when referring to the fragments and use 'ligandability' instead.

Figure legends should be more specific and contain more information. For example, all FP data is described as ‘Fluorescence polarization assay curves’, but it appears that these are displacement curves so the probe should be stated each time.

Response: Thanks a lot for pointing this out. We changed the figure legends accordingly.

Has the data been corrected for probe background fluorescence?

Response: All depicted FP assay data were normalized and contain a correction for the background fluorescence.

Fig 4B states ‘Structural representation and corresponding ITC data for the two screening hits (compounds 20 and 21)’, what proteins were used here? Fig 4D does not state what each of the different colored spectra correspond to.

Response: We provide now information on the protein (LC3A) and color code to the figure captions (1:0.125 – orange, 1:0.25 – yellow, 1:0.5 – green, 1:1 – cyan, 1:2 – blue and 1:4 - magenta).

Insufficient data are provided on the virtual screen – where were the compounds docked on the proteins? Does the NMR data shown in Fig 4D, E agree with the initial docking site?

Response: We docked to the well-characterised LDS site that has already been targeted with novobiocin (PDB ID 6TBE). At the request of reviewer 1, we have included the docking poses and the binding site description in RR_Figure 1 and RR_Figure 4, respectively. Indeed, the binding site indicated in Figure 4E was consistent with the binding site used for docking. We have now included panel F in Figure 4, where we have highlighted the residues that showed significant chemical shift perturbations in our NMR titration. The two carboxylic acid groups present in both screening hits shared a similar binding mode in which this moiety interacted tightly with an arginine and a lysine residue. This binding mode explained the loss of affinity after substitution of the acid group.

Some experimental details are not clear: Chemistry experimental methods are given for compounds 19, 17, 18, 16 (in a random order), but schemes are missing, and it is not stated where the molecules used in this paper came from.

Response: We have described the compounds in the order in which they are mentioned in the manuscript. As the compounds used to generate the affinity matrix were precursors of the previously mentioned tracers, these compounds have a higher number. We have updated the Methods section by adding schemes and corresponding compounds in the text.

Experimental method 'Affinity determination using spectral shift mode on Dianthus', it is not clear what data/assay this refers to, is it MST?

Response: All experiments were done with the spectral shift method (PMID: 35171002) using fluorescently labelled protein(s). We clarified this in the relevant section of the Materials and Methods.

'Fluorescence polarization assay (FP assay): For the complementation assay' – what is the complementation assay, should it be displacement assay?

Response: Indeed, we performed displacement assays. Thank you for spotting this error.

'Covalent compound screening: For screening of covalent compounds, 46 compounds were tested against all LC3/GABARAPs, purified as described above' – what are these 46 compounds? Fewer compounds were discussed in the text. Does 'purified as described above' refer to the compounds or the proteins?

Response: We initially screened 46 analogues of published covalent LC3 ligands. However, we concluded that ligands that did not interact with LC3 were of limited use and now only include the validation data for compounds 5-7 and additional data for compound 1. We have removed the statement "purified as described above", which referred to the protein used, as the purification has been described earlier in the paper.

Crystallization – What protein construct was used?

Response: We used the protein expression construct LC3B1-120 in the T7 vector pNIC28-Bsa4. This protein has been successfully crystallized in our group previously. Information about the construct has been added to the method section.

Fragment screening and structure solution – what concentrations were the compounds soaked at? What cryo protectant was used?

Response: For fragment screening, the DSI compound library had a stock concentration of 500 mM in DMSO. For the first round of data collection, the crystals were soaked at 20% of the starting concentration (100 mM). Under these conditions, the high DMSO concentration (20%) damaged some crystals, and a second screening campaign was run at 50 mM fragment concentration. No additional cryoprotectant was used. We have now included these experimental details in the Materials and Methods section.

Minor points

FP graphs should use a shorter y axis scale (e.g. 0 - 120mP) to make the data easier to see.

Response: We adjusted all graphs accordingly using the 0-120 mP scale.

Fig2D the yellow and green spectra are very similar in color; it would be better to use a different color for one of them.

Response: We now changed the colour to a brighter yellow to make the differences clearer.

Compound 16 and 17 structures are buried in the supplementary and difficult to find, they should be referenced more clearly when mentioned in the text.

Response: As compound 16 and compound 17 were only briefly discussed, we decided to publish the structures of these compounds in the SI only.

Reviewer #4:

In this manuscript, Schwalm et al. focus on LC3/GABARAP-targeting ligands, characterising previously developed molecules and identifying novel chemical scaffolds. Protein recruitment to LC3/GABARAP through bifunctional small molecules, named autophagosome tethering compounds (ATTECs), has been previously explored as one of the different approaches to induce targeted protein degradation. The authors set out to validate existing LC3/GABARAP ligands used in ATTECs design, testing target engagement with different biophysical methods. The results reveal weak binding (in some cases no binding) of existing ATTEC ligands to LC3/GABARAP, highlighting the need to further characterise the cellular mechanism of action of ATTECs. In line with this observation, a recent study (Xue et al) shows that arylidene-indolinones, previously reported as LC3B binders, mediate protein degradation via DCAF11. In this context, it is certainly essential to perform a thorough characterisation of widely used chemical probes. However, in some parts of the manuscript, the authors seem to undermine any measured binding with recombinant LC3/GABARAP as weak or insignificant, although it is not yet clear what binding affinities are necessary to induce autophagy via ATTECs.

Before publication, the authors should address the following points:

Some examples of paragraphs where the authors should consider rephrasing statements dismissing observed binding: Lines 136-139, referring to Fig 2c and supplementary figure 2D;

Response: We have reworded some of the statements. The reviewer pointed out that the affinity required for ATTEC development is not known. This is certainly true, but if we do not see an interaction up to 100 μ M against the designated targets, affinities in the mM range would certainly not justify the claim of an LC3-dependent mechanism. We also believe that it is not clear that an autophagy-based degradation mechanism would be catalytic rather than stoichiometric, as the cargo is encapsulated for degradation and new cargo cannot be recruited in later autophagy cycles. However, these experiments would have to wait for confirmed and probably more potent Atg8 ligands.

Lines 144-145: "none of the compounds 1-4 resulted in significant CSP even at higher compound concentrations in agreement with our FP binding data." Supplementary figure 3 shows some shifts for compound (4). Also, please improve the layout/quality of this figure, as well as supplementary Fig. 5, as it is difficult to look at the NMR spectra.

Response: Indeed, titration of compound 4 into LC3B induces some CSP values for LC3B resonances, but these are significantly smaller compared to those for novobiocin (K_D to LC3B is \sim 10 μ M), indicating a weak interaction. We have amended the section describing these data

to reflect this. We have also revised the supplementary NMR figures (3, 5 and 10), which we hope will improve the presentation of the data.

oLines 169-178: despite observing micromolar binding of (4) with GABARAP family members, the authors define it again as not significant.

Response: As discussed above, titration of 1005 (4) into GABARAPL2 induced prominent CSP for several GABARAPL2 resonances, but the affinities were too weak to be determined by NMR or other biophysical methods. We therefore defined interactions in the >50 μM range as weak and, given the nature of the ligands, unlikely to be selective.

oLines 186-187: "raising the possibility of further off-targets within the proteome, based on the reactivity of these compounds". The authors should not make this claim without supporting data.

Response: We have reworded this section. The interactions of compound 5-7 are not selective within the LC3/GABARAP subfamily, probably due to the high reactivity of this electrophile. We have not assessed proteome-wide selectivity and describe this interaction therefore as unselective inhibition of human Atg8 family proteins.

MST experiments: the compound labelling in supplementary figure 4a should be corrected. Did the authors test (12) or (13) or any of the peptides using MST? do they show binding?

We have corrected the labelling of all compounds in Supplementary Figure 4a and included positive controls. RR_Figure 7 shows an example of a TRIC (temperature related intensity change) experiment (analogue of MST) showing the binding of LC3A protein to the canonical p62 LIR peptide resulting in a K_D of 2 μM , in agreement with literature values.

Ligand	Highest Ligand Conc. [mM]	Buffer	Target Conc. [nM]	K_D [mM]	ΔF_{norm} [%]	S/N Ratio	Saturation
p62-LIR	0.051	10 mM Tris 0.05% Tween	5	0.00184	-9.4	9.4	96.5 %

RR_figure 7: Binding data of LC3A:p62 LIR interaction using the TRIC technology. The table shown in the lower panel provides all experimental parameters as well as determined K_D values.

Lines 155-157: “Novobiocin (12) revealed binding with a K_D (6.7 μM for LC3A), while titrations with AN2 (2) and 8F20 (3) did not yield significant binding heats”. These ITC experiments suggest no binding or weaker binding (not measurable at the concentrations tested). Dong et al. were able to measure binding of (3) using ITC ($K_d \sim 12 \mu\text{M}$) at higher protein and compound concentrations. Did the authors try performing the ITC titration in those conditions?

Response: In fact, Dong et al. included ITC data showing apparent binding. However, the authors used 1.5 mM protein concentration. We have not been able to achieve such a high protein concentration. In addition, the compound has limited solubility, making it unlikely, in our opinion, that the heat effects are due to binding heats. We speculate that the high protein concentrations resulted in significant dilution heats which were not corrected in the blank titrations. However, as we have not been able to reproduce these data, we can only speculate about the nature of the published titration data.

Regarding compounds (4) and (1): are they tested as E/Z mixture? was the mixture purified in previous papers? is one of the two isomers more active? Tracer (16) is instead represented as pure isomer in Supplementary figure 4C. If the (4) E/Z isomers bind differently, this will result in lower effective concentration of the ‘active’ isomer and thus affect its ability to displace the tracer. Additionally, in comparison to tracer (17), (16) contains an α,β -unsaturated carbonyl portion next to the Py-BODIPY moiety, is that a drawing mistake?

Response: We tried to purify the isomers as we suspected that a single isomer was the active form. Unfortunately, even after isolating the individual isomers, the purified compound quickly isomerised. This is a known property of the oxindole scaffold. The HPLC trace is shown in RR_Figure 8. The single isomer shown in Figure 4C has now been corrected along with the α,β -unsaturated carbonyl moiety. We thank the reviewer for pointing out this error.

RR_figure 8: Analysis of the E/Z isomerization by HPLC.

• Supplementary fig. 4D: did the authors try lower tween concentrations? or, in the absence of tween, displacement with one of their validated ligands (Novobiocin or peptides)?

Response: Based on previous experiments we knew that the use of Tween was important in suppressing unspecific binding. However, the experiment (protein titration and displacement) was initially performed to reproduce the titration reported by Dong et al. in the absence of Tween. Under these conditions, we confirmed the tracer binding reported by these authors. However, we routinely perform displacement experiments to control for unspecific binding and found no displacement using the parent ligands of the tracers. To avoid unspecific binding, we used 0.05% Tween, which suppressed unspecific interactions and the apparent FP signal detected in Tween-free buffer. As we concluded that the compound-based tracers did not monitor specific bind to the protein, we did not perform further FP experiments with these tracers. Due to the lack of small molecule tracers, we subsequently established the p62 LIR-based fluorescence polarisation assay.

• *Lines 196-206: This section describing pull-down experiments requires some clarifications (also in the corresponding methods section) and additional controls. The authors should show a positive control (e.g. using a peptide or one of the better binders) to demonstrate that it is possible to measure enrichment of LC3/GABARAP with this setup. Also, what does it mean that there was no enrichment “using 18 (up to 30uM)” if the compound is conjugated to Sepharose beads?*

Response: We apologise for not describing the assay well enough. We performed this assay in a ligand displacement format, i.e. immobilised proteins binding to the biotin adducts were eluted with the original ligands. We agree that a positive control would be necessary to prove that the probes do not bind to LC3/GABARAP in the competition assay setup. However, the successful pulldown and competition of the reported target KIF11 confirmed that the assay performed well in principle by identifying the known target of this inhibitor. We have attempted to clarify the description of the pull-down assay by amending the figure legend and the section describing these data in the manuscript.

• *It would be helpful to include a description of the NanoBRET assay in the main text. The authors say that there is a moderate BRET increase but no displacement. It would be helpful to further confirm this by testing displacement with one of the stronger binders.*

Response: We introduced the assay in the revised version and modified this section: BRET is a proximity assay between a fluorescent donor (in this case full length LC3A and LC3B) and a fluorescent acceptor (the dye adducts of 3 and 4). BRET occurs when the donor and acceptor are in proximity (<10 nm) and its fluorescence intensity is inversely proportional to the distance of the donor-acceptor pair. This technology has gained popularity as a live cell assay format for monitoring protein-protein interactions or target engagement with small molecules. As expected from our biochemical data, we observed no BRET signal using the 8F20 and 10O5 dye analogues (compounds 16 and 17) (Supplementary Figure 8a-c).

• *Supplementary fig. 10 - line 196: “Crystallized fragments bound to LC3A” contradicts the rest of the text, where it’s LC3B (e.g. page 39 line 738)*

Response: Thanks for spotting this error. We used LC3B for all crystallization studies and corrected this error.

• *Did the authors try to soak/co-crystallise any of the previously described ligands with LC3/GABARAP? Some of the compounds showing weak binding in this manuscript could potentially be useful starting points for optimisation, similarly to the newly identified fragments.*

Response: We used compounds 1, 4, 9, 11, LY22398, TH152 and TH200 in soaking experiments, but only obtained apo structures or weakly diffracting crystals. To support further studies on rational ligand design, we now include our docking model and highlighted residues with significant chemical shift perturbations in NMR (panel F in Figure 4). As discussed above, the COOH group of our docking hits showed a similar binding mode, interacting with an arginine and a lysine residue.

Reviewer #5:

The manuscript by Schwalm et al. "Targeting LC3/GABARAP for degrader development and autophagy modulation" is an impressive dualistic exercise where on one hand the authors rigorously test published ATTECs (autophagosome tethering compounds) for binding to LC3/GABARAP family proteins, and on the other hand perform in silico docking and large scale crystallographic fragment screening to identify a group of compounds they validate as ligands for future ATTEC development. Perhaps surprisingly, and certainly disappointingly, the authors find that the published ATTECs do not bind at all (almost all tested compounds), or too weakly, to LC3/GABARAPs for a specific biological effect to be explained. This is a very important finding to be communicated pointing to a too poor characterization and binding validation effort done in the published work on ATTECs: This is clearly damaging to the field as such and cries for better evaluation of such papers before they are accepted for publication. In the second part of their paper the authors use in silico docking and large scale crystallographic fragment screening to identify a number of compounds that do bind to the LC3/GABARAPs and also identify two new binding surfaces. A conclusion is also that it is the hydrophobic pocket 2 (HP2) that is the most druggable in the LIR Docking Site (LDS) while HP1 interacting with the aromatic residue of the core LIR is much less accessible for the compounds.

Response: We thank the reviewer for this summary.

This is clearly an important paper with thoroughly performed experiments using an impressive array of assays to validate binding both in vitro and in cells in addition to structure explorations techniques like NMR and x-ray crystallography. The "negative" half of the paper, showing that published ATTEC compounds basically do not bind to LC3/GABARAPs, deserves to be communicated widely in a very visible journal. The "positive" half of the paper stops when it starts to become exciting. A number of candidate compounds are presented that can be used to develop new ATTECs are identified, but we do not see any of these taken to the next level and we are left wondering: Is ATTECs at all a viable approach? For the paper to be acceptable for publication in Nature Communications the authors need to go further with one or more of the compounds they have identified and test in a proof of principle setting if they can see targeted degradation using such a compound equipped with a target recognition part connected with a linker to the LC3/GABARAP interacting compound.

Response: We agree with the reviewer that validation with LC3/GABARAP is still lacking. However, we wanted to avoid using non-optimal compounds to develop an LC3/GABARAP-based degrader. The *in silico* hits are probably the most potent ligands currently known, but the COOH group prevents efficient cell penetration. We have developed a series of compounds based on the novobiocin hit with the central coumarin scaffold (PMID: 33769048), but even these ligands are still weak (>20 μ M), suggesting that high concentrations will be required to achieve target engagement in cells. The use of high concentration ligands is the main reason why many chemical biology studies report inaccurate phenotypic responses. It is therefore our aim to develop the first ligands that meet the criteria of chemical probes (with on-target potencies better than 100 nM). This will require the synthesis and evaluation of hundreds of compounds. As the field of degraders is developing rapidly, we believe it is important to share negative data to prevent research groups from developing ATTECs based on current ligands. To date, 29 manuscripts have been published using ATTEC ligands. We therefore believe that the publication of negative data is important for the advancement of the field.

Reviewer #6:

In the paper titled 'Targeting LC3/GABARAP for degrader development and autophagy modulation' by Martin P. Schwalm et al. The authors shed light on the principles of autophagosome tethering compounds (ATTECs) based on the target recruitment to LC3/GABARAP. The researchers assessed the published ATTEC ligands using biophysical techniques including FP, ITC, and NMR. The authors note their surprise that the literature identified ligands do in fact not bind to their intended target LC3. They then go forth to use their world leading expertise to identify a number of small molecule ligands

through virtual screening and high-throughput fragment via X-ray crystallography. Presenting a diverse comprehensively validated series of ligands for future ATTEC development. The present study is well designed, and the authors have given a comprehensive insight into the binding mechanism of ATTEC compounds. The result of this study is of great use to the drug discovery community not only from the ligands for future ATTEC development but also from reagents produced in the LC3/GABARAP systems and experimental methods. I would suggest this study for publication with some minor alterations and points of discussion.

Response: We thank the reviewer for encouraging us to publish this work.

Minor Points:

1) When debunking a previously reported study it is difficult to be completely resolute in biophysical data, as a number of factors, such as PTM and binding partners may play a roll in the published ATTEC ligands that are hard to replicate in a reconstituted cell-free/cellular manner. Although the studies data is strong and very compelling, I would suggest a sentence to this extent.

Response: We agree with the reviewer that PTMs or other components may increase ligand affinity. We have therefore included MS pull-down experiments and cellular BRET-based assays in our ligand characterisation. As suggested, we have added a sentence in the Discussion section highlighting the modulation of target binding in the context of the cellular environment.

2) In a pleasant way, it is hard to read the publication and not ask “what’s next?” I assume this is out of the scope of this study and would be a major addition to the publication.

Response: This is quite right. We have also discussed the issue of developing ATTECs in our response to reviewer 5. We strongly believe that a potent and well validated LC3(GABARAP) ligand would be required for further studies and these activities are ongoing in our laboratory.

i) Building on the positive outcome from the virtual screen could the authors possibly resolve a structure of TH152 bound to LC3B

Response: Following the identification of TH152 as a potential ligand, we immediately initiated crystallisation approaches on LC3B, LC3A and GABARAPL2 proteins. However, co-crystallisation did not yield protein crystals and soaking experiments resulted in poorly diffracting crystals. However, we have now included the docking model confirmed by our NMR data in the revised manuscript.

ii) Could the assays built within the invalidation of the published ATTEC ligands be used against the crystallographic output, even if they are likely to be weak binders?

Response: We used the developed FP assay as well as SPR to determine the binding constants of the identified fragments and their closest homologs. However, the ligands identified in the crystallographic fragment screening were still too weak for accurate determination of K_D s. We estimate the affinities to be in the high micromolar or even millimolar range. The data are summarised in RR_Figure 9.

All compounds were tested in the FP displacement assay, where fluorescence-labelled p62 LIR peptide was displaced from LC3B and GABARAPL2 (RR_Figure 9). The data was normalised to p62 LIR displacement at 100 μ M to yield a relative readout. Hence, compounds below the threshold set by the p62 LIR could be potential binders.

RR_Figure 9: FP values normalised to p62 LIR binding at 100 μM concentration. The actual compound concentration (500 or 1000 μM) is indicated on each plot. The FP values above the normalisation (red line) did not provide data that allowed the K_D to be determined.

As follow up, the compounds were subjected to SPR measurements and screened in a one-shot format at 250 μM . Only 2 compounds (F15 and F17) gave data that allowed us to estimate K_D values in the high micromolar or low mM K_D range.